

# Goldstone equivalence and
# high energy Eeectroweak physics

Gabriel Cuomo[1], Luca Vecchi[1,2*] and Andrea Wulzer[1,2,3,4]

**1** Theoretical Particle Physics Laboratory (LPTP),
Institute of Physics, EPFL, Lausanne, Switzerland
**2** Istituto Nazionale Fisica Nucleare, Sezione di Padova, Italy
**3** CERN, Theoretical Physics Department, Geneva, Switzerland
**4** Dipartimento di Fisica e Astronomia, Università di Padova, Italy

⋆ luca.vecchi@epfl.ch

## Abstract

The transition between the broken and unbroken phases of massive gauge theories, namely the rearrangement of longitudinal and Goldstone degrees of freedom that occurs at high energy, is not manifestly smooth in the standard formalism. The lack of smoothness concretely shows up as an anomalous growth with energy of the longitudinal polarization vectors, as they emerge in Feynman rules both for real on-shell external particles and for virtual particles from the decomposition of the gauge field propagator. This makes the characterization of Feynman amplitudes in the high-energy limit quite cumbersome, which in turn poses peculiar challenges in the study of Electroweak processes at energies much above the Electroweak scale. We develop a Lorentz-covariant formalism where polarization vectors are well-behaved and, consequently, energy power-counting is manifest at the level of individual Feynman diagrams. This allows us to prove the validity of the Effective *W* Approximation and, more generally, the factorization of collinear emissions and to compute the corresponding splitting functions at the tree-level order. Our formalism applies at all orders in perturbation theory, for arbitrary gauge groups and generic linear gauge-fixing functionals. It can be used to simplify Standard Model loop calculations by performing the high-energy expansion directly on the Feynman diagrams. This is illustrated by computing the radiative corrections to the decay of the top quark.


# 1 Introduction

Studying Electroweak physics in reactions where the available center of mass energy $E$ is much larger than the Electroweak scale $m \sim m_{W,Z}$ is of both practical and theoretical interest.

The practical relevance stems from the fact that the LHC and its high-luminosity successor will allow us to take a first glance at this new energy regime, which furthermore is a promising one for the search of new physics. Indirect new physics searches by precise measurement of high-energy Electroweak processes deserve a special mention in this context because they require accurate Standard Model (SM) predictions. Radiative Electroweak corrections are enhanced at high energy [1–4], due to peculiar non-canceling IR effects that produce single or double (Sudakov) logarithms of $E^2/m^2$. Therefore, even if the accuracy of Electroweak LHC measurements above the TeV scale cannot go below few percent because of the limited statistics, including state-of-the-art calculations (at one-loop order) for such corrections is compulsory and going beyond the state-of-the-art would be desirable. The need for refined Electroweak calculations will dramatically increase at future colliders probing even higher energy scales. A particularly striking case can be made for a hypothetical tens-of-TeV muon collider, where the QCD corrections have limited impact and the final accuracy of our predictions will be driven by our ability to deal with Electroweak physics.

High-energy Electroweak interactions are also relevant theoretically, in connection with the general problem of IR physics in Quantum Field Theory (QFT). In the presence of a large scale separation $E \gg m$, it must be possible to visualise the reaction in terms of a hard scattering process dressed by soft radiation from the initial and from the final states. The hard scattering

is the component of the process that genuinely takes place at energy $E$ and therefore it probes physics at the shortest accessible distance $1/E$. The radiation emerges instead from energies ranging from some upper scale much below $E$ down to the IR scale $m$. It is insensitive to the details of the hard scattering process and to short-distance physics, and it takes a universal form. Conversely, the hard scattering should be nearly insensitive to the long-distance IR physics at the scale $m$, which should appear in that component of the process as a tiny power-like $m/E$ correction. The factorized picture above is supported by a number of results in QED and QCD, and by decades of QFT practice. Therefore is must undoubtably hold true for Electroweak physics as well. However it is not easy to substantiate the picture in the case of Electroweak interactions and to materialise it in a recipe for concrete calculations. The most challenging aspect of the problem is arguably the non-applicability [5] of the KLN theorem [1] to Electroweak interactions, because in "normal" QFT's this fundamental result is the starting point for the definition and the physical interpretation of the hard IR-insensitive component of the reaction. The IR problem for Electroweak physics is, in this respect, even more challenging than for QCD. But on the other hand it must be much simpler because Electroweak interactions stay perturbative at the IR scale $m$. Therefore it should be possible to address the problem in fully rigorous terms and to end up with accurate first-principle predictions without the need of extra phenomenological input unlike in QCD. High-energy Electroweak physics is an interesting corner of QFT and its many aspects continuously stimulate theoretical work (see e.g. [9–22]). Further interest and results are expected in the future.

In this paper we extensively and conclusively study one peculiar technical aspect of high-energy Electroweak physics, related to the characterization of the energy behavior of the Feynman amplitudes. Namely we ask ourselves whether and how this behavior can be systematically captured by a simple power-counting rule. Power-counting would not only inform us on the leading high-energy behavior of each process by inspecting its Feynman diagrams. It would also allow us to isolate the diagrams that give the leading contribution and to treat the others as perturbations in a well-organized $m/E$ expansion. Furthermore, controlling the energy behavior is essential in order to separate the short and long distance components of Feynman amplitudes. This in turn is a prerequisite to prove even the simplest (tree-level) version of any factorization theorem. Notice that in order to be useful, especially for the latter type of applications, power-counting should also hold for diagrams whose external legs are not exactly on the mass-shell of a physical particle.

Making power-counting manifest for the Electroweak theory requires us to depart slightly from the standard formalism for massive gauge theories. Indeed power-counting is notoriously hidden in standard massive gauge theory diagrams because of the anomalous $E/m$ behavior of the wave-functions associated to longitudinally polarized spin-one particles. The problem shows up already in the simplest textbook example of a high-energy Electroweak process, such as the scattering of longitudinal (i.e. helicity $h = 0$) vectors $V_0 V_0 \to V_0 V_0$, at the tree-level order. Longitudinal polarization vectors grow with energy as $\varepsilon_\mu^{h=0} \simeq k_\mu/m \sim E/m$ and since four of them are involved the diagrams containing gauge self-interaction vertices grow with energy as $E^4$. Diagrams with the Higgs have two inverse powers of energy from the propagator and grow like $E^2$. The energy growth cancels when all diagrams are summed up and the final physical amplitude scales like $E^2$ in the energy range (which of course would have existed only if the Higgs was heavy) from $m$ to the Higgs mass, and as $E^0$ afterwards. Similar cancellations take place in almost all high-energy processes involving longitudinal vectors.

Power-counting-violating cancellations are problematic. First of all, they prevent us from neglecting masses in the amplitude calculation, even if we were interested in the deep high-

---

[1]The theorem ensures the cancellation of IR divergences (or of IR enhancements, when a physical IR cutoff is present) in observables that are inclusive on the radiation [6, 7]. A weaker result, specific to soft singularities, is known as the Bloch-Nordsieck theorem [8] and it is also not applicable in Electroweak physics.

energy regime where they are expected to be (and eventually are) negligible in the final result. Of course modern tree-level calculation technologies easily allow us to make computations with finite masses, however at higher loop orders dealing with massless rather than massive integrals could be a crucial simplification. Second, cancellations of $E/m$ spurious enhancements are problematic because they only occur if the vector bosons momenta are on-shell, i.e. when their virtuality $Q^2 = k^2 - m^2$ vanishes. But the vector bosons are never exactly on-shell in the sub-diagrams that we would like to interpret as describing the hard scattering in factorization problems. In that case $Q$ is of order or much larger than $m$, while still much below the hard scale $E$. The expansion parameter that should ensure factorization as the product of a soft splitting of virtuality $Q^2$ times the hard short-distance reaction with on-shell vector bosons is indeed $Q^2/E^2 \ll 1$, which does not require $Q \ll m$.[2] Because of the cancellation, subleading terms in the $Q^2/E^2$ expansion of the off-shell amplitude can actually be of order $E^2/m^2 \cdot Q^2/E^2 = Q^2/m^2$ relative to the on-shell amplitude, i.e. not small at all. The latter terms do eventually get canceled in the total amplitude by seemingly unrelated diagrams where no low-virtuality vector boson is propagating in the internal lines, so that factorization holds. An explicit example of this behavior was discussed in Ref. [23] to illustrate the difficulty of proving the validity of the Effective $W$ Approximation (EWA) in the standard covariant formulation.

Having elaborated at length on the potential virtues of a formalism where longitudinal polarization vectors do not grow with energy, we discuss now how to get one by "Goldstone Equivalence". Goldstone Equivalence is the idea that at high energy the longitudinal degree of freedom of a massive vector gets transferred to the scalar degree of freedom associated with the excitations of the corresponding Goldstone field. This idea is supported by the famous Goldstone Boson Equivalence Theorem [24]. The aim of the present work is to turn Goldstone Equivalence into a rigorous exact reformulation of Feynman rules for massive gauge theories in which the polarization vectors are well-behaved and power-counting is manifest. Notice that ours is not quite a "different" formalism. It employs the exact same gauge-fixed Lagrangian as the standard one so that the Feynman rules for the vertices are standard, and all the standard calculation technologies straightforwardly apply. Only the longitudinal polarization vector is modified and acquires, as we will see, a component in the direction of the Goldstone field. Diagrams with external Goldstone legs are thus included in the calculation and their contribution is typically (but not always) leading at high-energy compatibly with the Equivalence Theorem. The standard longitudinal polarization vectors need to be replaced with well-behaved ones also in off-shell amplitudes because we need power-counting also for the latter diagrams in order to approach factorization problems, as we discussed. Moreover the SM vector bosons are unstable and therefore they are never exactly on-shell. The polarization vectors for off-shell and for unstable bosons are technically defined by the decomposition of propagator lines that connect two otherwise disconnected diagrams. An integral part of our formalism is thus the decomposition of such propagators in terms of well-behaved polarization vectors.

It is rather intuitive why Goldstone Equivalence allows us to get rid of the energy-growing polarization vectors. The wave-function factor for scalars is a constant, therefore by replacing the longitudinal bosons with the Goldstone scalars we should be able to turn the $E/m$ behavior into a constant one. In Ref. [25] one of us elaborated on this idea showing that energy growth is avoided by a suitable definition of the state that describes longitudinal vectors in the enlarged Fock space of the gauge-fixed theory. This differs from the standard one by a BRS-exact state so that it belongs to the same element of the BRS cohomology and consequently it possesses identical physical properties. The additional BRS-exact state contains one quantum of the Goldstone field, which gives rise to the previously-mentioned Goldstone component of the longitudinal state wave function. The BRS-exact state also contains a scalar excitation of the massive vector. Its wave function, proportional to $k^\mu/m$, combines with the one of the

---

[2]$Q$ is much smaller than $m$ only for decay processes.

ordinary longitudinal state and cancels the energy growth. Here we proceed in a slightly different way. We employ the standard representative states for the physical particles in the Fock space and we get rid of $k_\mu/m$ at a later stage of the scattering amplitude calculation. We do so by exploiting the generalized Ward identity that relates amputated amplitudes for the gauge field, contracted with the external momentum $k_\mu$, to Goldstone bosons amplitudes. Our approach might sound less appealing than the one of Ref. [25], but it is not. Indeed while adding a Goldstone component to the longitudinal states is in line with the intuitive picture of Goldstone Equivalence, it should be kept in mind that the specific representative of the BRS-cohomology element we decide to employ is deprived of any physical meaning. The present approach brings several advantages. It allows us to deal with a general gauge theory and in particular with the SM, while Ref. [25] only studied a toy model. It also allows us to deal with unstable particles, such as the physical $W$ and $Z$, and with off-shell vector bosons. Finally, the approach of Ref. [25] requires a specific choice of the gauge-fixing parameters, unlike ours.

The rest of the paper is organized as follows. In Section 2 we work out our Goldstone Equivalence formalism for a simple toy model. This will allow us to present the various steps of the derivation avoiding at first the extra algebraic and notational complications required to deal with a general spontaneously broken gauge theory, which we study in Section 3. The result essentially consists of an expression for the longitudinal (zero helicity) polarization vector $\mathcal{E}_M^0[k]$, with components $M = \mu$ along the gauge fields and a component $M = \pi$ along the Goldstone scalars. The gauge component $\mathcal{E}_\mu^0[k]$ does not grow with energy anymore, but rather it vanishes as $m/E$. Furthermore it takes a universal theory-independent form. The scalar component $\mathcal{E}_\pi^0[k] = \mathcal{E}_\pi^0(k^2)$ only depends on $k^2$ and thus it is constant in energy at fixed vector boson virtuality. It is given by a certain combination of vacuum polarization amplitudes, to be computed in each theory and for each external vector boson. One-loop explicit expressions for $\mathcal{E}_\pi^0(k^2)$ for the SM vector bosons are computed and reported in Section 4. Section 4.4 and 5 are devoted to applications. In Section 4.4 we apply our formalism to tree-level longitudinal vector bosons scattering and to the calculation of radiative corrections to the $t \to b\,W$ top quark decay. This has the purpose of illustrating the formalism and outlining the advantages of a manifest power-counting rule, and also of verifying in non-trivial examples that our approach produces results that are exactly identical to the standard ones. In Section 5 we instead use our formalism to derive the simplest possible "factorization theorem". Namely we show that collinear emissions factorize at tree-level into universal splitting amplitudes times the hard process amplitude. We saw above that proving this seemingly trivial fact, of which the EWA is a particular case, requires a formalism like ours where power-counting is manifest. Finally, we report our conclusions in Section 6.

## 2 Warm-up: The Higgs–Kibble Model

We begin discussing Goldstone Equivalence within the so-called Higgs–Kibble model (see e.g., [26, 27]), namely a SU(2) gauge theory fully broken by the vacuum expectation value (VEV) of a scalar doublet $H$. This will allow us to illustrate the logic of our derivation and to explain the result in a simple context, in preparation for the general discussion of Section 3. Before gauge-fixing, the Lagrangian simply reads

$$\mathcal{L}_0 = -\frac{1}{2}\mathrm{Tr}\left[W_{\mu\nu}W^{\mu\nu}\right] + (D_\mu H)^\dagger D^\mu H - \lambda \left(|H|^2 - \frac{\mu^2}{2\lambda}\right)^2 , \tag{2.1}$$

where $W^\mu = W_a^\mu \sigma^a/2$ ($a = 1, 2, 3$) are the gauge fields and the Higgs doublet is represented as [3]

$$H = \frac{1}{\sqrt{2}} \begin{pmatrix} -i(\pi_1 - i\pi_2) \\ v + h + i\pi_3 \end{pmatrix}. \tag{2.2}$$

The parameter $v$ is the Higgs VEV, $h$ is the physical Higgs scalar and $\pi_a$ the "eaten" Goldstone bosons. Notice that in the present section we consider bare fields, Lagrangians and parameters. The scalar fields $h$ and $\pi_a$ are defined to have zero VEV, therefore $v$ is not equal to $\mu/\sqrt{\lambda}$ beyond tree-level. The spectrum of the theory consists of 3 massive vectors and the Higgs scalar, with tree-level masses $m_0^2 = g^2 v^2/4$ and $m_{h,0}^2 = 2\lambda v^2$.

The standard Faddeev–Popov method allows us to turn the Lagrangian $\mathcal{L}_0$ into a concrete recipe for perturbative calculations. One introduces a ghost $\omega^a$ and an anti-ghost field $\overline{\omega}^a$ for each gauge vector, and a gauge-fixing term $\mathcal{L}_{\text{g.f.}}$, producing a Lagrangian

$$\mathcal{L} = \mathcal{L}_0 + \mathcal{L}_{\text{g.f.}} + \mathcal{L}_{\text{ghosts}}, \tag{2.3}$$

which is now suited to be studied in perturbation theory. The gauge-fixing term is given by

$$\mathcal{L}_{\text{g.f.}} = -\frac{1}{2} \sum_a \mathcal{F}_a(x) \mathcal{F}_a(x), \tag{2.4}$$

where $\mathcal{F}_a$ are three real gauge-fixing functionals, one for each local symmetry generator. The ghost Lagrangian is $\mathcal{L}_{\text{ghosts}} = -\bar{\omega}^a \delta_\omega \mathcal{F}_a$, where $\delta_\omega$ represents an infinitesimal gauge transformation with ghost parameters. The explicit form of $\mathcal{L}_{\text{ghosts}}$ will not be relevant in the following.

Throughout this work we restrict our attention to gauge-fixing functionals that are linear combination of the 4-divergence of the vector fields and of the scalars in the theory. In Section 3 we will deal with the most general gauge-fixing in this class, however for the illustrative purposes of the present section we consider the particular case

$$\mathcal{F}_a = \partial_\mu W_a^\mu / \sqrt{\xi} - \sqrt{\xi}\, \tilde{m} \pi_a, \tag{2.5}$$

where $\xi > 0$ and $\tilde{m}$ are free parameters. In particular, $\tilde{m}$ is not necessarily related to the mass of the vector bosons. The convenience of this gauge-fixing choice stems from the fact that the Lagrangian $\mathcal{L}_0$ in eq. (2.1) enjoys an exact global custodial $SU(2)_c$ symmetry under which $W_a^\mu$ and $\pi_a$ transforms as triplets, while $h$ is a singlet. The gauge-fixing functional in eq. (2.5) preserves custodial symmetry, making its implications manifest in the gauge-fixed theory.

## 2.1 Useful Identities

Spontaneously broken (or exact) gauge theories are among the most studied subjects in theoretical physics. Of this huge body of literature we review here only the results that are directly relevant for our discussion, starting from the Slavnov-Taylor identities that control the matrix elements of the gauge-fixing functional operators. We next study the implications of these identities on the amputated Feynman amplitudes, deriving "generalized Ward identities" that are the analog of the familiar QED Ward identities $k_\mu \mathcal{A}^\mu = 0$. The latter identities will be used in Sections 2.2 and 2.3 to get rid of the growing-with-energy longitudinal polarization vectors. The Slavnov-Taylor identities are presented for an arbitrary gauge theory, while their implications are discussed in the particular case of the Higgs–Kibble model. However the derivations are presented with a logic that allows for a relatively straightforward generalization.

---

[3] We also defined $W_{\mu\nu} = \partial_\mu W_\nu - \partial_\nu W_\mu - i g [W_\mu, W_\nu]$ and $D_\mu H = \partial_\mu H - i g W_\mu H$.

**Slavnov-Taylor Identities**

The starting point of our derivation is the set of identities [28]

$$\langle\beta| T\left\{\mathcal{F}_{a_1}(x_1)\cdots\mathcal{F}_{a_n}(x_n)O\right\}|\alpha\rangle \propto \prod \delta^{(4)}(x_i - x_j), \tag{2.6}$$

where $|\alpha\rangle, |\beta\rangle$ are arbitrary physical *in* and *out* states, $O$ is any gauge-invariant operator and $\mathcal{F}_a$ are the gauge-fixing functionals. These are given by eq. (2.5) in the particular case of the Higgs–Kibble model, but the equation above is of fully general validity. The r.h.s. of the equation consists of contact terms that do not contribute to connected components of the correlators in Fourier space. Therefore the equation ensures the cancellation of connected diagrams with any number $n \geq 1$ of gauge-fixing operator $\mathcal{F}_a$, for arbitrary physical particles on the external legs and possibly the insertion of a generic gauge-invariant operator. A particular case of eq. (2.6), for which we will need to know the pre-factor, is

$$\langle 0| T\left\{\mathcal{F}_a(x)\mathcal{F}_b(y)\right\}|0\rangle = -i\delta_{ab}\delta^{(4)}(x - y). \tag{2.7}$$

We will see later the implication of the above relation for the bosonic 2-point correlators.

The identities (2.6) are so important for our work that it is worth justifying their validity here, on top of relying on the proof in Ref. [28]. Following Ref. [29], we recall that the Faddeev-Popov method establishes the independence of the path integral of gauge-invariant operators on the choice of the gauge-fixing functionals. In particular we can consider a shift $\mathcal{F}_a \to \mathcal{F}_a + J_a$, with $J_a(x)$ a field-independent local source. Because $J_a$ is field-independent the ghost action is not affected, so the only change in eq. (2.3) appears in the gauge-fixing term. Independence of $J_a$ thus implies that any number of functional derivatives of

$$\int \mathcal{D}(\text{fields})\, O\, e^{i\int d^4x\left[\mathcal{L}_0 + \mathcal{L}_{\text{ghosts}} - \frac{1}{2}(\mathcal{F}_a + J_a)^2\right]}, \tag{2.8}$$

with respect to $J_a$ vanishes. It is now a trivial exercise to reproduce eq. (2.7). The more general form of the identity in eq. (2.6) follows from the non-trivial fact (see e.g., [30]) that any physical particle can be excited from the vacuum by a gauge-invariant operator.

**Generalized Ward Identities**

The Slavnov-Taylor identities hold for connected amplitudes. Turning them into relations for the amputated Feynman amplitudes is conceptually straightforward, since the amputated amplitudes are simply obtained from the connected ones by factoring out the propagators on the external legs. However taking this step requires us to get some control on the structure of the propagator (or, more precisely, of the all-orders two-point function), and to establish some notation.

In a general theory, all the scalar fields can mix with the gauge vectors and the two-point function we are seeking is a $(4N_V + N_S)$-dimensional matrix, where $N_S$ and $N_V$ are, respectively, the total number of scalar and of vector fields that are present in the theory. Clearly Lorentz invariance implies a number of simplifications on the structure of the matrix, still leaving however a rather complicated structure we will have to deal with in the next section. The situation is much simpler in the Higgs–Kibble model. The custodial $SU(2)_c$ symmetry implies that the singlet $h$ does not mix with the $W_\mu^a$ and $\pi_a$ triplets, and furthermore the two-point functions in the $W/\pi$ sector are proportional to the identity in the custodial indices space. This allows us to treat separately each of the 3 gauge fields together with its corresponding Goldstone and to suppress the custodial indices altogether. The fields are collected in a 5-components vector

$$\Phi_M = \left(W_\mu, \pi\right), \tag{2.9}$$

where $M = \{\mu, \pi\}$ runs over the four Lorentz indices $\mu$ and on a fifth (Goldstone) component $M = \pi$. Vectors with upper 5D indices are defined by acting with a 5D metric $\eta^{MN} = \text{diag}(\eta^{\mu\nu}, 1)$. With this notation the two-point function matrix in momentum space is defined as [4]

$$i\,G_{MN}[k] \equiv \int d^4x \; e^{ik_\mu x^\mu} \langle 0 | T\{\Phi_M(x)\Phi_N(0)\} | 0 \rangle. \tag{2.10}$$

Notice that from Bose statistics and translation invariance follows that

$$G_{MN}[-k] = G_{NM}[k]. \tag{2.11}$$

For a given 4-momentum vector $k^\mu$, we introduce in the 5D space a transverse projector $\mathcal{P}^\perp[k]$ and two longitudinal vectors $\mathcal{P}_i[k]$. Taking the index $i$ to run over two values $i = \text{V}, \text{S}$ denoting "vector" and "scalar", we define

$$\mathcal{P}^\perp_{MN} = \begin{pmatrix} \eta_{\mu\nu} - \dfrac{k_\mu k_\nu}{k^2} & \mathbf{0}_{4\times 1} \\ \mathbf{0}_{1\times 4} & 0 \end{pmatrix}, \tag{2.12}$$

$$\mathcal{P}_{\text{V}M} = \left( -i\dfrac{k_\mu}{k}, 0 \right),$$

$$\mathcal{P}_{\text{S}M} = \left( \mathbf{0}_{1\times 4}, 1 \right),$$

where $k = \sqrt{k^2}$. We have defined $\mathcal{P}^\perp$ to be a projector, namely $\mathcal{P}^{\perp\;L}_{\;M}\mathcal{P}^\perp_{LN} = \mathcal{P}^\perp_{MN}$, that annihilates the longitudinal vectors $\mathcal{P}_{iM}$. The latter are normalized to $\mathcal{P}_{iM}[k]\mathcal{P}^M_j[-k] = \delta_{ij}$, furthermore we have that $\mathcal{P}^\perp[-k] = \mathcal{P}^\perp[k]$. The completeness relation

$$\mathcal{P}^\perp_{MN}[k] + \sum_{i=\text{S},\text{V}} \mathcal{P}_{iM}[k]\mathcal{P}_{iN}[-k] = \eta_{MN}, \tag{2.13}$$

also holds. In terms of these objects, exploiting Lorentz invariance, we can parametrize the two-point function as

$$G_{MN}[k] = G_\perp(k^2)\mathcal{P}^\perp_{MN}[k] + \mathcal{P}_{iM}[k]\big[G_L(k^2)\big]_{ij}\mathcal{P}_{jN}[-k], \tag{2.14}$$

where the sum over $i, j = \text{V}, \text{S}$ is understood. In eq. (2.14), $G_\perp(k^2)$ is a scalar form-factor that parametrizes the transverse component of the propagator, while $G_L(k^2)$ is a $2 \times 2$ matrix of form-factors associated to the two "longitudinal" (in a 5D sense) modes V and S. Notice that $G_L$ is a symmetric matrix because of eq. (2.11).

The notation also allows us to express the Slavnov-Taylor identity in a compact form. We first write down the gauge-fixing (2.5) in momentum space

$$\mathcal{F}[k] = -\sum_{i=\text{V},\text{S}} f_i\,\mathcal{P}^M_i[-k]\Phi_M[k], \tag{2.15}$$

where $f$ is a 2-vector in the V-S space

$$f_i(k^2) = (k/\sqrt{\xi}, \sqrt{\xi}\,\tilde{m}). \tag{2.16}$$

The general Slavnov-Taylor identity in eq. (2.6) reads (with $O = 1$)

$$\Big(\sum_{i_1} f_{i_1}\mathcal{P}^{M_1}_{i_1}[-k_1]\Big) \cdots \Big(\sum_{i_n} f_{i_n}\mathcal{P}^{M_n}_{i_n}[-k_n]\Big)\langle\beta|\Phi_{M_1}[k_1]\cdots\Phi_{M_n}[k_n]|\alpha\rangle_c = 0, \tag{2.17}$$

---

[4]In order to avoid confusion we employ square brackets, e.g., "$[k]$", to indicate dependence on the full 4-momentum $k^\mu$, as opposed to its norm $k = \sqrt{k^2}$.

with an obvious notation for the connected matrix elements in Fourier space (divided by $(2\pi)^4$ times the Dirac delta of momentum conservation).

We now turn to amputated amplitudes. We denote them as $\mathcal{A}\{\Phi, \cdots, \Phi\}$ suppressing for shortness the labels $\alpha$ and $\beta$ for the external states. The connected amplitudes are equal to the amputated ones times the propagators on the external legs. In particular for one external leg we have

$$\langle \beta | \Phi_M[-k] | \alpha \rangle_c = i\, G_{MN}[-k] \mathcal{A}\{\Phi^N[k]\}. \tag{2.18}$$

Notice that with this definition the momentum "$k$" is incoming in the amputated amplitude. By applying eq. (2.17) for $n = 1$ we obtain

$$\Big(\sum_{i,j} f_i [G_L]_{ij} \mathcal{P}_{jM}[k]\Big) \mathcal{A}\{\Phi^M[k]\} = 0. \tag{2.19}$$

The equation states that the connected amplitude vanishes if contracted with a certain $k$-dependent 5D vector constructed from the gauge-fixing parameters (through $f$) and involving the longitudinal propagator matrix. For future applications it is convenient to rescale this vector to have minus one component along $\mathcal{P}_V$. Namely, we define

$$\mathcal{K}_M[k] \equiv -\mathcal{P}_{VM}[k] - \frac{\sum_i f_i [G_L]_{iS}}{\sum_i f_i [G_L]_{iV}} \mathcal{P}_{SM}[k] = \big(i\, k_\mu/k,\, \mathcal{K}_\pi\big), \tag{2.20}$$

where $\mathcal{K}_\pi(k^2)$ is the $\mathcal{P}_S$ component of $\mathcal{K}$. The $n{=}1$ Ward identity now reads

$$\mathcal{K}_M[k] \mathcal{A}\{\Phi^M[k]\} = 0 \quad \Leftrightarrow \quad i\, k_\mu \mathcal{A}\{W^\mu[k]\} = -k\, \mathcal{K}_\pi(k^2) \mathcal{A}\{\pi[k]\}, \tag{2.21}$$

while for generic $n$ we simply have

$$\mathcal{K}_{M_1}[k_1] \cdots \mathcal{K}_{M_n}[k_n] \mathcal{A}\{\Phi^{M_1}[k_1], \cdots, \Phi^{M_n}[k_n]\} = 0. \tag{2.22}$$

Eq. (2.22) is all we need. Unsurprisingly it is the same fundamental identity that underlies the proof of the Equivalence Theorem developed in Ref.s [24, 31–36] (see [37] for a review). It is called "generalized Ward identity" because it is the closest generalization we can get in a massive gauge theory of the QED Ward identity. In the analogy with QED, the 5D vector $\mathcal{K}_M$ could be interpreted (up to proportionality factors) as the generalization of the 4-momentum $k_\mu$ or, more usefully in this context, as the generalization of the polarization vector for the "scalar" component of the photon. We will see in Section 2.3 that the generalized Ward identity implies that the scalar polarization does not propagate, in close analogy with the standard QED result.

By looking at the $n{=}1$ case in eq. (2.21) we can easily understand how the Ward identities will allow us to get rid of the growing-with-energy polarization vectors for zero-helicity (longitudinal) vectors. The energy growth is associated to a component of the standard longitudinal polarization vectors that is proportional to $k_\mu$. Once contracted with the amputated amplitude the energy-growing term thus takes the form of the l.h.s. of eq. (2.21), which we can replace with r.h.s. that manifestly does not grow with energy. We will implement this mechanism systematically by subtracting the scalar polarization vector $\mathcal{K}_M$ from the standard longitudinal polarization. Notice that here we are taking for granted that the high-energy limit is taken at fixed $k = \sqrt{k^2}$, such that $\mathcal{K}_\pi$ is constant because it only depends on $k^2$ by Lorentz symmetry. This is the only limit worth discussing, and the only one in which the longitudinal polarization diverges. In concrete applications $k^2$ will be either $m^2$ for on-shell particles or $k^2 = Q^2 + m^2 \ll E^2$ for virtual ones.

**The Scalar Polarization Vector**

The definition of $\mathcal{K}$ in eq. (2.20) is rather cumbersome. Before moving forward to the study of the implications of the generalized Ward identity it is thus worth showing how it can be expressed in simpler terms, and eventually computed in perturbation theory.

The central object here is the inverse of the two-point function $G$ in eq. (2.14), which we parametrize for convenience as

$$
\begin{aligned}
G_{MN}^{-1} &= G_\perp^{-1}\mathcal{P}_{MN}^\perp[k] + \mathcal{P}_{iM}[k][G_L^{-1}]_{ij}\mathcal{P}_{jN}[-k] \\
&\equiv \Gamma_\perp \mathcal{P}_{MN}^\perp[k] + \mathcal{P}_{iM}[k][\widetilde{\Gamma} - F]_{ij}\mathcal{P}_{jN}[-k],
\end{aligned}
\tag{2.23}
$$

where $F$ is a $2 \times 2$ matrix constructed out of the gauge-fixing 2-vector $f$

$$
F_{ij} = f_i f_j = \begin{pmatrix} k^2/\xi & k\widetilde{m} \\ k\widetilde{m} & \widetilde{m}^2\xi \end{pmatrix}.
\tag{2.24}
$$

This seemingly obscure definition requires some explanation. The two-point function is the inverse of the quadratic effective action, namely it is the inverse of the Hessian of the effective action $\Gamma$ around the vacuum. When studying gauge theories it is often convenient to introduce a "tilded" effective action $\Gamma - \int dx\, \mathcal{L}_{\text{g.f.}}$ by subtracting and isolating the tree-level contribution from the gauge-fixing term. This is what we did above, namely we isolated in the matrix $F$ the contribution from $\mathcal{L}_{\text{g.f.}}$ in eq. (2.4), computed using eq. (2.15). The contribution from the "tilded" effective action appears instead in $\widetilde{\Gamma}$ for the longitudinal, and in $\Gamma_\perp$ for the transverse part of the propagator.

We now recall that the two-point function obeys the Slavnov–Taylor identity (2.7), that gives

$$
\vec{f}^{\,t} G_L \vec{f} = -1\,, \;\Rightarrow\; \vec{f}^{\,t} G_L F = -\vec{f}^{\,t}\,,
\tag{2.25}
$$

with an obvious vector notation for the gauge-fixing 2-vector $\vec{f} = (f_V \;\; f_S)^t$. Using that $G_L^{-1} = \widetilde{\Gamma} - F$, this is also equivalent to

$$
\vec{f}^{\,t} G_L \widetilde{\Gamma} = \vec{f}^{\,t} G_L(G_L^{-1} + F) = \vec{f}^{\,t} - \vec{f}^{\,t} = \vec{0}^{\,t}\,.
\tag{2.26}
$$

The above equation has two components, both of which can be used to simplify the expression for $\mathcal{K}_\pi$ in eq. (2.20). Writing them down explicitly we find

$$
\mathcal{K}_\pi = -\frac{\sum_i f_i(G_L)_{iS}}{\sum_i f_i(G_L)_{iV}} = \frac{\widetilde{\Gamma}_{VS}}{\widetilde{\Gamma}_{SS}} = \frac{\widetilde{\Gamma}_{VV}}{\widetilde{\Gamma}_{VS}}\,,
\tag{2.27}
$$

where we exploited the fact that $\widetilde{\Gamma}$ is symmetric. The second equality entails one relation among the three elements of $\widetilde{\Gamma}$, namely

$$
\widetilde{\Gamma}_{VS}^2 = \widetilde{\Gamma}_{SS}\widetilde{\Gamma}_{VV}\,.
\tag{2.28}
$$

This is equivalent to the "$B^2 = AC$" relation among the longitudinal form-factors derived in [27].

Having expressed $\mathcal{K}_\pi$ in terms of the inverse propagator we can easily set up its calculation in perturbation theory. Working for instance in bare perturbation theory one would write the inverse propagator in the form

$$
G_{MN}^{-1} = \Delta_{MN}^{-1} + \Pi_{MN}\,,
\tag{2.29}
$$

where $\Delta$ is the bare tree-level propagator (times $-i$) and $\Pi$ is the vacuum polarization amplitude

$$
\Delta^{-1} = \begin{pmatrix} (m_0^2 - k^2)\left(\eta_{\mu\nu} - \frac{k_\mu k_\nu}{k^2}\right) + \left(m_0^2 - \frac{k^2}{\xi}\right)\frac{k_\mu k_\nu}{k^2} & -ik_\mu(m_0 - \widetilde{m}) \\ ik_\nu(m_0 - \widetilde{m}) & k^2 - \widetilde{m}^2\xi \end{pmatrix},
$$
$$
\Pi = \begin{pmatrix} \Pi_{WW}^T(k^2)\left(\eta_{\mu\nu} - \frac{k_\mu k_\nu}{k^2}\right) + \Pi_{WW}^L(k^2)\frac{k_\mu k_\nu}{k^2} & -ik_\mu \Pi_{W\pi}(k^2) \\ ik_\nu \Pi_{W\pi}(k^2) & \Pi_{\pi\pi}(k^2) \end{pmatrix}.
\tag{2.30}
$$

By comparing with eq. (2.23) we obtain $G_\perp^{-1} = \Gamma_\perp = m_0^2 - k^2 + \Pi_{WW}^T$ and

$$\widetilde{\Gamma} = \begin{pmatrix} \widetilde{\Gamma}_{VV} & \widetilde{\Gamma}_{VS} \\ \widetilde{\Gamma}_{VS} & \widetilde{\Gamma}_{SS} \end{pmatrix} = \begin{pmatrix} m_0^2 + \Pi_{WW}^L & k[m_0 + \Pi_{W\pi}] \\ k[m_0 + \Pi_{W\pi}] & k^2 + \Pi_{\pi\pi} \end{pmatrix}. \tag{2.31}$$

We thus express $\mathcal{K}_\pi$ in terms of vacuum polarization amplitudes as

$$\mathcal{K}_\pi(k^2) = \sqrt{\frac{m^2 + \Pi_{WW}^L(k^2)}{k^2 + \Pi_{\pi\pi}(k^2)}}, \tag{2.32}$$

or in any of the other equivalent forms that can be obtained using eq. (2.28).

Eq. (2.32) could be now used to compute $\mathcal{K}_\pi$ in perturbation theory, giving operative meaning to eq. (2.22). The explicit result is not of interest in the toy Higgs–Kibble model. In the case of the SM we will compute $\mathcal{K}_\pi$ at one loop using eq. (2.32), or more precisely using its generalization derived in Section 3.2. A remarkable fact about eq. (2.32) is that it does not show explicit dependence on the gauge-fixing parameters $\widetilde{m}$ and $\xi$, in spite of the fact that it descends from the Slavnov–Taylor identities (2.6) where these parameters do appear. Yet, $\mathcal{K}_\pi$ implicitly depends on $\widetilde{m}$ and $\xi$ through the vacuum polarization amplitudes. However at tree-level $\Pi = 0$ and $\mathcal{K}_\pi$ is indeed gauge-independent and equal to $m/k$.

## 2.2 On-Shell Stable Vectors

The generalized Ward identities straightforwardly allow us to define a well-behaved longitudinal polarization vector that is fully equivalent to the ordinary one, in the sense that it gives the exact same results for all physical quantities. We consider here the case in which the massive vectors are stable external particles and show that the amplitudes computed using our modified polarization vectors as wave-functions for the external legs are identical to the standard ones.

We have seen that the Ward identities are conveniently derived and stated with a notation where the $W_\mu$ and the $\pi$ fields are collected in a single 5-components field $\Phi_M$, $M = (\mu, \pi)$, as in eq. (2.9). We can incorporate this notation in Feynman diagrams by a graphical representation of $\Phi_M$ lines on the external legs. Since $\Phi_M$ contains both vector and scalar lines it is natural to represent it with a double line as in Figure 1. The double line carries a 5D index $M$, to be contracted with polarization vectors $\mathcal{E}_M$ that also live in the 5D space. One might have decided to express in terms of $\Phi_M$ all the Feynman rules of the theory, including the propagator (whose 5D form was written down in the previous section) and the vertices, in which case only double lines would appear in any internal or external line of the diagram and all calculations could in principle be carried on directly in the 5D notation. However the standard Feynman rules are expressed separately for gauge and Goldstone legs, therefore in order to apply them we must break down the double lines on the external legs as shown in the figure. This entails that we need to compute 2 different amputated amplitudes for each external particle, one with a gauge and one with a Goldstone external leg, and sum them up with coefficients given by the polarization vector. The total number of amplitudes to be evaluated thus increases by a factor of two for each external longitudinal vector boson relative to the one that is needed in the standard formalism. Explicit applications will be shown later in Section 4.4. The usefulness of the double line notation has been first noticed in Ref. [38] for tree-level applications of the Equivalence Theorem and emphasized in Ref. [25].

The standard formalism is recovered in this notation by 5D polarization vectors with vanishing $M = \pi$ component. For incoming and outgoing particles with 4-momentum $k^\mu$ we have, respectively

$$(\mathcal{E}_{\text{st.}}^h)_M[k] \equiv (\varepsilon_\mu^h[k], 0),$$

$$(\overline{\mathcal{E}}_{\text{st.}}^h)_M[k] \equiv (\overline{\varepsilon}_\mu^h[k], 0), \tag{2.33}$$



Figure 1: Feynman rules for incoming (top) and outgoing (bottom) external longitudinal states.

where $h = \pm, 0$ is the helicity and $\varepsilon_\mu^h$ denote the standard 4D polarization vectors, reported in Appendix A. The transverse ($h = \pm$) polarizations do not display an anomalous energy behavior. The longitudinal ($h = 0$) one is instead

$$\varepsilon_\mu^0[k] = \overline{\varepsilon}_\mu^0[k] = \left\{ \frac{|\vec{k}|}{k}, -\frac{k_0}{k}\frac{\vec{k}}{|\vec{k}|} \right\} \overset{k_0/k \to \infty}{\sim} \frac{\sqrt{k_0^2}}{k_0} \left\{ \frac{k_0}{k}, -\frac{1}{k}\vec{k} \right\} = \frac{\sqrt{k_0^2}}{k_0}\frac{k_\mu}{k}, \qquad (2.34)$$

with $|\vec{k}| = \sqrt{k_0^2 - k^2}$. Note that we keep $k$ generic in preparation for the next section where we will consider off-shell vectors, possibly with complex momentum. Clearly $k = \sqrt{k^2} = m$ in the case at hand, and $k_0 > 0$ so that $\sqrt{k_0^2}/k_0 = 1$. In high-energy reactions where the particle energy $k_0 \sim E$ is much larger than $k = m$, $\varepsilon_\mu^0$ diverges and approaches $k_\mu/k$. It is therefore convenient to define

$$e_\mu^0[k] \equiv \varepsilon_\mu^0[k] - \frac{k_\mu}{k} \overset{\text{if } \Re(k_0)>0}{=} -\frac{k}{k_0 + |\vec{k}|}\left\{1, \frac{\vec{k}}{|\vec{k}|}\right\}. \qquad (2.35)$$

Also in this definition we consider generic complex momentum. We thus specified that $k_0$ must have positive real part (so that $\sqrt{k_0^2} = k_0$) for $e_\mu^0[k]$ to have the simple form on the right of the equation and a non-singular high energy limit. If it is so, $e_\mu^0[k] \sim k/k_0$, i.e., $m/E$ for on-shell momentum.

Thanks to the Ward identities in eq. (2.22) we are allowed to shift $\mathcal{E}_{\text{st.}}^0[k]$ and $\overline{\mathcal{E}}_{\text{st.}}^0[k]$ by any vector proportional to $\mathcal{K}[k]$. The shift will indeed cancel out when the polarization vector is contracted with the amputated amplitude. More precisely, since the external legs of amputated amplitudes are labeled by *incoming* 4-momenta, the polarization vectors for outgoing states should be shifted by $\mathcal{K}[-k]$ because they will have to be contracted to an amplitude with external leg $\Phi^M[-k]$. We see in eq. (2.20) that $\mathcal{K}_\mu[k] = -\mathcal{K}_\mu[-k] = i\,k_\mu/k$. We should thus manage to get rid of the anomalous energy growth by defining

$$\mathcal{E}_M^0[k] \equiv (\mathcal{E}_{\text{st.}}^0)_M[k] + i\,\mathcal{K}_M[+k] = \left( e_\mu^0[k], +i\,\mathcal{K}_\pi(k^2) \right),$$
$$\overline{\mathcal{E}}_M^0[k] \equiv (\overline{\mathcal{E}}_{\text{st.}}^0)_M[k] - i\,\mathcal{K}_M[-k] = \left( e_\mu^0[k], -i\,\mathcal{K}_\pi(k^2) \right), \qquad (2.36)$$

to be our new polarization vectors.

It is immediate to verify that indeed, when contracted with the appropriate amputated amplitudes and evaluated on the mass-shell, eq. (2.36) produces the exact same physical scattering matrix element as the standard polarization vectors. Consider a generic scattering process with "$n$" external longitudinal vectors and an arbitrary set $\alpha$ and $\beta$ of non-longitudinal incoming and outgoing particles. The matrix element computed with our on-shell polarizations is

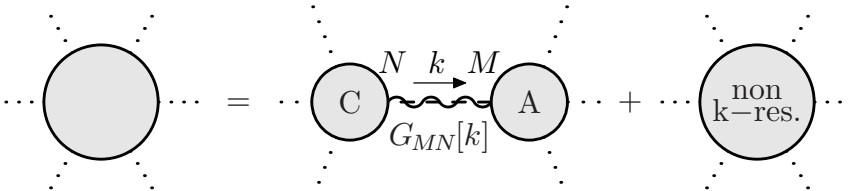

Figure 2: The decomposition of a generic scattering amplitude into resonant and non-resonant diagrams, for a given intermediate vector boson line with momentum $k$. Remember that (as, e.g., for Dirac fields) the momentum flows from the second towards the first index of the propagators.

equal to that evaluated with the standard ones

$$
\begin{aligned}
i\mathcal{M} &= Z_W^{n/2} \big((\mathcal{E}_{\text{st.}}^0)_{M_1}[k_1] \pm i\mathcal{K}_{M_1}[\pm k_1]\big) \cdots \big((\mathcal{E}_{\text{st.}}^0)_{M_n}[k_n] \pm i\mathcal{K}_{M_n}[\pm k_n]\big) \mathcal{A}\{\Phi^{M_1}[\pm k_1], \cdots, \Phi^{M_n}[\pm k_n]\} \\
&= Z_W^{n/2} (\mathcal{E}_{\text{st.}}^0)_{M_1}[k_1] \cdots (\mathcal{E}_{\text{st.}}^0)_{M_n}[k_n] \mathcal{A}\{\Phi^{M_1}[\pm k_1], \cdots, \Phi^{M_n}[\pm k_n]\}.
\end{aligned}
\tag{2.37}
$$

In the equation, $Z_W$ denotes the wave-function renormalization factor and the $+$ or $-$ sign is for incoming or outgoing particles as previously explained. The equality comes from expanding the product and noticing that in each monomial the amputated amplitude is contracted with $r = 0, \cdots, n$ powers of $\mathcal{K}$, while the remaining $n - r$ external legs are contracted with the standard polarization vectors. The latter contractions produce additional particles in the external states $\alpha$ and $\beta$, but this is immaterial because the Ward identity in equation eq. (2.22) holds for arbitrary (possibly longitudinal) physical states. All terms thus cancel by the Ward identity for "$r$" legs, apart obviously from the first term with $r = 0$ that gives us back the standard matrix element.

Our new polarization vectors (2.36) do not grow with energy. Their Goldstone component $\mathcal{E}_\pi$ stays constant while the gauge component $\mathcal{E}_\mu = \mathrm{e}_\mu^0$ scales as $m/k_0 \sim m/E$ and vanishes in the high-energy limit. This produces a suppression factor of the gauge contribution to the amplitude relative to the Goldstone one, which is nothing but the technical statement of the Goldstone Equivalence Theorem. Notice however that one should not superficially interpret this result as the dominance of Goldstone diagrams over the gauge ones. Whether or not the Goldstones dominate depends on the high-energy behavior of the Goldstone amplitudes relative to the gauge ones in the specific process at hand. Namely, the $m/E$ suppression of $\mathrm{e}_\mu^0$ is only one of the elements of the power-counting rule. Other $m/E$ factors might well emerge from the vertices that appear in the Goldstone amplitudes and compensate the wave-function suppression factor of the gauge diagrams. Explicit examples are discussed in Section 4.4.1.

Finally, we briefly comment on the relation with Ref. [25], where the modified polarization vectors (2.36) were defined and employed to make power-counting manifest like we do here. Our result is identical, but more general in that we proved that the modified polarization vectors are equivalent to the standard ones for an arbitrary choice of the gauge-fixing parameters $\xi$ and $\tilde{m}$. In the approach of Ref. [25] instead the gauge-fixing parameters are set by requiring two conditions (mass-degeneracy and dipole cancellation) on the gauge-fixed theory. This in turn was needed for the extended Fock space of the theory, including unphysical states, to have a structure compatible with the sought redefinition of the longitudinal state.

## 2.3 Unstable or Off-Shell Vectors

Our results up to now are of limited practical interest because the SM massive vectors are not stable asymptotic states. Hence their scattering matrix elements, and in turn their polarization vectors, cannot be defined through the LSZ reduction formula as we implicitly did in eq. (2.37).

Furthermore in order to study factorization problems we will have to deal with diagrams where the external vector bosons are off-shell. Namely in that kind of problems the virtual vector boson $k^2$ is much smaller than the hard scale $E^2 \gg m^2$, but in general not close to $m^2$.

In order to deal with unstable or with off-shell vectors one needs to start from the complete scattering amplitude involving only true asymptotic particles on the external legs. Among the diagrams that contribute to the complete amplitude one isolates the "resonant" ones containing a vector boson propagator that connects two otherwise disconnected components of the diagram as in Figure 2. The momentum "$k$" flowing into the propagator should be interpreted as the momentum of a virtual boson, which is created and annihilated in the left ("creation", C) and right ("annihilation", A) components of the resonant diagram. Therefore we orient $k$ to have positive energy component, or positive real part of the energy, for complex kinematics

$$\Re(k_0) > 0. \tag{2.38}$$

The creation subprocess corresponds to the production of the virtual vector in association with other particles, either originating from the two initial particles or from a single one in the case of an initial state splitting. The annihilation subprocess represents either the decay of the vector boson or the scattering of the vector boson with a particle in the initial state. In all cases the virtual vector boson can be uniquely associated to a partition of the external states into the two sets involved in the production and in the decay. Notice that diagrams with scalar–scalar and with mixed vector–scalar propagators are included in the resonant component of the scattering amplitude, therefore the propagator is denoted with a double line in the figure. The line represents the complete all-orders two-point function $G_{MN}$ as in eq. (2.14).

The on-shell matrix elements for unstable particles are defined in a perfectly gauge-invariant fashion in terms of the residue of the complete amplitude at the complex pole $k^2 = m^2 \in \mathbb{C}$. Clearly this requires studying the amplitude with complex kinematics, a fact which however does not raise any particular issue because the generalized Ward identities hold in the entire complex plane by the analyticity properties of the Feynman amplitudes. The pole originates exclusively from the resonant diagrams, and the residue of the complete amplitude equals the one of the propagator multiplied by the creation and annihilation amplitudes evaluated at complex $k^2 = m^2$. The Feynman rules for on-shell unstable particles creation and annihilation thus emerge from the decomposition of the propagator at the complex mass pole, as we will readily see. Notice that defining gauge-invariant on-shell matrix elements for unstable particles is concretely useful because it allows to simplify the calculation of the complete scattering amplitude (including the decay) with real kinematics by expanding in the number of resonant poles as in Ref.s [39–42]. The reason for focusing on the resonant diagrams in the off-shell case stems from the fact that in a formalism like ours, where power-counting is manifest, the resonant diagrams are enhanced by $E^2/k^2$ relative to the non-resonant ones and thus dominate in the factorization limit $k^2/E^2 \ll 1$. The propagator decomposition we are about to work out is thus essential for the proof of collinear factorization in Section 5.

In order to extend our discussion to off-shell and to unstable vectors we should further study the 2-point function which, using eq. (2.23), reads

$$G_{MN}[k] = \Gamma_\perp^{-1} \mathcal{P}_{MN}^\perp[k] + \mathcal{P}_{iM}[k] \big[ G_L \big]_{ij} \mathcal{P}_{j,N}[-k], \tag{2.39}$$

where $G_L = (\widetilde{\Gamma} - F)^{-1}$. We first notice that the familiar completeness relation for the 4D polarization vectors allows us to decompose $\mathcal{P}_{MN}^\perp[k]$, for arbitrary (complex, in general) $k^\mu$ in terms of the 5D standard polarization vectors defined in eq. (2.33). We can thus write

$$G_{MN}[k] = -\Gamma_\perp^{-1} \sum_{h=\pm} \mathcal{E}_M^h[k] \overline{\mathcal{E}}_N^h[k] - \Gamma_\perp^{-1} (\mathcal{E}_{\text{st.}}^0)_M[k] (\overline{\mathcal{E}}_{\text{st.}}^0)_N[k] + \mathcal{P}_{iM}[k] \big[ G_L \big]_{ij} \mathcal{P}_{j,N}[-k], \tag{2.40}$$

where we suppressed the "st." subscript from the transverse polarization vectors because they are well-behaved with energy and thus they do not need to be redefined.

From the propagator decomposition in eq. (2.40) one could immediately recover the standard definition of the polarization vectors for unstable particles. Indeed in this case one is exclusively interested in the first two terms of the decomposition because they have a pole at the physical (complex) mass $m$. Actually the mass is defined precisely as the point in the complex plane where $\Gamma_\perp$ vanishes, i.e. by the relation

$$\Gamma_\perp(m^2) = m_0^2 - m^2 + \Pi_{WW}^T(m^2) = 0 \,. \tag{2.41}$$

Instead the longitudinal part of the propagator, namely the third term in eq. (2.40), will not have in general a pole at the same location and it can be ignored in the calculation of the residue. We thus express the residue as a sum over helicities, and interpret each term as the product of the polarization vectors to be contracted with the amplitudes at the two endpoints of the propagator line. The one on the left leg, with index "$N$", corresponds to the creation of a particle, the one on the right, with index "$M$", to annihilation. This leads to the definition of on-shell amplitudes for unstable vectors, as extensively discussed in the literature (see e.g. Ref. [43]).

The standard longitudinal polarization vectors display the usual anomalous energy behavior (2.34). However we can trade them for the well-behaved objects defined in eq. (2.36) and write the $h = 0$ term of the decomposition (2.40) as

$$(\mathcal{E}_{\text{st.}}^0)_M[k](\overline{\mathcal{E}}_{\text{st.}}^0)_N[k] = \left(\mathcal{E}^0[k]\right)_M\left(\overline{\mathcal{E}}^0[k]\right)_N \tag{2.42}$$
$$+\left(i\,\mathcal{E}^0[k] + \frac{1}{2}\mathcal{K}[k]\right)_M \mathcal{K}_N[-k] + \mathcal{K}_M[k]\left(-i\,\overline{\mathcal{E}}^0[k] + \frac{1}{2}\mathcal{K}[-k]\right)_N \,.$$

Importantly, the signs in front of the $i\,\mathcal{K}$ terms in eq. (2.36) ensure that no high-energy growth is present in the modified polarization vectors, as in the on-shell case, because we chose $\Re(k_0) > 0$ such that eq. (2.35) applies. Furthermore, having employed $\mathcal{K}[+k]$ in the definition of $\mathcal{E}_M$ and $\mathcal{K}[-k]$ in that of $\overline{\mathcal{E}}_N$ ensures that the terms in the second line of eq. (2.42) are proportional to $\mathcal{K}[+k]_M$ and $\mathcal{K}[-k]_N$. Because the index $M$ is contracted with the annihilation sub-amplitude (see Figure 2) where the $k$ momentum is incoming while $N$ is contracted with the creation one where $k$ is outgoing, these terms do not contribute to the complete amplitude by the Ward identity in eq. (2.22). We will prove that this is indeed the case at the end of this section.

Taking for granted that the second line can be dropped, eq. (2.42) is all we need in order to deal with unstable on-shell vectors. It allows us to express the residue at the propagator pole, and in turn to define the gauge invariant on-shell matrix elements, in terms of two transverse and one longitudinal polarization vectors that are well-behaved with energy. In particular the longitudinal polarization vectors for an unstable on-shell particle are those in eq. (2.36) for $k = m$. They are identical to those for a stable particle up to the fact that the mass $m$ now is complex. The situation is more complicated for off-shell vectors because also the longitudinal (in the 5D sense) component of eq. (2.40) now matters. Indeed in the off-shell case we are interested in propagators where $k^2 - m^2$ does not vanish exactly, so not only the residue of the pole is relevant. Rather we are interested in configurations where $k = \sqrt{k^2}$ is either of order or much larger than $m$, but much smaller than the virtual vector energy and momentum $k^\mu \sim E$. The longitudinal component of the propagator, in the current form, does not possess a smooth high energy limit because it contains up to two powers of $k^\mu$ from the $\mathcal{P}_V$ vector (see eq. (2.12)). One extra step is thus needed in order to express the propagator in a form that is suited to deal with factorization problems.

First, we get rid of any occurrence of $\mathcal{P}_V$ in the longitudinal propagator by rewriting it as

$$\mathcal{P}_{V,M}[k] = -\mathcal{K}_M[k] + \frac{\widetilde{\Gamma}_{VS}}{\widetilde{\Gamma}_{SS}}\mathcal{P}_{S,M}[k], \tag{2.43}$$

in light of eq.s (2.20) and (2.27). After a straightforward calculation one can check that the result can be expressed as

$$\begin{aligned}\mathcal{P}_{i,M}[k]\big[G_L\big]_{ij}\mathcal{P}_{j,N}[-k] &= \widetilde{\Gamma}_{SS}^{-2}\mathcal{P}_{S,M}[k]\big[\widetilde{\Gamma}\cdot G_L\cdot\widetilde{\Gamma}\big]_{SS}\mathcal{P}_{S,N}[-k] \\ &\quad -\mathcal{V}[k]_M\mathcal{K}[-k]_N - \mathcal{K}[k]_M\mathcal{V}[-k]_N\,,\end{aligned} \tag{2.44}$$

in terms of some vector $\mathcal{V}$. The explicit form of $\mathcal{V}$ need not be reported because the second line of the previous equation will cancel out by the same considerations we made below eq. (2.42). Finally, we notice that eq. (2.26) implies that $FG_L\widetilde{\Gamma} = 0$, so that

$$\widetilde{\Gamma}\cdot G_L\cdot\widetilde{\Gamma} = \big[\widetilde{\Gamma} - F\big]\cdot G_L\cdot\widetilde{\Gamma} = \widetilde{\Gamma}\,, \tag{2.45}$$

as $G_L = [\widetilde{\Gamma} - F]^{-1}$. Summing up eq. (2.44) and eq. (2.42), we rewrite the complete propagator as

$$G_{MN}[k] = G_{MN}^{\text{eq}}[k] - \mathcal{K}_M[k]\mathcal{V}_N[-k] - \mathcal{V}_M[k]\mathcal{K}_N[-k]\,, \tag{2.46}$$

where we reabsorbed into "$\mathcal{V}$" the terms on the second line of eq. (2.42) and the "equivalent" propagator $G^{\text{eq}}$ is defined to be

$$G_{MN}^{\text{eq}}[k] \equiv -\Gamma_{\perp}^{-1}\sum_{h=\pm,0}\mathcal{E}_M^h[k]\overline{\mathcal{E}}_N^h[k] + \widetilde{\Gamma}_{SS}^{-1}\mathcal{P}_{S,M}[k]\mathcal{P}_{S,N}[-k]\,. \tag{2.47}$$

We will prove below that this is "equivalent" to $G$, namely that it can be used in place of $G$ in resonant propagators. Taken this for granted, $G^{\text{eq}}$ possesses all the required properties. Namely it contains no energy growth neither in the $h = \pm, 0$ nor in the scalar part, so that it will allow us to take the $k^2/E^2 \ll 1$ limit smoothly when studying factorization in Section 5. On the physical mass complex pole it straightforwardly leads us to the definition of the on-shell polarization vectors for unstable particles as previously explained.

We now prove that $G^{\text{eq}}$ defined as in eq. (2.47) is indeed "equivalent" to the complete propagator $G$ thanks to the Ward identity (2.22). More precisely, the statement is that in a generic scattering amplitude with stable asymptotic particles on the external legs it is possible to replace resonant propagators $G[k]$ with $G^{\text{eq}}[k]$ in all the Feynman diagrams without affecting the result. In turn, this will imply that the same is true if the external particles are unstable but on their complex mass-shell. The resonant propagators are those that connect two otherwise disconnected components of the diagram, propagators involved in closed loops are thus excluded and cannot be replaced with $G^{\text{eq}}$. Also notice that the equivalence holds only provided any occurrence of $G[k]$, for a given intermediate boson momentum $k$, is replaced in all diagrams. Substituting $G[k]$ with $G^{\text{eq}}[k]$ in some diagram and not in others would be inconsistent. On the other hand, we are not obliged to replace all propagators at once. Namely one can choose an arbitrary set of intermediate bosons momenta $k_i$, $i = 1, \ldots, m$, each corresponding to a given creation/annihilation subprocess and in turn to the signed sum of a given subset of external particles momenta, and perform the replacement $G[k_i] \to G^{\text{eq}}[k_i]$ only for the corresponding propagators. In particular high-virtuality internal lines, for which the decomposition (2.47) of $G^{\text{eq}}$ is of no help, need not be replaced.

It is trivial to establish the equivalence when only one propagator has to be replaced. It suffices to combine Figure 2 with eq. (2.46) and to notice that the shift induced by the replacement produces two terms with $\mathcal{K}[-k]$ and $\mathcal{K}[+k]$ contracted with the creation and annihilation amplitudes, respectively. The latter amplitudes have physical external states so that

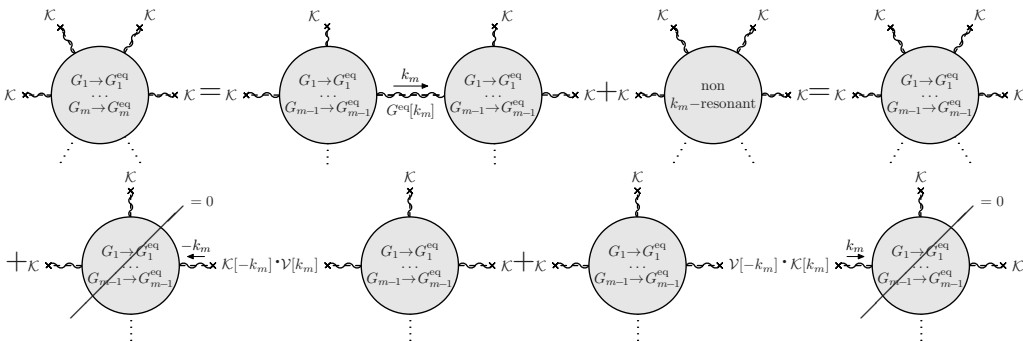

Figure 3: The relation by which the equivalence of the $G^{\mathrm{eq}}$ propagator is established.

they vanish when contracted with $\mathcal{K}$ by using eq. (2.22) for $n = 1$. In order to deal with the general case we have to employ the diagrammatic relation in Figure 3. In the figure we consider a generic scattering amplitude with a generic number $n \geq 0$ of additional gauge/scalar external legs, contracted with the 5D scalar polarization vector $\mathcal{K}$. Each scalar polarization is of course evaluated on the (outgoing) momentum of the corresponding external leg. The replacement operation $G[k_i] \to G^{\mathrm{eq}}[k_i]$ has been performed for a generic number $m \geq 1$ of propagators and it is indicated as $G_i \to G_i^{\mathrm{eq}}$ for shortness in the figure. The replacement of the last propagator affects only the component of the amplitude that is resonant with respect to the $k_m$ momentum, producing two terms in which $\mathcal{K}[-k_m]$ is contracted with the creation sub-amplitude and $\mathcal{K}[+k_m]$ is contracted with the annihilation one. By recombining the resonant and non-resonant components we reconstruct the original amplitude with one less propagator replaced, plus additional terms proportional to the sub-amplitudes with at least one external leg contracted with $\mathcal{K}$ as in the second line of the figure. The sub-amplitudes do not contain $G[k_m]$ propagators by definition, therefore also in the latter ones only the propagators from $G[k_1]$ to $G[k_{m-1}]$ have been replaced. Notice that we cannot trivially conclude that the latter sub-amplitudes vanish because the Ward identity (2.22) only holds a priori for "normal" propagators in the internal lines. However they do vanish as indicated in the figure because we can further apply the diagrammatic relation to each of them, focusing this time on the $G[k_{m-1}]$ propagator, obtaining the same amplitude with $m-2$ replaced propagators plus additional amplitudes with one more external $\mathcal{K}$ and, once again, only $m-2$ replaced propagators. By repeating the operation one can eliminate all the replacements and eventually apply the Ward identity (2.22). The terms on the last line of the figure thus vanish and we conclude that the original amplitude is equal to the one with one less propagator replaced. By applying this equality to the physical scattering amplitude with no external $\mathcal{K}$ insertions (i.e., $n = 0$) and "$m$" propagators replaced one finally concludes that it is identical to the one with "$m-1$" replacements and eventually to the one with no replacement at all, as it had to be proven.

# 3 General Gauge Theory

We now turn our attention to a generic theory with arbitrary gauge group ($\mathcal{G}$) and field content. We denote the gauge fields as $V_\mu^a$, $a = 1, \dots, N_{\mathrm{V}} = \dim(\mathcal{G})$, while $\phi_{\tilde{a}}$ are the $N_{\mathrm{S}}$ scalars, which we take real without loss of generality. The presence of fermionic fields is irrelevant for the discussion that follows. The gauge-invariant Lagrangian prior to gauge-fixing is also arbitrary and not necessarily renormalizable. An arbitrary symmetry breaking pattern $\mathcal{G} \to \mathcal{H}$, giving mass to some of the vectors and leaving some of the scalars as physical Higgs bosons, is allowed. We quantize the theory with the standard Faddeev–Popov method as described in

Section 2 for the Higgs–Kibble model obtaining a Lagrangian as in eq. (2.3). The $N_V$ gauge-fixing functionals are chosen to be linear in the fields, i.e.

$$\mathcal{F}_a = \sum_b \Xi_{ab} \partial_\mu V_b^\mu - \sum_{\tilde{b}} \widetilde{\mu}_{a\tilde{b}} (\phi_{\tilde{b}} - \langle \phi_{\tilde{b}} \rangle), \tag{3.1}$$

where $\langle \phi_{\tilde{b}} \rangle$ denotes the VEV of the scalar fields. The derivation of the Slavnov–Taylor identities discussed in Section 2.1 is completely general, therefore eq.s (2.6) and (2.7) straightforwardly apply to the present case as well. Notice in particular that the Slavnov–Taylor identity obtained by setting $n = 1$ in eq. (2.6) is the reason why we subtracted the scalar fields VEV in the definition (3.1) of the gauge-fixing functional. This identity reads $\langle \mathcal{F}_a(x) \rangle = 0$, which is consistent with $\langle A_\mu \rangle = 0$ only provided the scalars appearing in the gauge-fixing functional have vanishing VEV. If we had not subtracted the scalars VEV, the theory would have developed a non-Poincaré invariant VEV in order to satisfy the identity, and manifest Poincaré symmetry would have been lost.

The general discussion of Goldstone Equivalence is conceptually identical to the one we presented in the previous section for the Higgs–Kibble model. It merely requires a slightly more heavy notation, which we establish in Section 3.1. We then move (in Section 3.2) to the derivation of the generalized Ward identities and finally present (in Section 3.3) our well-behaved longitudinal polarization vectors and propagator.

## 3.1 Notation

We begin by collecting all the bosonic fields of the theory in a single vector $\Phi_M$ with $4N_V + N_S$ real components. The capital index $M$ ranges both over the $4N_V$ Lorentz-times-gauge pairs (i.e., $M = \{\mu, a\}$) that label the vector fields and over the $N_S$ indices of the scalars (i.e., $M = \tilde{a}$). Actually it is more convenient to shift the scalars by their VEVs and to define

$$\Phi_M = \begin{pmatrix} V_{\mu a} & \text{for } M = \{\mu, a\} \\ \phi_{\tilde{a}} - \langle \phi_{\tilde{a}} \rangle & \text{for } M = \tilde{a} \end{pmatrix}. \tag{3.2}$$

The "$M$" indices are raised by a metric $\eta^{MN}$ acting like the 4D Lorentz metric on the vector components $M = \{\mu, a\}$ and as the identity on the scalar ones $M = \tilde{a}$. The need of collecting all the bosonic fields in a single object stems from the fact that in general there is no way to associate a particular scalar field combination to each $V_\mu^a$ vector. Namely there is no useful notion of the "Goldstone field" associated to the vector. Concretely, the point is that all scalars in the theory mix a priori with all vectors, hence all bosonic fields have to be treated together in order to deal with the two-point function and in turn to derive the Ward identities. In the Higgs–Kibble model it was possible to identify the Goldstones, and thus to work with a small (5D) $\Phi_M$ multiplet, merely because of the presence of an exact custodial symmetry. The only exact symmetry that is necessarily present in the general case is the one associated with the unbroken gauge group $\mathcal{H}$, which however in general is insufficient to identify the Goldstones uniquely. We thus work with the large $\Phi_M$ multiplet and ignore the possible presence of exact symmetries in the theory. In the presence of symmetries it is possible to collect the fields in separate subspaces that do not mix with each other, as we will do in the SM by imposing charge and CP conservation. All the results that follow apply to each sector separately, provided we identify $\Phi_M$ with the short multiplet of each subspace.

We introduce, as in Section 2.1, a number of objects $\mathcal{P}_{aM}[k]$, $\mathcal{P}_{\tilde{a}M}[k]$ and $\mathcal{P}_{MN}^{\perp; ab}[k]$, depending on the Lorentz 4-momentum $k_\mu$. The former are vectors in the $(4N_V + N_S)$-dimensional

space

$$\mathcal{P}_{aM}[k] = \begin{pmatrix} -i\,\dfrac{k_\mu}{k}\delta_{aa'} & \text{for } M = \{\mu, a'\} \\ 0 & \text{for } M = \tilde{a}' \end{pmatrix}, \quad \mathcal{P}_{\tilde{a}M}[k] = \begin{pmatrix} 0 & \text{for } M = \{\mu, a'\} \\ \delta_{\tilde{a}\tilde{a}'} & \text{for } M = \tilde{a}' \end{pmatrix},$$

(3.3)

that correspond to the $N_V + N_S$ fields in the theory that are orthogonal to the transverse gauge fields. We refer to them collectively as "longitudinal", notice however that they consist both of the "longitudinal vectors" in the 4D sense and of the scalar components of $\Phi_M$. The longitudinal vectors are further collected into a single object $\mathcal{P}_{\dot{I}M}[k]$ by introducing a dotted capital index "$\dot{I}$" ranging over the $N_V$ vector indices $a$ and over the $N_S$ scalar ones $\tilde{a}$. The capital dotted index "$\dot{I}$" ranges over the $N_V + N_S$ longitudinal fields, and it should not be confused with the undotted index "$M$" that labels all the $4N_V + N_S$ bosonic fields. We also define a set of transverse "projectors"

$$\mathcal{P}_{MN}^{\perp;\,ab}[k] = \begin{pmatrix} (\eta_{\mu\nu} - \dfrac{k_\mu k_\nu}{k^2})\delta_{aa'}\delta_{bb'} & \text{for } M = \{\mu, a'\} \text{ and } N = \{\nu, b'\} \\ 0 & \text{otherwise} \end{pmatrix}.$$

(3.4)

Notice that the $\mathcal{P}^{\perp}$'s are not projectors in the strict mathematical sense, still they obey the relation

$$\mathcal{P}_M^{\perp;\,ab\,L}\mathcal{P}_{LN}^{\perp;\,cd} = \delta^{bc}\mathcal{P}_{MN}^{\perp;\,ad}.$$

(3.5)

Other useful properties of the $\mathcal{P}$'s include

$$\mathcal{P}_M^{\perp;\,ab\,L}\mathcal{P}_{\dot{I}L} = 0, \quad \mathcal{P}_{\dot{I}M}[k]\mathcal{P}_{\dot{j}}^M[-k] = \delta_{\dot{I}\dot{j}},$$

(3.6)

and the completeness relation

$$\sum_a \mathcal{P}_{MN}^{\perp;\,aa}[k] + \sum_{\dot{I}} \mathcal{P}_{\dot{I}M}[k]\mathcal{P}_{\dot{I}N}[-k] = \eta_{MN}.$$

(3.7)

Finally, we also have

$$\mathcal{P}_{MN}^{\perp;\,ab}[-k] = \mathcal{P}_{MN}^{\perp;\,ab}[k], \quad \mathcal{P}_{NM}^{\perp;\,ab}[k] = \mathcal{P}_{MN}^{\perp;\,ba}[k].$$

(3.8)

The above relations are implicitly used below, in particular in order to invert the propagator and to impose Bose symmetry.

We now use our notation to express in compact form the basic objects we will need in the next section, starting from the gauge-fixing. The Fourier space expression of eq. (3.1) is, similarly to eq. (2.15) for the Higgs–Kibble model

$$\mathcal{F}_a[k] = -\sum_{\dot{I}} f_{a\dot{I}}\,\mathcal{P}_{\dot{I}}{}^M[-k]\Phi_M[k],$$

(3.9)

in terms of an $N_V \times (N_V + N_S)$ matrix $f$ with components

$$f_{a\dot{I}} = (k\,\Xi_{aa'}, \widetilde{\mu}_{a\tilde{a}'}),$$

(3.10)

for $\dot{I} = a'$ and $\dot{I} = \tilde{a}'$, respectively. The parameters $\Xi$ and $\widetilde{\mu}$ are generic real tensors, subject however to a consistency condition associated with the fact that the gauge-fixing functional $\mathcal{F}_a$'s should be sufficient to fix the gauge completely. Namely there should not exist a family of local gauge transformations that leave all the $\mathcal{F}_a$'s invariant. It is relatively easy to show that this implies that the matrix $f$ has maximal rank $N_V$ up to isolated singularities in the $k^2$ space. The same condition can also be obtained by imposing that the ghost kinetic term is

non-singular up to isolated poles in $k^2$, that correspond to the ghost tree-level masses. Taking the matrix $\Xi$ to be invertible is one way to fulfill the condition. For future convenience we introduce, as in eq. (2.24), the symmetric rank-$N_V$ matrix

$$F_{\dot{I}\dot{J}}(k^2) = \sum_a f_{a\dot{I}}(k^2)f_{a\dot{J}}(k^2) = (f^t f)_{\dot{I}\dot{J}}. \qquad (3.11)$$

We now turn to the propagator, defined as in eq. (2.10). Similarly to what we did in eq. (2.23) for the Higgs–Kibble model, we write its inverse as

$$G_{MN}^{-1}[k] = [\Gamma_\perp(k^2)]_{ab}\mathcal{P}_{MN}^{\perp;\,ab}[k] + \mathcal{P}_{\dot{I}M}[k][\widetilde{\Gamma}(k^2)-F(k^2)]_{\dot{I}\dot{J}}\mathcal{P}_{\dot{J}N}[-k], \qquad (3.12)$$

where the sums over $a$, $b$, $\dot{I}$ and $\dot{J}$ are understood. The main difference compared to eq. (2.23) is that the transverse component $\Gamma_\perp^{-1}$ is now a $N_V \times N_V$ matrix rather than a single form-factor and the longitudinal component $\widetilde{\Gamma} - F$, which was $2 \times 2$, is now $(N_V + N_S) \times (N_V + N_S)$. Bose symmetry implies $G_{MN}[-k] = G_{NM}[k]$, hence $\Gamma_\perp$ and $\widetilde{\Gamma}$ are symmetric matrices. By inverting, we find

$$G_{MN}[k] = [\Gamma_\perp^{-1}]_{ab}\mathcal{P}_{MN}^{\perp;\,ab}[k] + \mathcal{P}_{\dot{I}M}[k][(\widetilde{\Gamma}-F)^{-1}]_{\dot{I}\dot{J}}\mathcal{P}_{\dot{J}N}[-k]. \qquad (3.13)$$

We will often denote $G_L \equiv (\widetilde{\Gamma} - F)^{-1}$ in what follows.

## 3.2 Generalized Ward Identities

The derivation follows closely the one for the Higgs–Kibble model. However for brevity we present it in a slightly different order than the one we followed in Section 2.1. Namely we start by first working out the implications of the Slavnov–Taylor identities on the propagator and next we apply the result to the proof of the Ward identity.

The Slavnov–Taylor identity in eq. (2.7) gives, similarly to eq. (2.25)

$$(f G_L f^t)_{ab} = -\delta_{ab}, \;\Rightarrow\; f G_L F = -f. \qquad (3.14)$$

From here we immediately derive the analog of eq. (2.26):

$$(f G_L \widetilde{\Gamma})_{a\dot{I}} = 0. \qquad (3.15)$$

The above equation has $N_V + N_S$ components for each "$a$", labeled by a capital dotted index $\dot{I}$ spanning vector ($\dot{I} = b$) and scalar ($\dot{I} = \tilde{a}$) indices. We now write down its vector and scalar components separately by expressing $\widetilde{\Gamma}$ in the block form

$$\widetilde{\Gamma} = \begin{pmatrix} \widetilde{\Gamma}_{VV} & \widetilde{\Gamma}_{VS} \\ \widetilde{\Gamma}_{VS}^t & \widetilde{\Gamma}_{SS} \end{pmatrix}, \qquad (3.16)$$

where $\widetilde{\Gamma}_{VV}$ is a $N_V \times N_V$ symmetric matrix, $\widetilde{\Gamma}_{SS}$ is symmetric and $N_S \times N_S$, and $\widetilde{\Gamma}_{VS}$ is a $N_V \times N_S$ matrix with components $[\widetilde{\Gamma}_{VS}]_{a\tilde{a}}$. We obtain

$$\sum_b (f G_V)_{ab}[\widetilde{\Gamma}_{VV}]_{bc} = -\sum_{\tilde{a}}(f G_S)_{a\tilde{a}}[\widetilde{\Gamma}_{VS}]_{c\tilde{a}},$$

$$\sum_{\tilde{a}} (f G_S)_{a\tilde{a}}[\widetilde{\Gamma}_{SS}]_{\tilde{a}\tilde{b}} = -\sum_b (f G_V)_{ab}[\widetilde{\Gamma}_{VS}]_{b\tilde{b}}, \qquad (3.17)$$

where we introduced a compact notation $(f G_L)_{a\dot{I}} = \{(f G_V)_{ab}, (f G_S)_{a\tilde{a}}\}$ for the vector and the scalar components of $f G_L$.

We now notice that the matrix $f G_L$ has rank $N_V$ like $f$, because $G_L$ is invertible up to isolated poles. The matrix $\widetilde{\Gamma}_{SS}$ is also invertible up to isolated poles, from which we can easily

conclude by eq. (3.17) that $(f G_V)$ has maximal rank $N_V$. Indeed if $\sum_a w_a (f G_V)_{ab} = 0$ for some non-vanishing $w_a$, we could contract $w$ with the second line of eq. (3.17) and prove that also $\sum_a w_a (f G_S)_{a\tilde{a}} = 0$, given that $\widetilde{\Gamma}_{SS}$ is invertible. Hence $\sum_a w_a (f G_L)_{ai}$ would vanish, which cannot be since $f G_L$ has maximal rank. Exploiting that $(f G_V)$ has maximal rank and it is invertible, as well as $\widetilde{\Gamma}_{SS}$, eq. (3.17) can be turned into the generalization of eq. (2.28)

$$\widetilde{\Gamma}_{VV} = \widetilde{\Gamma}_{VS} \widetilde{\Gamma}_{SS}^{-1} \widetilde{\Gamma}_{VS}^t, \tag{3.18}$$

plus a relation analog to the second equality in eq. (2.27)

$$(f G_V)^{-1} (f G_S) = -\widetilde{\Gamma}_{VS} \widetilde{\Gamma}_{SS}^{-1}. \tag{3.19}$$

The reason for considering this particular combination of $(f G_V)$ and $(f G_S)$ will become clear in the following paragraph.

We next derive the generalized Ward identities exactly like we did in Section 2.1 for the Higgs–Kibble model. Being $\mathcal{A}\{\Phi^N[k]\}$ the amputated connected amplitude as in eq. (2.18), by applying the Slavnov–Taylor identity (2.17) for $n = 1$ we find

$$\Big( \sum_{i,j} f_{ai} [G_L]_{ij} \mathcal{P}_{jM}[k] \Big) \mathcal{A}\{\Phi^M[k]\} = 0. \tag{3.20}$$

We then define a set of vectors

$$\begin{aligned} \mathcal{K}_{aM}[k] &\equiv -\sum_b [(f G_V)^{-1}]_{ab} \Big( \sum_{i,j} f_{bi} [G_L]_{ij} \mathcal{P}_{jM}[k] \Big) \\ &= -\mathcal{P}_{aM}[k] + [\widetilde{\Gamma}_{VS} \widetilde{\Gamma}_{SS}^{-1}]_{a\tilde{a}} \mathcal{P}_{\tilde{a}M}[k], \end{aligned} \tag{3.21}$$

and write the Ward identity as

$$\mathcal{K}_{aM}[k] \mathcal{A}\{\Phi^M[k]\} = 0, \quad \forall a. \tag{3.22}$$

The generalization to an arbitrary number of external $\Phi$ legs is straightforward

$$\mathcal{K}_{a_1 M_1}[k_1] \cdots \mathcal{K}_{a_n M_n}[k_n] \mathcal{A}\{\Phi^{M_1}[k_1], \cdots, \Phi^{M_n}[k_n]\} = 0, \quad \forall a_1, \dots, a_n. \tag{3.23}$$

Clearly the $\mathcal{K}_a$'s (one for each gauge fields) correspond to the scalar polarization vector we encountered in the Higgs–Kibble model. The difference is that each of them is now a vector in the $(4 N_V + N_S)$-dimensional space spanned by $M$, namely

$$\mathcal{K}_{aM}[k] = \begin{pmatrix} i \dfrac{k_\mu}{k} \delta_{aa'} & \text{for } M = \{\mu, a'\} \\ [\mathcal{K}_\pi(k^2)]_{a\tilde{a}'} \equiv [\widetilde{\Gamma}_{VS} \widetilde{\Gamma}_{SS}^{-1}]_{a\tilde{a}'} & \text{for } M = \tilde{a}' \end{pmatrix}. \tag{3.24}$$

Notice that $\mathcal{K}_\pi(k^2)$ is now a $N_V \times N_S$ matrix of form-factors, but it plays here the same role as in the Higgs–Kibble model. In particular the Ward identity (3.22) can be written as

$$i k_\mu \mathcal{A}\{V_a^\mu[k]\} = -k \sum_{\tilde{a}} [\mathcal{K}_\pi(k^2)]_{a\tilde{a}} \mathcal{A}\{(\phi - \langle\phi\rangle)_{\tilde{a}}[k]\}, \tag{3.25}$$

showing how $\mathcal{K}_\pi$ connects the high-energy limit of amplitudes involving longitudinal vectors (whose polarization vector approaches $k_\mu$) to amplitudes involving scalars.

Before moving forward and showing how the Ward identities lead to the definition of well-behaved longitudinal polarization vectors and propagators, it is interesting to outline some particular aspects of the general results of this section. We first consider a gauge theory without scalar fields such as QED or QCD. In our formalism we can recover this case in the limit

where some scalars are actually present in the theory, such that $\widetilde{\Gamma}_{SS}$ is non-vanishing and invertible, but they are decoupled. This means in particular that $\widetilde{\Gamma}_{VS}$ vanishes and correspondingly $\mathcal{K}_\pi = \widetilde{\Gamma}_{VS}\widetilde{\Gamma}_{SS}^{-1} = 0$. Therefore the Ward identities reduce to the familiar $k_\mu \mathcal{A}^\mu = 0$ relations. Moreover in this limit eq. (3.18) becomes $\widetilde{\Gamma}_{VV} = 0$. Recall that $\widetilde{\Gamma}_{VV}$ parametrizes (see eq. (3.13) and (3.16)) the contributions to the longitudinal vector-vector inverse propagator that emerge from radiative corrections on top of those (equal to $-k^2\Xi$) of the gauge-fixing term. The condition $\widetilde{\Gamma}_{VV} = 0$ thus means that the longitudinal propagator equals $-\Xi^{-1}/k^2$ to all orders in perturbation theory, which matches the standard formula where $\Xi = \xi^{-1}$. Also notice that $\widetilde{\Gamma}_{VV}$ is connected with the transverse inverse propagator matrix $\Gamma_\perp$ at zero momentum. This is because the inverse propagator is regular, therefore the $-k^\mu k^\nu/k^2$ singularity of the transverse projector in eq. (3.13) must be compensated by the $k^\mu k^\nu/k^2$ singularity in the vector–vector part of the $\mathcal{P}_{aM}\mathcal{P}_{bN}$ term. Therefore

$$\Gamma_\perp(k^2) = \widetilde{\Gamma}_{VV}(0) + k^2\,\Gamma_\perp'(0) + \mathcal{O}(k^4). \tag{3.26}$$

Since $\widetilde{\Gamma}_{VV}$ vanishes, we have proven that all components of the transverse propagator $\Gamma_\perp^{-1}$ have a pole at $k^2 = 0$ and all the vectors are massless at all orders, as they should in an unbroken theory.

In a general gauge theory, eq. (3.26) should be supplemented with another regularity condition (note that here $'$ indicates $d/dk$, rather than $d/dk^2$)

$$\widetilde{\Gamma}_{VS}(k) = k\left[\widetilde{\Gamma}_{VS}'(0) + \mathcal{O}(k^2)\right], \tag{3.27}$$

as it follows from the need of canceling the singularity in the vector-scalar propagator that emerges from the $k^\mu/k$ term in $\mathcal{P}_{aM}$. Eq. (3.18) thus implies that $\widetilde{\Gamma}_{VV}(0)$ can be non-vanishing, and in turn $\Gamma_\perp(0) \neq 0$ so that some of the vectors can acquire a mass, only provided $\widetilde{\Gamma}_{SS}^{-1}$ has a massless pole. More precisely we see that the rank of $\Gamma_\perp(0)$, i.e. the number of massive vectors, is smaller or equal than the rank of $k^2\widetilde{\Gamma}_{SS}^{-1}$ at $k^2 = 0$, which is nothing but the standard Higgs mechanism.

## 3.3 Equivalent Propagator and Longitudinal Vectors

The discussion of the present section follows very closely the one in Section 2.3 for the Higgs–Kibble model. Actually several derivations are identical and will not be repeated here. The goal is to define longitudinal polarization vectors that are well behaved in energy and derive an equivalent form of the propagator, which decomposes in terms of these vectors and is thus also well-behaved. We start by defining the polarization vectors as an obvious generalization of eq. (2.36)

$$(\mathcal{E}_a^0[k])_M \equiv (\mathcal{E}_{st.,a}^0[k] + i\,\mathcal{K}_a[+k])_M \overset{\text{if } \Re(k_0)>0}{=} \begin{pmatrix} e_\mu^0[k]\delta_{aa'} & \text{for } M = \{\mu, a'\} \\ +i\,[\mathcal{K}_\pi(k^2)]_{a\,\tilde{a}'} & \text{for } M = \tilde{a}' \end{pmatrix},$$

$$(\overline{\mathcal{E}}_a^0[k])_M \equiv (\overline{\mathcal{E}}_{st.,a}^0[k] - i\,\mathcal{K}_a[-k])_M \overset{\text{if } \Re(k_0)>0}{=} \begin{pmatrix} e_\mu^0[k]\delta_{aa'} & \text{for } M = \{\mu, a'\} \\ -i\,[\mathcal{K}_\pi(k^2)]_{a\,\tilde{a}'} & \text{for } M = \tilde{a}' \end{pmatrix}, \tag{3.28}$$

where $e_\mu^0[k]$ was introduced in eq. (2.35). The standard longitudinal polarization vectors $\mathcal{E}_{st.,a}^0$ and $\overline{\mathcal{E}}_{st.,a}^0$ that appear in the equation above, and the transverse ones that will appear later, are simply equal to the 4D vectors $\varepsilon_\mu^h$ (see e.g. eq. (2.34)) times $\delta_{aa'}$ for $M = \{\mu, a'\}$ and they vanish for $M = \tilde{a}$. The polarization vectors will be evaluated on virtual particle momentum $k_\mu$ whose energy component has positive real part as in Section 2.3.

It is straightforward to decompose the propagator in terms of the new polarization vectors. By the standard 4D completeness relation we have

$$\mathcal{P}_{MN}^{\perp;\,ab}[k] = -\sum_{h=\pm,0} (\mathcal{E}_{st.,a}^h[k])_M (\overline{\mathcal{E}}_{st.,b}^h[k])_N, \tag{3.29}$$

which allows us to decompose the transverse part of the propagator (3.13). We further rewrite the longitudinal term similarly to eq. (2.42), obtaining

$$G_{MN}[k] = -\sum_{h=\pm,0} \mathcal{E}_{aM}^h[k][\Gamma_\perp^{-1}]_{ab}\overline{\mathcal{E}}_{bN}^h[k] + \mathcal{P}_{iM}[k][(\widetilde{\Gamma}-F)^{-1}]_{ij}\mathcal{P}_{jN}[-k] \tag{3.30}$$

$$-\left(i\,\mathcal{E}_a^0[k]+\frac{1}{2}\mathcal{K}_a[k]\right)_M[\Gamma_\perp^{-1}]_{ab}(\mathcal{K}_b[-k])_N - (\mathcal{K}_a[k])_M[\Gamma_\perp^{-1}]_{ab}\left(-i\,\overline{\mathcal{E}}_b^0[k]+\frac{1}{2}\mathcal{K}_b[-k]\right)_N.$$

Finally we get rid of the residual anomalous energy growth by eliminating the gauge (i.e., $\dot{I}=a$) $\mathcal{P}_{iM}$'s vectors in favor of the $\mathcal{K}_{aM}$'s and we follow the exact same steps that led us to eq. (2.47) in the Higgs–Kibble model. We obtain (the sum over repeated indices is understood)

$$G_{MN}[k] = G_{MN}^{\text{eq}}[k] - \mathcal{K}_{aM}[k]\mathcal{V}_{aN}[-k] - \mathcal{V}_{aM}[k]\mathcal{K}_{aN}[-k], \tag{3.31}$$

in terms of some vectors $\mathcal{V}_a$ and with

$$G_{MN}^{\text{eq}}[k] \equiv -\sum_{h=\pm,0}\mathcal{E}_{aM}^h[k][\Gamma_\perp^{-1}]_{ab}\overline{\mathcal{E}}_{bN}^h[k] + \mathcal{P}_{\tilde{a}M}[k][\widetilde{\Gamma}_{SS}^{-1}]_{\tilde{a}\tilde{b}}\mathcal{P}_{\tilde{b}N}[-k]. \tag{3.32}$$

Notice that eq. (2.45), which is readily seen to apply also for a general gauge theory, needs to be used in order to obtain this result. The final step consists in proving that $G^{\text{eq}}$ can be used in place of $G$ in resonant processes. The proof relies on the generalized Ward identities in eq. (3.23) and is completely identical to the one presented at the end of Section 2.3 for the Higgs–Kibble model.

**On-Shell Vectors**

In preparation for the study of the SM in the next section, we now discuss the structure of the propagator around its poles $k^2 = M_V^2$ associated to spin-one particles and derive the corresponding Feynman rules. The mass $M_V$ is complex for an unstable particle, but on-shell scattering amplitudes and the associated Feynman rules can still be defined in terms of the pole residue as explained in Section 2.3.

We start from massive vectors, $M_V \neq 0$. Barring the peculiar situation where a scalar resonance happens to have the exact same mass, the scalar part of the propagator in eq. (3.32) does not contribute to the pole, which entirely emerges from $\Gamma_\perp^{-1}$. Assuming furthermore for notational simplicity that no vectors are degenerate in mass,[5] we have

$$\lim_{k^2 \to M_V^2}(k^2 - M_V^2)(\Gamma_\perp^{-1})_{ab} = -\sqrt{Z_{Va}}\sqrt{Z_{Vb}}, \tag{3.33}$$

namely the residue of $\Gamma_\perp^{-1}$ has unit rank and can be expressed as the matrix product of a wave-function vector $\sqrt{Z_{Va}}$. We thus arrive at

$$\lim_{k^2 \to M_V^2}\left\{(k^2 - M_V^2)G_{MN}^{\text{eq}}[k]\right\} = \sum_{h=\pm,0}\sqrt{Z_{Va}}\mathcal{E}_{aM}^h[k]\Big|_{k^2=M_V^2}\sqrt{Z_{Vb}}\,\overline{\mathcal{E}}_{bN}^h[k]\Big|_{k^2=M_V^2}, \tag{3.34}$$

where the sum over "$a$" and "$b$" is understood. We can now define the amplitude for the creation/annihilation of a massive vector resonance with on-shell momentum $k$ (i.e., $k^2 = M_V^2$, $\Re(k_0) > 0$), and helicity $h = \pm, 0$, as

$$i\mathcal{M}(\alpha \to \beta + V_h[k]) \equiv \sum_a \sqrt{Z_{Va}}\,\overline{\mathcal{E}}_{aM}^h[k]\mathcal{A}\left\{\Phi^M[-k]\right\}, \tag{3.35}$$

$$i\mathcal{M}(\alpha + V_h[k] \to \beta) \equiv \sum_a \sqrt{Z_{Va}}\,\mathcal{A}\left\{\Phi^M[k]\right\}\mathcal{E}_{aM}^h[k], \tag{3.36}$$

---

[5]Degeneracies due to symmetry can be easily dealt with like in the Higgs–Kibble model.

where $\mathcal{A}$ denotes the amputated amplitude. If the vectors are stable asymptotic particles, the above Feynman rules can also be derived by the LSZ reduction formula.

Notice that the vector index "$a$" is summed over in eq.s (3.35) and (3.36). This is a consequence of the fact that the resonance is in general interpolated by several fields as in the standard formalism. What is different in our formalism is that the polarization vectors (3.28) have components $M = \tilde{a}$ along the scalar fields and that these components are not universal and theory-independent but rather they are theory-specific since they are proportional to $[\mathcal{K}_\pi]_{a\tilde{a}}$ evaluated at $k^2 = M_V^2$. In the standard formalism one can work in the "pole scheme", namely reabsorb $\sqrt{Z_{Va}}$ in a redefinition of the vector fields such that a single one interpolates for the resonance and no summation over "$a$" appears in the Feynman rules. We may do the same in our formalism by reabsorbing also $\sqrt{Z_V} \cdot \mathcal{K}_\pi(M_V^2)$ in the scalar fields, however taking this step would bring no practical advantage in the applications that follow.

For massless vectors, $M_V = 0$, the Feynman rules are the standard ones. We could establish this fact by just working with the standard "$G$" propagator, never use eq. (3.31) to turn it into $G^{\text{eq}}$, and going through the standard textbook discussion. It is however an interesting consistency cross-check to verify the result by starting directly from eq. (3.32). We work for simplicity under the assumption that all the scalars involved in eq. (3.32) are associated to massless would-be Goldstone bosons, eaten by the massive vectors. More precisely we assume that

$$\widetilde{\Gamma}_{SS}(k^2) = k^2 \widetilde{\Gamma}'_{SS}(0)[1 + \mathcal{O}(k^2)], \tag{3.37}$$

and that $\widetilde{\Gamma}_{VV}(0)$ has rank $N_S$ (that implies $N_S \leq N_V$). Since $\widetilde{\Gamma}_{VV}(0) = \Gamma_\perp(0)$ is the vector bosons mass-matrix (see eq. (3.26)), this is just the statement that the theory has $N_S$ massive and $N_V - N_S$ massless vectors. The above assumption is realized in the SM.

The transverse "$h = \pm 1$" terms in eq. (3.32) possess a pole at $k^2 = 0$ that is identical to the one of the standard propagator and thus produces the standard Feynman rules for massless vectors. We simply have to check that the "rest" of the propagator

$$\mathcal{R}_{MN}[k] \equiv -\mathcal{E}^0_{aM}[k][\Gamma_\perp^{-1}]_{ab}\overline{\mathcal{E}}^0_{bN}[k] + \mathcal{P}_{\tilde{a}M}[k][\widetilde{\Gamma}_{SS}^{-1}]_{\tilde{a}\tilde{b}}\mathcal{P}_{\tilde{b}N}[-k], \tag{3.38}$$

is regular at $k^2 = 0$. Recalling the explicit expression of $\mathcal{E}^0_{aM}[k]$ given in eq. (3.28) we see that $\mathcal{R}$ contains a vector–vector component $\mathrm{e}^0\Gamma_\perp^{-1}\mathrm{e}^0$, a vector–scalar component $-i\mathrm{e}^0(\Gamma_\perp^{-1})\mathcal{K}_\pi$ and a scalar–scalar component $\mathcal{R}_{SS} = \widetilde{\Gamma}_{SS}^{-1} - \mathcal{K}_\pi^{\ t}\Gamma_\perp^{-1}\mathcal{K}_\pi$. The vector–vector part is manifestly regular because $\mathrm{e}^0 \propto k$ and $\Gamma_\perp^{-1} \lesssim k^{-2}$. In order to deal with the other terms we recall the $k^2 \to 0$ behavior of the form-factor matrices in eq.s (3.26), (3.27) and (3.37) and that

$$\widetilde{\Gamma}_{VV}(0) + k^2\Gamma'_\perp(0) = [\widetilde{\Gamma}'_{VS}(0)][\widetilde{\Gamma}'_{SS}(0)]^{-1}[\widetilde{\Gamma}'_{VS}(0)]^t + k^2\Gamma'_\perp(0), \tag{3.39}$$

by eq. (3.18). The relation above implies in particular that under our hypotheses (that $\widetilde{\Gamma}_{VV}(0)$ and $\widetilde{\Gamma}'_{SS}(0)$ have rank $N_S$) the mixing matrix $\widetilde{\Gamma}'_{VS}(0)$ has rank $N_S$, therefore it can be written as

$$\widetilde{\Gamma}'_{VS}(0) = \Theta \begin{pmatrix} \mathbf{0}_{(N_V-N_S)\times N_S} \\ \overline{\Gamma}'_{VS} \end{pmatrix}, \tag{3.40}$$

by a suited orthogonal $N_V \times N_V$ matrix $\Theta$, where $\overline{\Gamma}'_{VS}$ is invertible and $N_S \times N_S$. Physically, the rotation $\Theta$ brings the vector fields into a basis in which the first $N_V - N_S$ vectors do not mix with the scalars and therefore they interpolate for the massless particles. It is not unique because rotations of the massless modes would leave that relation unchanged, but this ambiguity has no effect in our discussion. By employing eq.s (3.26) and (3.39) we see that, in the new basis, $\Gamma_\perp$ is proportional to $k^2$ in the upper left $(N_V-N_S)$–dimensional block while it is finite in the others, leading to

$$\Gamma_\perp^{-1}(k) = \Theta \begin{pmatrix} \mathcal{O}(k^{-2}) & \mathcal{O}(1) \\ \mathcal{O}(1) & [\overline{\Gamma}'^{\ t}_{VS}]^{-1}[\widetilde{\Gamma}'_{SS}(0)][\overline{\Gamma}'_{VS}]^{-1} + \mathcal{O}(k^2) \end{pmatrix} \Theta^t. \tag{3.41}$$

Finally from the definition of $\mathcal{K}_\pi$ in eq. (3.24) we find

$$\mathcal{K}_\pi(k) = k^{-1}\widetilde{\Gamma}'_{\mathrm{VS}}(0)[\widetilde{\Gamma}'_{\mathrm{SS}}(0)]^{-1} + \mathcal{O}(k) = \Theta \begin{pmatrix} \mathbf{0}_{(N_V - N_S) \times N_S} + \mathcal{O}(k) \\ k^{-1}\,\overline{\Gamma}'_{\mathrm{VS}}\widetilde{\Gamma}'_{\mathrm{SS}}(0) + \mathcal{O}(k) \end{pmatrix}, \tag{3.42}$$

and we can straightforwardly conclude that also the vector–scalar and scalar-scalar components of $\mathcal{R}$ are of order $k^0$. This is because the vector–scalar term is proportional to $e^0$, of $\mathcal{O}(k)$, times $\Gamma_\perp^{-1} \cdot \mathcal{K}_\pi$, which does not pick up the $\mathcal{O}(k^{-2})$ pole in $\Gamma_\perp^{-1}$ and is of $\mathcal{O}(k^{-1})$. The scalar–scalar component $\mathcal{R}_{\mathrm{SS}} = \widetilde{\Gamma}_{\mathrm{SS}}^{-1} - \mathcal{K}_\pi{}^t\Gamma_\perp^{-1}\mathcal{K}_\pi$ is also seen to be finite by direct substitution.

## 3.4 Renormalization Scheme (In-)Dependence

Nowhere in the present section we had to specify whether we have been working with bare or renormalized fields and parameters. All that matters for our derivations to apply is that the gauge-fixing functionals $\mathcal{F}_a$, as they are written down in eq. (3.1), are the "bare" gauge-fixing functionals by which the Faddeev-Popov quantization is carried on. Otherwise the Slavnov-Taylor identities and in particular eq. (2.7) would not hold true. Therefore if the fields $V_\mu$ and $\phi$ are bare, the gauge-fixing parameters $\Xi$ and $\widetilde{\mu}$ are the bare ones, while if the fields are renormalized, the gauge-fixing parameters have to be renormalized accordingly to preserve eq. (3.1). More precisely the point is that the gauge-fixing parameters $\Xi$ and $\widetilde{\mu}$ appearing (through the matrix $F$) in the definition (3.13) of $\widetilde{\Gamma}$ for renormalized fields are necessarily the ones renormalized with the above prescription. If this is the case all the results of the previous section hold in any field basis, in particular the polarization vectors are given by eq. (3.28) with

$$\mathcal{K}_\pi = \widetilde{\Gamma}_{\mathrm{VS}}\widetilde{\Gamma}_{\mathrm{SS}}^{-1}, \tag{3.43}$$

provided of course the form-factors are correctly interpreted as those of the corresponding fields. Any other gauge-fixing renormalization prescription can of course be adopted, but the definition of $\widetilde{\Gamma}$ in eq. (3.13) must be modified accordingly.

It is easy to relate the $\mathcal{K}_\pi$ matrices in two different field bases. For instance if the relation between bare "$(b)$" and renormalized "$(r)$" fields takes the form

$$\begin{aligned} V_{a\mu}^{(b)} &= [\mathcal{Z}_{\mathrm{VV}}]_{ab} V_{b\mu}^{(r)}, \\ \phi_{\tilde{a}}^{(b)} &= [\mathcal{Z}_{\mathrm{SS}}]_{\tilde{a}\tilde{b}} \phi_{\tilde{b}}^{(r)}, \end{aligned} \tag{3.44}$$

the renormalized $\mathcal{K}_\pi^{(r)}$, to be employed in our polarization vectors for renormalized fields connected amplitudes, is related to the bare one as

$$\mathcal{K}_\pi^{(r)} = \mathcal{Z}_{\mathrm{VV}}^t \mathcal{K}_\pi^{(b)} (\mathcal{Z}_{\mathrm{SS}}^t)^{-1}. \tag{3.45}$$

In particular this implies, since $\mathcal{K}_\pi^{(b)}$ is independent of the scale "$\mu$" employed for renormalization, the Callan-Symanzik equation

$$\mu\frac{\partial}{\partial\mu}\mathcal{K}_\pi^{(r)} + \sum_{C^{(r)}}\beta_{C^{(r)}}\frac{\partial}{\partial C^{(r)}}\mathcal{K}_\pi^{(r)} - \gamma_{\mathrm{VV}}^t\mathcal{K}_\pi^{(r)} + \mathcal{K}_\pi^{(r)}\gamma_{\mathrm{SS}}^t = 0, \tag{3.46}$$

where by $C^{(r)}$ we denote all renormalized parameters of the theory (and $\beta_{C^{(r)}} \equiv (\mu\, d/d\mu)C^{(r)}$), and $\gamma_{\mathrm{VV, SS}}$ are the anomalous dimension matrices

$$\begin{aligned} \mu\frac{d}{d\mu}\mathcal{Z}_{\mathrm{VV}} &= \mathcal{Z}_{\mathrm{VV}}\gamma_{\mathrm{VV}}, \tag{3.47} \\ \mu\frac{d}{d\mu}\mathcal{Z}_{\mathrm{SS}} &= \mathcal{Z}_{\mathrm{SS}}\gamma_{\mathrm{SS}}. \end{aligned}$$

Eq. (3.46) may be practically relevant when performing precise calculations of high energy processes, where the resummation of large logs $\ln \mu^2/M_V^2$ due to RG running might become necessary. Clearly for a successful resummation one should also run the wave-function "$\sqrt{Z}$" factors in front of the scattering amplitude in eq.s (3.35) and (3.36). The corresponding Callan-Symanzik equations are the standard ones and need not be discussed here.

# 4   The Standard Model

We are now ready to discuss Goldstone Equivalence in the Standard Model (SM) theory. We start, in Section 4.1, by setting up our notation and specifying the renormalization scheme. We next (in Section 4.2) specialize the general results of the previous section to the SM and express the relevant form-factor matrices in terms of 1PI vacuum polarization amplitudes for the charged and neutral SM bosonic fields. The $\mathcal{K}_\pi$ form-factors are computed explicitly at one-loop order in Section 4.3. Finally, we apply our formalism to the calculation of $W^+W^- \to W^+W^-$ at tree-level (Section 4.4.1) and of the $\mathcal{O}(y_t^2)$ radiative corrections to the $t \to Wb$ decay (Section 4.4.2). These simple processes are selected with the purpose of illustrating how concrete calculations are performed in our formalism and to provide a cross-check that the latter correctly reproduces standard results.

## 4.1   Setup

We work in renormalized perturbation theory, with a gauge-fixed Lagrangian

$$\mathcal{L} = \mathcal{L}_0 + \mathcal{L}_{\text{g.f.}} + \mathcal{L}_{\text{ghosts}} + \mathcal{L}_{\text{c.t.}} , \tag{4.1}$$

where $\mathcal{L}_{\text{c.t.}}$ contains the divergent counterterms.

The bosonic part of $\mathcal{L}_0$ reads [6]

$$\mathcal{L}_0^{\text{bos.}} = -\frac{1}{2}\text{Tr}\left[W_{\mu\nu}W^{\mu\nu}\right] - \frac{1}{4}B_{\mu\nu}B^{\mu\nu} + (D_\mu H)^\dagger D^\mu H - \lambda\left(|H|^2 - \frac{v^2}{2}\right)^2 , \tag{4.2}$$

where $W_\mu = W_\mu^a \sigma^a/2$ and $B_\mu$ denote the renormalized $\text{SU}(2)_L \times \text{U}(1)$ gauge fields. The associated field strength tensors are defined in the standard way, as well as the charge- and mass-eigenstates

$$
\begin{aligned}
W_{\pm\mu} &= \left(W_\mu^1 \mp iW_\mu^2\right)/\sqrt{2}, \\
Z_\mu &= c_{\text{w}}W_\mu^3 - s_{\text{w}}B_\mu , \\
A_\mu &= s_{\text{w}}W_\mu^3 + c_{\text{w}}B_\mu .
\end{aligned}
\tag{4.3}
$$

The fields $W_\pm$, $Z$ and $A$ diagonalize the mass-matrix of the renormalized Lagrangian $\mathcal{L}_0$, with "tree-level" masses $m_W^2 = g^2v^2/4$ and $m_Z^2 = m_W^2/c_{\text{w}}^2$. The sine and the cosine of the Weak angle, $s_{\text{w}}$ and $c_{\text{w}}$, are defined as $s_{\text{w}}/c_{\text{w}} = g'/g$ in terms of the renormalized gauge couplings $g$ and $g'$. The actual complex masses of the $W$ and $Z$ bosons will be denoted with capital letter, i.e. $M_{W,Z}^2 \in \mathbb{C}$.

The Higgs is a doublet with Hypercharge $+1/2$, which we parameterize as

$$H = \frac{1}{\sqrt{2}}\begin{pmatrix} -i\sqrt{2}\,\pi_+ \\ v + h + i\pi_0 \end{pmatrix} , \tag{4.4}$$

---

[6]Matter fermions and QCD interactions, included in $\mathcal{L}_0$, need not be discussed explicitly.

in terms of the physical Higgs $h$ and of the Goldstone bosons $\pi_0$ and $\pi_+ = \pi_-^\dagger$. This parametrization (see eq. (2.2)) makes the implications of the custodial $SU(2)_c$ approximate symmetry more transparent resulting in simpler tree-level formulas. Under the CP symmetry, $\pi_0$ is odd like the $Z$ and the photon, while $\pi_+ \to -\pi_-$ similarly to the $W$'s. The physical Higgs field $h$ is CP-even and its "tree-level" mass is $m_H^2 = 2\lambda v^2$.

The gauge-fixing Lagrangian is taken to preserve Lorentz, charge and CP symmetry, namely

$$\mathcal{L}_{\text{g.f.}} = -\mathcal{F}_+\mathcal{F}_- - \frac{1}{2}\mathcal{F}_A^2 - \frac{1}{2}\mathcal{F}_Z^2 , \tag{4.5}$$

with linear gauge-fixings functionals of the form

$$
\begin{aligned}
\mathcal{F}_- &= \partial_\mu W_-^\mu/\sqrt{\xi} - \sqrt{\xi}\,\widetilde{m}_W \pi_- = \mathcal{F}_+^\dagger , \\
\mathcal{F}_A &= (\partial_\mu A^\mu + \theta_Z \partial_\mu Z^\mu)/\sqrt{\alpha} - \sqrt{\alpha}\,\widetilde{m}_A \pi_Z , \\
\mathcal{F}_Z &= (\partial_\mu Z^\mu + \theta_A \partial_\mu A^\mu)/\sqrt{\eta} - \sqrt{\eta}\,\widetilde{m}_Z \pi_Z .
\end{aligned}
\tag{4.6}
$$

The associated ghost Lagrangian is $\mathcal{L}_{\text{ghosts}} = -\overline{\omega}_A \delta_\omega \mathcal{F}_A - \overline{\omega}_Z \delta_\omega \mathcal{F}_Z - \overline{\omega}_+ \delta_\omega \mathcal{F}_- - \overline{\omega}_- \delta_\omega \mathcal{F}_+$, where $\delta_\omega$ is an infinitesimal gauge transformation with ghost parameters. Its explicit form need not be reported here. The results of Section 4.2 will hold for any gauge-fixing in this class. Explicit calculations will be performed in the Feynman—'t Hooft gauge $\xi = \alpha = \eta = 1$, $\widetilde{m}_{W/Z} = m_{W/Z}$ and $\theta_{A,Z} = 0$.

The counterterms are obtained from the bare version of the Lagrangian by introducing multiplicative renormalization constants for the bare parameters $g_0$, $g_0'$, $\lambda_0$ and $v_0 \equiv \mu_0/\sqrt{\lambda_0}$, plus wave-function renormalizations for the $W_0$, $B_0$ and $H_0$ fields and an independent shift for the bare physical Higgs field $h_0$. The ghosts and the matter fermion fields are also renormalized, as well as the Yukawa couplings. The bare gauge-fixing parameters are renormalized in order to compensate for the wave-function renormalization of the fields, such as to ensure that the renormalized $\mathcal{F}$'s in eq. (4.6) are equal to the bare $\mathcal{F}$'s through which Faddeev–Popov quantization is carried on. This is important because the $\widetilde{\Gamma}$ form-factors are defined in eq. (3.12) by subtracting the contribution of the complete bare gauge-fixing Lagrangian to the inverse propagator (see also Section 3.4).

Loops are evaluated in Dimensional Regularization and the counterterms are fixed with the $\overline{\text{MS}}$ prescription. A conceptually and practically convenient alternative (e.g. [44]) is to require complete cancellation of the Higgs field tadpole, including its finite part.[7] Or, which is the same, to require that the renormalized $h$ field has exactly zero VEV. Both options will be considered in what follows.

## 4.2 The Goldstone-Equivalent Standard Model

We now apply to the SM the general results of Section 3. The bosonic fields consist of 4 vectors and 4 scalars, however thanks to symmetries we do not need to study all of them simultaneously. Charge conservation forbids mixings between the charged ($W^\pm$ and $\pi^\pm$) and the neutral ($Z$, $A$, $\pi_0$ and $h$) fields, allowing us to treat the charged and the neutral sectors separately. One further simplification emerges in the neutral sector because of the CP symmetry. CP is broken in the SM, but CP-breaking $h$ mixings with $Z$, $A$ or $\pi_0$ are suppressed by the Jarlskog invariant, of order $10^{-5}$, and furthermore first emerge at 3 loops. Since the effect is very small we can safely neglect it in all practical purposes, and consider a restricted neutral sector consisting of the $Z$, $A$ and $\pi_0$ fields. Notice however that our general formalism would allow to take $h$ mixing into account, and that this might be relevant in extensions of the SM

---

[7]Specifically, we add a counterterm proportional to the Higgs doublet mass-term, $|H|^2$, with a finite coefficient set by requiring tadpole cancellation.

$$V_a^\mu \xrightarrow{k} \xrightarrow{k} V_b^\nu \quad = \quad i\left(\eta^{\mu\nu} - \frac{k^\mu k^\nu}{k^2}\right)\Pi^T_{V_a V_b}(k^2) + i\frac{k^\mu k^\nu}{k^2}\Pi^L_{V_a V_b}(k^2)$$

$$\pi_{\tilde{a}} \xrightarrow{k} \xrightarrow{k} V_b^\mu \quad = \quad k^\mu \Pi_{V_b \pi_{\tilde{a}}}(k^2)$$

$$\pi_{\tilde{a}} \xrightarrow{k} \xrightarrow{k} \pi_{\tilde{b}} \quad = \quad i\Pi_{\pi_{\tilde{a}} \pi_{\tilde{b}}}(k^2)$$

Figure 4: Diagrammatic definition of the 1PI amplitudes. The arrow denotes momentum flow.

with larger CP-breaking effects. Also note that here we are exploiting the implications of symmetries on gauge-dependent quantities such as the 2-point functions. It is thus essential that the gauge-fixing respects the symmetries as we assumed in eq. (4.6).

**Charged Sector**

The relevant degrees of freedom are encoded in one complex vector with $N_S = N_V = 1$

$$\Phi_\pm^M = \begin{pmatrix} W_\pm^\mu \\ \pi_\pm \end{pmatrix}. \tag{4.7}$$

In Section 3 we parametrized all the degrees of freedom in terms of real fields, however it is straightforward to adapt the results to the complex notation by regarding the real and imaginary parts of $\Phi_\pm$ as a doublet of the electromagnetic U(1) symmetry. The CP symmetry, which is an excellent approximation in the present context as discussed above, also needs to be employed for the results that follow. In particular it ensures that the $\widetilde{\Gamma}$ matrix is symmetric as in eq. (4.9).

The form factors $\Gamma_\perp$ and $\widetilde{\Gamma}$ are defined in eq. (3.12) and consist of a tree-level contribution plus vacuum polarization terms "$\Pi(k^2)$" due to radiative corrections. The $\Pi$'s parametrize the amputated 1PI 2-point functions as shown in Figure 4. With this notation the transverse form factor $\Gamma_\perp$ reads

$$\Gamma_\perp = m_W^2 - k^2 + \Pi^T_{WW}, \tag{4.8}$$

whereas the longitudinal form factor matrix $\widetilde{\Gamma}$ is given by

$$\widetilde{\Gamma} = \begin{pmatrix} \widetilde{\Gamma}_{VV} & \widetilde{\Gamma}_{VS} \\ \widetilde{\Gamma}_{VS} & \widetilde{\Gamma}_{SS} \end{pmatrix} = \begin{pmatrix} m_W^2 & k\,m_W \\ k\,m_W & k^2 \end{pmatrix} + \begin{pmatrix} \Pi^L_{WW} & k\Pi_{W\pi} \\ k\Pi_{W\pi} & \Pi_{\pi\pi} \end{pmatrix}. \tag{4.9}$$

Notice that the tree-level contribution of the gauge-fixing term, encapsulated in the "$F$" matrix in eq. (3.12), has been duly subtracted from the definition of $\widetilde{\Gamma}$. The constraint in eq. (3.18) among the longitudinal form factors, as dictated by the Slavnov–Taylor identity, translates into the relation

$$\det\widetilde{\Gamma} = [m_W^2 + \Pi^L_{WW}][k^2 + \Pi_{\pi\pi}] - k^2[m_W + \Pi_{W\pi}]^2 = 0, \tag{4.10}$$

among the longitudinal $\Pi$'s.

The expressions of the form-factor in terms of the 1PI amplitudes, to be computed at each order in perturbation theory, give operative meaning to the results of Section 3. Namely we could now evaluate explicitly the equivalent $W$ propagator in eq. (3.32) and the well-behaved

longitudinal polarization vectors in eq. (3.28). For this we need the $\mathcal{K}_\pi$ matrix defined in eq. (3.24), which in the charged sector reduces to a single form factor

$$[\mathcal{K}_\pi]_{W\pi} = \frac{k[m_W + \Pi_{W\pi}]}{k^2 + \Pi_{\pi\pi}} = \sqrt{\frac{m_W^2 + \Pi_{WW}^L}{k^2 + \Pi_{\pi\pi}}} = \frac{m_W}{k} + \mathcal{O}(\text{loop}),$$ (4.11)

which we wrote in two different but equivalent forms by the constraint in eq. (4.10). Finally, the Equivalent Feynman rules for longitudinal $W$ bosons external states are obtained as a straightforward application of eq.s (3.35) and (3.36)

$$
\begin{aligned}
i\mathcal{M}(\alpha \to \beta + W_{h=0}^\pm(k)) &= \sqrt{Z_{WW}} \left[ \overline{\mathcal{E}^0}_{W^\pm M}[k] \mathcal{A}\{\Phi_\pm^M[-k]\}\right]_{k^2=M_W^2} \\
&= \sqrt{Z_{WW}} \left[ e_\mu^0[k] \mathcal{A}\{W_\pm^\mu[-k]\} - i[\mathcal{K}_\pi]_{W\pi}\mathcal{A}\{\pi_\pm[-k]\}\right]_{k^2=M_W^2}, \\
i\mathcal{M}(\alpha + W_{h=0}^\pm(k) \to \beta) &= \sqrt{Z_{WW}} \left[ \mathcal{E}^0_{W^\pm M}[k] \mathcal{A}\{\Phi_\pm^M[k]\}\right]_{k^2=M_W^2} \\
&= \sqrt{Z_{WW}} \left[ e_\mu^0[k] \mathcal{A}\{W_\pm^\mu[k]\} + i[\mathcal{K}_\pi]_{W\pi}\mathcal{A}\{\pi_\pm[k]\}\right]_{k^2=M_W^2}.
\end{aligned}
$$ (4.12)

As in the standard Feynman rules, $M_W^2$ is defined by the condition $\Gamma_\perp(M_W^2) = 0$ and $Z_{WW}$ is the wave-function factor $Z_{WW}^{-1} = \lim_{k^2 \to M_W^2}[(M_W^2 - k^2)\Gamma_\perp]$. Not surprisingly, all these results are identical in form to the ones we obtained for the Higgs–Kibble model in Section 2.

**Neutral Sector**

The neutral sector fields form a real multiplet

$$\Phi_0^M = \begin{pmatrix} A^\mu \\ Z^\mu \\ \pi_0 \end{pmatrix},$$ (4.13)

with $N_V = 2$, $N_S = 1$. Notice that CP invariance enforces $\langle \pi_0 \rangle = 0$. This allows $\pi_0$ to appear in the gauge-fixing functionals as in eq. (4.6) and makes our definition of $\Phi_0^M$ comply with eq. (3.2). By parametrizing the 1PI neutral 2-point functions in terms of tree-level contribution plus vacuum polarization terms, the $\Gamma_\perp$ and $\widetilde{\Gamma}$ form-factor matrix are immediately obtained by comparing with the general definition in eq. (3.12) to Figure 4. The transverse $\Gamma_\perp$ is a $2 \times 2$ matrix

$$\Gamma_\perp = \begin{pmatrix} \Pi_{AA}^T - k^2 & \Pi_{ZA}^T \\ \Pi_{ZA}^T & m_Z^2 - k^2 + \Pi_{ZZ}^T \end{pmatrix},$$ (4.14)

that includes the mixing between $Z$ and $A$ due to radiative corrections. The longitudinal $\widetilde{\Gamma}$ reads

$$\widetilde{\Gamma} = \begin{pmatrix} \widetilde{\Gamma}_{VV} & \widetilde{\Gamma}_{VS} \\ \widetilde{\Gamma}_{VS}^t & \widetilde{\Gamma}_{SS} \end{pmatrix} = \begin{pmatrix} 0 & 0 & 0 \\ 0 & m_Z^2 & k\,m_Z \\ 0 & k\,m_Z & k^2 \end{pmatrix} + \begin{pmatrix} \Pi_{AA}^L & \Pi_{ZA}^L & k\Pi_{A\pi} \\ \Pi_{ZA}^L & \Pi_{ZZ}^L & k\Pi_{Z\pi} \\ k\Pi_{A\pi} & k\Pi_{Z\pi} & \Pi_{\pi\pi} \end{pmatrix}.$$ (4.15)

The matrices $\widetilde{\Gamma}_{VV}$, $\widetilde{\Gamma}_{VS}$, and $\widetilde{\Gamma}_{SS}$ are defined in general in eq. (3.16). In the particular case at hand they are $2 \times 2$, $2 \times 1$ and $1 \times 1$ (i.e., a single number), respectively. The constraint in eq. (3.18) translates into three independent relations among the six $\Pi$'s

$$
\begin{aligned}
\Pi_{AA}^L[k^2 + \Pi_{\pi\pi}] &= k^2 \Pi_{A\pi}^2, \\
[m_Z^2 + \Pi_{ZZ}^L][k^2 + \Pi_{\pi\pi}] &= k^2[m_Z + \Pi_{Z\pi}]^2, \\
\Pi_{ZA}^L[k^2 + \Pi_{\pi\pi}] &= k^2[m_Z + \Pi_{Z\pi}]\Pi_{A\pi}.
\end{aligned}
$$ (4.16)

From eq. (3.24) we obtain $\mathcal{K}_\pi$, which is a 2-vector

$$
\begin{pmatrix} [\mathcal{K}_\pi]_{A\pi} \\ [\mathcal{K}_\pi]_{Z\pi} \end{pmatrix} = \begin{pmatrix} \dfrac{k\Pi_{A\pi}}{k^2 + \Pi_{\pi\pi}} \\ \dfrac{k[m_Z + \Pi_{Z\pi}]}{k^2 + \Pi_{\pi\pi}} \end{pmatrix} = \begin{pmatrix} 0 + \mathcal{O}(\text{loop}) \\ \dfrac{m_Z}{k} + \mathcal{O}(\text{loop}) \end{pmatrix}. \tag{4.17}
$$

Eq.s (3.35) and (3.36) gives us the longitudinal $Z$ boson Feynman rule

$$
\begin{aligned}
i\mathcal{M}(\alpha \to \beta + Z_{h=0}(k)) &= \left[ \sqrt{Z_{ZA}}\, \overline{\mathcal{E}^0}_{AM}[k] \mathcal{A}\{\Phi^M[-k]\} + \sqrt{Z_{ZZ}}\, \overline{\mathcal{E}^0}_{ZM}[k] \mathcal{A}\{\Phi^M[-k]\} \right]_{k^2 = M_Z^2} \\
&= \left[ \sqrt{Z_{ZA}}\, e^0_\mu[k] \mathcal{A}\{A^\mu[-k]\} + \sqrt{Z_{ZZ}}\, e^0_\mu[k] \mathcal{A}\{Z^\mu[-k]\} \right. \\
&\quad \left. - i\left( \sqrt{Z_{ZA}} [\mathcal{K}_\pi]_{A\pi} + \sqrt{Z_{ZZ}} [\mathcal{K}_\pi]_{Z\pi} \right) \mathcal{A}\{\pi_0[-k]\} \right]_{k^2 = M_Z^2}, \\
i\mathcal{M}(\alpha + Z_{h=0}(k) \to \beta) &= \left[ \sqrt{Z_{ZA}}\, \mathcal{E}^0_{AM}[k] \mathcal{A}\{\Phi^M[k]\} + \sqrt{Z_{ZZ}}\, \mathcal{E}^0_{ZM}[k] \mathcal{A}\{\Phi^M[k]\} \right]_{k^2 = M_Z^2} \\
&= \left[ \sqrt{Z_{ZA}}\, e^0_\mu[k] \mathcal{A}\{A^\mu[k]\} + \sqrt{Z_{ZZ}}\, e^0_\mu[k] \mathcal{A}\{Z^\mu[k]\} \right. \\
&\quad \left. + i\left( \sqrt{Z_{ZA}} [\mathcal{K}_\pi]_{A\pi} + \sqrt{Z_{ZZ}} [\mathcal{K}_\pi]_{Z\pi} \right) \mathcal{A}\{\pi_0[k]\} \right]_{k^2 = M_Z^2}.
\end{aligned} \tag{4.18}
$$

The wave function factors $\sqrt{Z_{ZZ}}$, $\sqrt{Z_{ZA}}$ are obtained from the decomposition of the propagator residue at the pole as in eq. (3.33). These are the same wave functions factors that appear in the standard Feynman rules. Feynman rules for transversely polarized $Z$ particles are the standard ones also in our formalism.

The Feynman rules for the photon are also standard, as we extensively discussed in Section 3.3. It is nevertheless interesting to show explicitly that the general results derived there hold in the context of the SM. We notice that eq. (4.16) implies

$$
\det \widetilde{\Gamma}_{VV} = \Pi^L_{AA}[m_Z^2 + \Pi^L_{ZZ}] - (\Pi^L_{ZA})^2 = 0. \tag{4.19}
$$

Since $\widetilde{\Gamma}_{VV}(0) = \Gamma_\perp(0)$, this means that the transverse propagator possesses an exactly massless photon pole at all orders in perturbation theory. The existence of such pole was one of the conditions we relied on in the study of massless vectors presented in Section 3.3. The second condition we used there (see eq. (3.37)) was that $\widetilde{\Gamma}_{SS} = k^2 + \Pi_{\pi\pi}$ vanishes as $k^2$. This is ensured by the second line of eq. (4.16). We thus confirm that the result of Section 3.3 applies.

### 4.3 $\mathcal{K}_\pi$ at One Loop

We evaluated, using the `FeynArts/FormCalc` package [45], the one-loop expressions of the vacuum polarization amplitudes described in the previous section and we cross-checked the Slavnov–Taylor relations in eq.s (4.10) and (4.16). Notice in particular that the first line of eq. (4.16) implies that $\Pi^L_{AA}$ vanishes at one-loop, compatibly with what we find. From the $\Pi$'s we computed the form factors $[\mathcal{K}_\pi]_{W\pi}$, $[\mathcal{K}_\pi]_{Z\pi}$ and $[\mathcal{K}_\pi]_{A\pi}$ which appear in the Feynman rules in eq.s (4.12) and (4.18).[8] In the Feynman-'t'Hooft gauge and in the $\overline{\text{MS}}$ scheme they are given by

$$
\begin{aligned}
[\mathcal{K}_\pi]_{W\pi}(k^2) &= \frac{m_W}{k}\left( 1 + \frac{g^2}{32\pi^2}(\delta_W + \overline{\delta}) \right), \\
[\mathcal{K}_\pi]_{Z\pi}(k^2) &= \frac{m_Z}{k}\left( 1 + \frac{g^2}{32\pi^2}(\delta_Z + \overline{\delta}) \right), \\
[\mathcal{K}_\pi]_{A\pi}(k^2) &= \frac{m_Z}{k}\left( 0 + \frac{g^2}{32\pi^2}\delta_A \right),
\end{aligned} \tag{4.20}
$$

---

[8]Actually only $[\mathcal{K}_\pi]_{A\pi}$ contributes at the two-loops order in eq. (4.18) because it is multiplied by $\sqrt{Z_{ZA}}$.

where

$$\delta_W = 2(1-c_{\mathrm{w}}^2)B_0\left(k^2,0,m_W^2\right) + \frac{4c_{\mathrm{w}}^4+3c_{\mathrm{w}}^2-1}{2c_{\mathrm{w}}^2}B_0\left(k^2,m_W^2,m_Z^2\right) - \frac{1}{2}B_0\left(k^2,m_h^2,m_W^2\right),$$

$$\delta_Z = \left(4c_{\mathrm{w}}^2-1\right)B_0\left(k^2,m_W^2,m_W^2\right) - \frac{1}{2c_{\mathrm{w}}^2}B_0\left(k^2,m_h^2,m_Z^2\right),$$

$$\delta_A = -4s_{\mathrm{w}}c_{\mathrm{w}}B_0\left(k^2,m_W^2,m_W^2\right),$$

$$\overline{\delta} = \frac{1}{2}\left[1-\log\frac{m_W^2}{\mu^2}\right] + \frac{1}{4c_{\mathrm{w}}^2}\left[1-\log\frac{m_Z^2}{\mu^2}\right] + \frac{3m_h^2}{4m_W^2}\left[1-\log\frac{m_h^2}{\mu^2}\right]$$

$$+ \frac{m_W^2}{m_h^2}\left\{3\left[1-\log\frac{m_W^2}{\mu^2}\right] + \frac{3}{2c_{\mathrm{w}}^4}\left[1-\log\frac{m_Z^2}{\mu^2}\right] - 2\sum_f\frac{m_f^4}{m_W^4}\left[1-\log\frac{m_f^2}{\mu^2}\right] - \frac{1+2c_{\mathrm{w}}^4}{c_{\mathrm{w}}^4}\right\}.$$

(4.21)

In the above equations, $\mu$ is the renormalization scale, $\sum_f = \sum_l + N_c\sum_q$ denotes the sum over all SM leptons ($l$) and quarks ($q$, with $N_c = 3$), and $B_0$ stands for the scalar two-point integral

$$B_0\left(k^2,m_1^2,m_2^2\right) = -\int_0^1 dx\,\log\left(\frac{-x(1-x)k^2+(1-x)m_1^2+xm_2^2}{\mu^2}\right).$$

(4.22)

We note that the most involved term in eqs. (4.20), $\overline{\delta}$, emerges from the Higgs tadpole diagrams, which do not cancel out in the pure $\overline{\mathrm{MS}}$ scheme. In the modified $\overline{\mathrm{MS}}$ scheme where the Higgs tadpole is canceled, $\overline{\delta} = 0$ and the expressions for the $\mathcal{K}_\pi$'s are simpler.

## 4.4 Applications

In this section we apply the Goldstone Equivalence formalism to two simple calculations. The first one, tree-level $WW$ scattering, illustrates the concrete advantages of manifest power-counting. The second one, the $\mathcal{O}(y_t^2)$ radiative corrections to top decay, provides a cross-check of our results beyond the tree-level approximation.

### 4.4.1 Power-Counting in WW Scattering

Consider the tree-level amplitude $\mathcal{M}(W_0^+W_0^- \to W_0^+W_0^-)$ for the scattering of four longitudinal $W$ bosons in the center of mass frame. We are interested in the fully hard kinematical regime where the transverse momentum $k_T$ of the final particles is large, much above the Electroweak scale $m \sim 100$ GeV. This is the configuration where the $W$ bosons energy $E = \sqrt{s}/2$ is much larger than $m$ and the scattering angle $\theta$ is central so that $k_T \sim E \gg m$. In this regime the amplitude $\mathcal{M}$ is well approximated by a power series in $m^2/E^2$.[9] To the second non-trivial order

$$\mathcal{M} = \mathcal{M}_0 + \mathcal{M}_1 + \mathcal{O}\left(\frac{m^4}{E^4}\right),$$

(4.23)

where the amplitude coefficients $\mathcal{M}_0$ and $\mathcal{M}_1$ are of $\mathcal{O}(1)$ and $\mathcal{O}(m^2/E^2)$, respectively.

Computing $\mathcal{M}_0$ and $\mathcal{M}_1$ is not straightforward in the standard formalism because the longitudinal polarization vector (2.34) grows with the vector boson energy as $\varepsilon_\mu^0 \sim E/m_W \sim E/m$. As a consequence, individual non-gauge invariant Feynman diagrams display an unphysical growth with energy that cancels out only when summing them together. In particular the pure gauge diagrams scale as $(E/m)^4$ individually, see Fig. 5. However when the contact diagrams are summed to those with a virtual vector one finds a remarkable cancellation and a milder behavior with the energy $\mathcal{M}_{\mathrm{gauge}} \sim (E/m)^2$. Similarly, diagrams with Higgs exchange grow as

---

[9]No odd powers of $m/E$ can appear because of a spurionic symmetry discussed below.

$$\mathrel{\rlap{\scriptsize \sim}} gE \qquad \mathrel{\rlap{\scriptsize \sim}} gE \qquad \mathrel{\rlap{\scriptsize \sim}} g^2 v \qquad \mathrel{\rlap{\scriptsize \sim}} \lambda v$$

$$\mathrel{\rlap{\scriptsize \sim}} g^2 \qquad \mathrel{\rlap{\scriptsize \sim}} g^2 \qquad \mathrel{\rlap{\scriptsize \sim}} \lambda$$

Figure 5: Coupling and energy power-counting of the vertices relevant to tree level $WW$ scattering. Factors of sines and cosines of the weak angle are treated as numbers of order unity for simplicity. They could be straightforwardly included allowing us, for instance, to exploit the mild hierarchy between $g$ and $g'$.

$\mathcal{M}_{\text{Higgs}} \sim (E/m)^2$. It is only after combining them with the gauge contribution that the final result $\mathcal{M} = \mathcal{M}_{\text{gauge}} + \mathcal{M}_{\text{Higgs}}$ scales like $(E/m)^0$ as it must since no power-like growth with energy is possible in a renormalizable theory such as the SM. These cancellations would make computing $\mathcal{M}_0$ and $\mathcal{M}_1$ from the expansion of Feynman diagrams a painful exercise. Since 2 powers of $(E/m)^2$ will cancel, the gauge diagrams should be Laurent-expanded in $(m/E)^2$ up to the third order (and the Higgs one up to the second order) just to get the leading term $\mathcal{M}_0$. One more order would be needed for $\mathcal{M}_1$. This is as involved as first computing the exact amplitude and subsequently expanding it, obtaining

$$
\begin{aligned}
\mathcal{M}_0 &= -4\lambda + g^2 \frac{1}{4c_{\text{w}}^2}\left(\frac{3+c_\theta^2}{1-c_\theta}\right), \\
\mathcal{M}_1 &= \frac{\lambda}{2}\frac{m_H^2}{E^2}\left(\frac{1+c_\theta}{1-c_\theta}\right) + 2\lambda \frac{m_W^2}{E^2}\left(\frac{1+c_\theta}{1-c_\theta}\right) \\
&+ g^2 \frac{m_W^2}{E^2}\frac{1+c_\theta}{(1-c_\theta)^2}\left[\frac{-9+10c_\theta-5c_\theta^2}{4} + \frac{3-c_\theta}{2c_{\text{w}}^2} + \frac{-6+3c_\theta-c_\theta^2}{16c_{\text{w}}^4}\right], \quad (4.24)
\end{aligned}
$$

where $c_\theta = \cos\theta$. The final result also displays another shortcoming of the standard formalism. The spurious $E/m_W$ factors from the polarization vectors hide the dependence of the final result not only on the energy, but also on the couplings. For example, the term of order $\lambda$ in $\mathcal{M}_0$ does not emerge from any of the vertices involved in the calculation (see Fig. 5). Rather it appears, from a term of order $g^2 m_H^2/m_W^2$, as a reminder of the cancellation between gauge and Higgs diagrams expanding the Higgs propagator for $E \gg m_H$. More generally, the problem is that the negative powers of $m_W \propto g v$ from the polarization vectors can cancel positive powers of $g$ from the vertices and modify the dependence of the final result on the couplings.

The situation is radically different in our Goldstone-Equivalent formalism. The external longitudinal $W$ bosons are represented with double lines as in Figure 1, indicating that amplitudes both with an external vector and with an external Goldstone line should be included for each external $W$ particle. The precise recipe by which these amplitudes have to be combined is provided by eq. (4.12). Goldstone amplitudes are multiplied, since we are at tree-level and the $W$'s are on-shell, by $\pm i[\mathcal{K}_\pi]_{W\pi} = \pm i$, which is constant in energy. Vector amplitudes are multiplied by $e_\mu^0[k]$ (see eq. (2.35)), with $k^2 = m_W^2$, that scales like $m/E$. Clearly our formalism does not bring any advantage if the aim is to compute the exact amplitude $\mathcal{M}$,[10] but it greatly simplifies the calculation in the high-energy limit. Because there are no unphysical energy growths of the polarization vectors, in this formalism one can straightforwardly isolate which contributions are needed at any given order in $m/E$. Furthermore, the dependence on the couplings of each Feynman diagram directly translates into the one of the final result. As

---

[10]But it allows to do so. We cross-checked that it produces the same result as the standard formalism.

Figure 6: Summary of the leading contributions to $\mathcal{M}(W_0 W_0 \to W_0 W_0)$ and their scaling with $m/E$ and $g^2$, $\lambda$. The column $\mathcal{O}(m/E)$ is empty according to the selection rule in eq. (4.25).

seen in Fig. 5, trilinear vertices with one vector and two scalars, or with three vectors, scale like $g\,E$. Vertices with two vectors and a physical Higgs are of order $g\,m_W$ while quartics involving vectors and scalars are of order $g^2$. Quartics with only scalars are instead of order $\lambda$ and scalar trilinears scale like $\lambda\,v$. Finally, the scalar and vector propagators scale like $1/E^2$. Notice that we are working in the Feynman–'t Hooft gauge where there is no mixed scalar/vector propagator.

These simple power-counting rules allow us to identify the diagrams contributing to $\mathcal{M}_0$ and $\mathcal{M}_1$, as schematically reported in Figure 6. Because each $e_\mu^0$ carries a $\sim m_W/E$ suppression, the dominant contribution comes from diagrams with only Goldstones on the external legs, evaluated with massless vector bosons momenta and massless propagators. These consistently match $\mathcal{M}_0$ in eq. (4.24). The terms of order $g^2$ and $\lambda$ directly emerge from the Goldstone-vector vertices and from the Goldstone quartic coupling, respectively. Computing the next order term $\mathcal{M}_1$ is also straightforward. The first term in $\mathcal{M}_1$ comes from the diagrams with two scalar trilinear vertices, of order $(\lambda v)^2/E^2 \sim \lambda m_H^2/E^2$. The second one originates from diagrams with one vector (which comes with one $m_W/E$ factor from $e_\mu^0$) and tree Goldstone external lines, one scalar trilinear and one Goldstone-vector vertex, which is of order $m_W \cdot \lambda\,v \cdot g/E^2 \sim \lambda m_W^2/E^2$. The last one, of order $g^2 m_W^2/E^2$, comes from diagrams with two external vectors and two external Goldstones, plus the contribution of the leading order diagrams with one insertion of the vector boson mass term (denoted by $\otimes$ in the figure) from the expansion of the propagator.

The above discussion also illustrates the connection between our Goldstone-Equivalent formalism and the standard Equivalence Theorem. The Equivalence Theorem, in our formalism, is merely the statement that diagrams with vector external legs are suppressed by $e_\mu^0 \sim m/E$ relative to the Goldstone ones, as in Figure 1. This in itself does not mean that scattering amplitudes for external longitudinal bosons are dominated by Goldstone diagrams, as a naive formulation of the Equivalence Theorem would suggest. Indeed we saw that the suppression from $e_\mu^0$ is only one of the elements of the power-counting rule, to be combined with all the other factors from the vertices and the propagators. These factors were such that the naive Equivalence Theorem holds in our example, therefore $\mathcal{M}_0$ could have also been guessed naively. Clearly there would be no way to obtain $\mathcal{M}_1$ without our formalism. In fact we saw that $\mathcal{M}_1$ emerges also from diagrams with vector external legs. Yet, a naive application of the Equivalence Theorem can produce wrong results even at the leading order in the $m/E$ expansion. Consider for instance the process $W_\pm^+ W_\pm^- \to W_\pm^+ W_0^-$. It so happens (see below) that the amputated Feynman diagrams with one external Goldstone and three vector legs are of order $g^2 m_W/E$. The diagrams with four vector legs are instead of order $g^2$. Taking into account the polarization vectors and their energy scaling, both classes of diagrams contribute to the

leading order scattering amplitude and should be retained.

When studying the high-energy limit of SM scattering amplitudes it is useful to keep in mind the following spurionic symmetry. Consider the $Z_2$ transformation $H \rightarrow -H$ and $\psi_L \rightarrow -\psi_L$, that changes sign to the Higgs and to the fermion doublets. This operation is part of the $SU(2)_L$ gauge group and thus it is a symmetry of the Lagrangian before gauge-fixing. The symmetry acts as $h \rightarrow -h$ and $\pi \rightarrow -\pi$ on the physical Higgs and on the Goldstones, plus the parameter transformation $v \rightarrow -v$. The Higgs VEV $v$ is interpreted as a spurion of the $Z_2$ symmetry. The gauge-fixings in eq. (4.6) further break the symmetry and introduces three more spurions $\widetilde{m}_{W,A,Z} \rightarrow -\widetilde{m}_{W,A,Z}$. If we collectively denote as "$m$" the gauge-fixing masses and the masses of all the (bosonic and fermionic) fields in the theory, and we trade "$v$" for one of these masses, the $Z_2$ symmetry acts as

$$(h, \pi, V_\mu, \psi_L, \psi_R, m) \rightarrow (-h, -\pi, V_\mu, -\psi_L, \psi_R, -m). \qquad (4.25)$$

By this symmetry we can understand better the energy dependence of the $WW$ scattering amplitudes discussed above. The amputated Feynman amplitude with 3 vectors and 1 Goldstone leg involves an odd number of external scalars. Therefore it is $Z_2$-odd and must scale like $(m/E)^{2n+1}$ (with $n \geq 0$, since the theory is renormalizable). In the absence of accidental cancellations the leading term is of order $(m/E)^1$, and this is what is found. Amplitudes with 4 vector legs are instead even and they scale as $(m/E)^0$ barring cancellations. It is also straightforward to draw the implications of the $Z_2$ symmetry directly on the scattering amplitudes, in spite of the fact that our formalism mixes up amputated amplitudes with external vector and Goldstone fields (see Figure 6), which have opposite $Z_2$ parity. Indeed, $e_\mu^0$ is manifestly odd and compensates for the different parities. In particular we can conclude that $\mathcal{M}(W_0 W_0 \rightarrow W_0 W_0)$ is even, therefore its high-energy expansion can only contain even powers of $m/E$ as anticipated above eq. (4.23).

### 4.4.2 Radiative Corrections to Top Decay

At the leading order in the $m_W/m_t \ll 1$ expansion the dominant decay mode of the top is $t \rightarrow W_0^+ b$, with longitudinally polarized $W$. This well-known result is immediately recovered in our formalism (or using the standard Equivalence Theorem) by noticing that the charged Goldstone couples to $t$ and $b$ with strength $y_t$, where $y_t = \sqrt{2} m_t/v$ is the top Yukawa coupling. The coupling with the charged vector is instead the gauge coupling $g$. Since $y_t \gg g$ in the heavy-top limit, the decay to longitudinal $W$ is enhanced relative to the one to transverse by the diagram with the Goldstone on the external leg. In this section we consider radiative corrections to the $\mathcal{M}(t \rightarrow W_0^+ b)$ decay amplitude, focusing in particular on the leading ones, of order $y_t^2/16\pi^2$ (and $y_t^4/\lambda 16\pi^2$, see below) relative to the tree-level. The calculation will be performed in our Goldstone-Equivalent formalism and compared with standard results (see e.g. [46–49]).

Before proceeding, few technical remarks are in order. We compute the proper gauge-invariant decay amplitude, with the momentum of the external top on the complex mass-shell $k_t^2 = M_t^2 \in \mathbb{C}$. This is conceptually important because our formalism is equivalent to the standard one only for gauge-invariant (hence physical) quantities. We should proceed in the same way for the final-state $W$, however the $W$ is stable (i.e., $M_W \in \mathbb{R}$) at the order we are interested in. The $b$ quark is taken massless and stable. We work in the Feynman-'t Hooft gauge and in the $\overline{\text{MS}}$ scheme, but we also show the result in the modified $\overline{\text{MS}}$ scheme discussed at the end of Section 4.1. The anomalously large $\mathcal{O}(y_t^4/\lambda)$ corrections are an artifact of $\overline{\text{MS}}$ due to the Higgs tadpole contribution and they disappear in the modified $\overline{\text{MS}}$ scheme as noticed in Ref. [44]. Calculations are performed with the `Mathematica` package `FeynArts/FormCalc` [45].



Figure 7: One-loop corrections to the top wave-functions to $\mathcal{O}(y_t^2)$.

Let us first summarize the standard calculation. The tree-level diagram, evaluated with all-orders kinematics $k_{t,W}^2 = M_{t,W}^2$, and taking in to account the wave-function factors, gives

$$\sqrt{Z_t^L Z_b^L Z_W} \frac{g}{\sqrt{2}} (\bar{u}_b \gamma^\mu P_L u_t) \bar{\varepsilon}_\mu^0 = \sqrt{Z_t^L Z_b^L Z_W} \frac{g}{\sqrt{2}} \frac{M_t}{M_W} \bar{u}_b P_R u_t, \;\Rightarrow\; \mathcal{M}^{(\text{tree})} = y_t \bar{u}_b P_R u_t, \quad (4.26)$$

with the standard longitudinal polarization vector $\bar{\varepsilon}^0$ as given in eq. (2.34). In the above equation we exploited momentum conservation and the Dirac equation (with $m_b = 0$) for the spinors. Notice that we did not exploit the $m_W/m_t \ll 1$ condition. Namely eq. (4.26), and in particular the resulting tree-level result $\mathcal{M}^{(\text{tree})}$, is exact at all orders in $m_W/m_t$. The one-loop correction to the amplitude, $\mathcal{M}^{(1)}$, receives three kinds of contributions. First we have the corrections to the wave-function factors in the tree diagram, $Z_{t,b}^L = 1 + \delta Z_{t,b}^L$ and $Z_W = 1 + \delta Z_W$. To order $\mathcal{O}(y_t^2, y_t^4/\lambda)$ we have $\delta Z_W = 0$ so the latter will be ignored. Second we have corrections to the masses $M_W^2 = m_W^2 + \delta M_W^2$ and $M_t^2 = m_t^2 + \delta M_t^2$. Finally, we have the genuine one-loop vertex corrections to the amputated amplitude which emerges at this order from Goldstone and Higgs loops. The final result reads

$$\mathcal{M}^{(1)} = \mathcal{M}^{(0)} \left[ 1 + \delta_{\text{vert}} + \frac{1}{2} \left( \delta Z_b^L + \delta Z_t^L \right) + \frac{1}{2} \frac{\delta M_t^2}{m_t^2} - \frac{1}{2} \frac{\delta M_W^2}{m_W^2} \right], \quad (4.27)$$

where the vertex correction $\delta_{\text{vert}}$ is

$$\delta_{\text{vert}} = \frac{g^2 m_t^2}{64\pi^2 m_W^2} \left[ 2B_0(m_t^2, 0, m_t^2) + \log \frac{m_t^2}{\mu^2} - 1 \right], \quad (4.28)$$

with $B_0$ as in eq. (4.22). Notice that the $W$ and the Higgs mass have been neglected compared to $m_t$ in the vertex correction. This is legitimate at $\mathcal{O}(y_t^2)$.

Let us now compute $\mathcal{M}^{(1)}$ with our formalism. A quick inspection of the vertex correction diagrams immediately reveals that there is none contributing to our order. The one with the Goldstone on the external leg and the Goldstone/Higgs trilinear vertex is of order $\lambda$ relative to the tree-level, the one with the Goldstone/Goldstone/vector vertex is of order $g^2$ and the others are even smaller. Diagrams with a vector external leg, such as the one contributing in the standard formalism, are suppressed by the polarization vector factor $e_\mu^0 \sim m_W/E \sim m_W/m_t$. Therefore in the Goldstone-Equivalent formalism there are no vertex corrections and the result entirely comes from the tree-level decay diagrams. Furthermore it so happens that the tree-level diagram with external vector exactly vanishes and we are left with only the Goldstone leg, that gives

$$\mathcal{K}_\pi \sqrt{Z_t^R Z_b^L Z_W} y_t \bar{u}_b P_R u_t, \quad (4.29)$$

with $\mathcal{K}_\pi = [\mathcal{K}_\pi]_{W\pi} = 1 + \delta \mathcal{K}_\pi$ as in eq. (4.20). The standard formalism result for the tree-level amplitude $\mathcal{M}_0$ is immediately recovered. The full one-loop amplitude in the Goldstone Equivalent formalism reads

$$\mathcal{M}_{\text{GE}}^{(1)} = \mathcal{M}^{(0)} \left\{ 1 + \frac{1}{2} \left( \delta Z_b^L + \delta Z_t^R \right) + \delta \mathcal{K}_\pi \right\}, \quad (4.30)$$

which looks different from eq. (4.27) in several respects. We do not have vertex corrections, nor corrections due to the masses. Instead, we have the correction $\delta\mathcal{K}_\pi$ from the Goldstone component of the longitudinal polarization vector. Moreover, we have wave-function $\delta Z_t^R$ corrections to the right-handed top quark field rather than to the left-handed one as in eq. (4.27). This is because the gauge coupling involves the left-handed top, while the Goldstone coupling which is relevant in our formalism involves the right-handed field. We get $\mathcal{M}_{\mathrm{GE}}^{(1)} = \mathcal{M}^{(1)}$ only provided

$$\frac{1}{2}\frac{\delta M_t^2}{m_t^2} \stackrel{?}{=} -\delta_{\mathrm{vert}} + \frac{1}{2}\left(\delta Z_t^R - \delta Z_t^L\right) + \left(\frac{1}{2}\frac{\delta M_W^2}{m_W^2} + \delta\mathcal{K}_\pi\right). \tag{4.31}$$

In order to check eq. (4.31) we use eq. (4.20), duly evaluated at $k_W^2 = M_W^2$, obtaining

$$\begin{aligned}
\delta\mathcal{K}_\pi &= \frac{m_W}{M_W}\left(1 + \frac{g^2}{32\pi^2}(\delta_W + \overline{\delta})\right) - 1 \\
&\simeq -\frac{1}{2}\frac{\delta M_W^2}{m_W^2} + \left(-6\frac{g^2}{32\pi^2}\frac{m_t^4}{m_W^2 m_h^2}\left(1 - \log\frac{m_t^2}{\mu^2}\right)\right),
\end{aligned} \tag{4.32}$$

where in the last line we retained only terms up to $\mathcal{O}(y_t^4/\lambda, y_t^2)$. Those come entirely from $\overline{\delta}$, which in turn originates from the tadpole contribution. The term in brackets on the second line of eq. (4.32) would thus be absent in the modified $\overline{\mathrm{MS}}$ scheme. We also compute explicitly the diagrams in Figure 7, obtaining

$$\delta Z_t^R = \delta Z_t^L - \frac{g^2 m_t^2}{64\pi^2 m_W^2}B_0\left(m_t^2, 0, 0\right), \tag{4.33}$$

at $\mathcal{O}(y_t^4/\lambda, y_t^2)$. Plugging into eq. (4.31), using also eq.s (4.32) and (4.28), we obtain that $\mathcal{M}_{\mathrm{GE}}^{(1)}$ is equal to $\mathcal{M}^{(1)}$ if

$$\begin{aligned}
\frac{1}{2}\frac{\delta M_t^2}{m_t^2} \quad \stackrel{?}{=} \quad & -\frac{g^2 m_t^2}{64\pi^2 m_W^2}\left[2B_0(m_t^2, 0, m_t^2) + \frac{1}{2}B_0\left(m_t^2, 0, 0\right)\log\frac{m_t^2}{\mu^2} - 1\right] \\
& + \left(-6\frac{g^2}{32\pi^2}\frac{m_t^4}{m_W^2 m_h^2}\left(1 - \log\frac{m_t^2}{\mu^2}\right)\right).
\end{aligned} \tag{4.34}$$

This is precisely the relation between the pole and $\overline{\mathrm{MS}}$ top masses at $\mathcal{O}(y_t^2, y_t^4/\lambda)$, given for instance in Ref. [50]. The term in parentheses on the second line, of $\mathcal{O}(y_t^4/\lambda)$, is absent in the modified $\overline{\mathrm{MS}}$ scheme and consistently disappears from the top mass formula as shown in Ref. [44]. Notice that $\mathcal{O}(y_t^4/\lambda)$ corrections also disappear from the decay amplitude in the modified $\overline{\mathrm{MS}}$ scheme. This is because no such term is present (in any scheme) in the wave-function corrections and the one in eq. (4.32) drops.

We have thus confirmed that $\mathcal{M}_{\mathrm{GE}}^{(1)} = \mathcal{M}^{(1)}$ at the order of interest. This constitutes a non-trivial check of our formalism and of the one-loop calculation of $\mathcal{K}_\pi$ in Section 4.3.

# 5 Collinear Factorization and Splitting Functions

We saw that manifest power-counting makes the Goldstone Equivalent formalism simpler and more transparent for explicit calculations of specific processes in the high energy limit. For the sake of proving general properties of the high-energy amplitudes, where manifest power-counting is essential, our formalism is instead not only a simplification, but an absolute need.

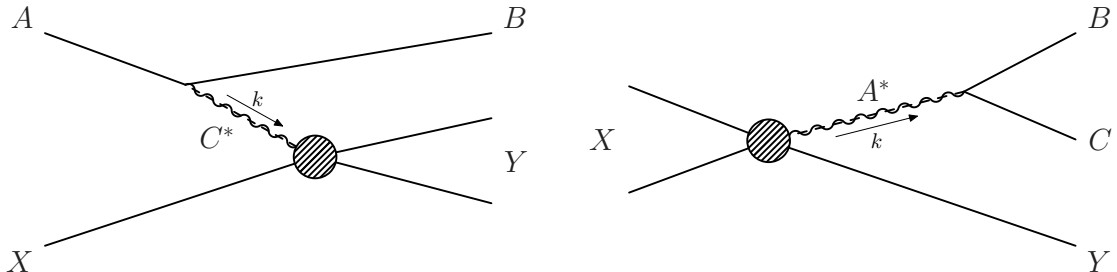

Figure 8: Pictorial representation of the leading contributions in the factorizable $AX \to BY$ and $X \to BCY$ processes. The dashed blob represents the hard reaction.

This point is illustrated in the present Section, where we prove the factorization of collinear splittings at the tree-level order and compute the splitting functions in the SM.

Let us first state collinear factorization for the SM, in the form we will prove it. We start from initial-state splitting topologies depicted on the left panel of Figure 8. Consider a scattering process of the type $AX \to BY$, with $A, B$ two arbitrary particles and $X, Y$ unspecified (multi-particle, in the case of $Y$) states. Assume that there exist a virtual particle $C^*$ that can be emitted from $A$ by the $A \to BC^*$ splitting, and absorbed by $X$ producing $Y$ through the $C^* X \to Y$ reaction. If the hardness "$E$" of the $C^* X \to Y$ process is much larger than the Electroweak scale $m = 100$ GeV and if the $A \to BC^*$ splitting is collinear, up to small corrections "$\delta$" the amplitude factorizes

$$i\mathcal{M}(AX \to BY) = \sum_C i\mathcal{M}^{\mathrm{hard}}(CX \to Y) \frac{i}{Q^2} i\mathcal{M}^{\mathrm{split}}(A \to BC^*)[1 + \mathcal{O}(\delta)], \qquad (5.1)$$

as the product of the matrix element for the "hard" $CX \to Y$ process (with on-shell $C$ particle) times a "splitting amplitude" $\mathcal{M}^{\mathrm{split}}$.

The kinematical regime where eq. (5.1) holds and the size of the corrections will be discussed in detail in the rest of this section (see also Ref. [23]). The corrections are controlled by the expansion parameters $\delta_m \equiv m/E \ll 1$ and $\delta_\perp \equiv |\mathbf{k}_\perp|/E \ll 1$, where $\mathbf{k}_\perp$ denotes the momentum of $B$ transverse to the direction of $A$. The former condition ensures that the characteristic scale of the hard process is indeed much above the Electroweak scale. Adding the latter guarantees that $k^2 = (k_A - k_B)^2 \ll E^2$ is small, such that the $A \to BC^*$ splitting is instead a low-scale process.[11] Since $m_C \lesssim m$, for any SM particle $C$, the two conditions also imply that the virtuality of $C^*$, i.e. $Q^2 \equiv k^2 - m_C^2$ that appears in the denominator of eq. (5.1), is much smaller than $E^2$. Notice that $\delta_m$ and $\delta_\perp$ are independent expansion parameters and their relative magnitude is arbitrary, a priori. We will see that the most interesting configurations are $\delta_m \sim \delta_\perp$ and $\delta_m \ll \delta_\perp$. In the first case the virtuality $Q^2$ is of the order of the Electroweak scale $m$. In the second one, $Q^2 \gg m^2$ and the splitting is itself a high-energy process relative to the Electroweak scale. The relative corrections to the factorized expression for the amplitude in eq. (5.1) are of order $\delta = \mathrm{Max}[\delta_\perp, \delta_m]$ or smaller. The regime where $\delta_m \gg \delta_\perp$ will be discussed later.

Similar considerations hold for final-state splittings, as on the right panel of Figure 8. The process is $X \to BCY$, with $X$ a two-particle and $Y$ a single or multi-particle state. The factorization formula reads

$$i\mathcal{M}(X \to BCY) = \sum_A i\mathcal{M}^{\mathrm{hard}}(X \to AY) \frac{i}{Q^2} i\mathcal{M}^{\mathrm{split}}(A^* \to BC)[1 + \mathcal{O}(\delta)], \qquad (5.2)$$

---

[11]We further need to ensure that the splitting is collinear and not soft. This is achieved by taking the momentum fraction "$x$" (see eq. (5.16)) away from the extremes.

and the expansion parameters are again $\delta_m \equiv m/E$ and $\delta_\perp \equiv |\mathbf{k}_\perp|/E$, where $\mathbf{k}_\perp$ still denotes the momentum of $B$ transverse to $A$. For final-state splittings, unlike the previous case, kinematical configurations exist where $Q^2 \equiv k^2 - m_A^2 = (k_B + k_C)^2 - m_A^2$ is much smaller than $|\mathbf{k}_\perp|^2$ and $m^2$. For instance if $A$ is unstable and $BC$ are its decay products, one might consider the exactly resonant configuration where $Q^2 = 0$. For such resonant configurations, which we exclude from our discussion, the corrections to the factorized approximation are smaller.

The applicability and the implications of the factorization formulas in eq.s (5.1) and (5.2) need to be further clarified. We stated factorization assuming that neither the hard nor the splitting amplitudes are power-like suppressed with energy. Or, better said, that this holds for at least one of the virtual particles $C^*$ and $A^*$ (with given helicities) in the sums. For the hard reaction, the assumption means

$$\mathcal{M}^{\text{hard}} \sim E^{4-L}, \tag{5.3}$$

where "$L$" is the number of external legs so that $4-L$ is the energy dimension of the amplitude. Power suppressions in $m/E$, due for instance to the mass-parity symmetry in eq. (4.25), are excluded. The splitting amplitudes, of energy dimension 1, are instead assumed to scale as $E^0$ and be either of order $|\mathbf{k}_\perp|$ or of order $m$. This is the maximum allowed energy scaling, as it is easy to show by exploiting Lorentz invariance. Which one of the two options $\mathcal{M}^{\text{split}} \sim |\mathbf{k}_\perp|$ or $\mathcal{M}^{\text{split}} \sim m$ is realized is determined by selection rules and confirmed by explicit calculations on a case-by-case basis. Similarly one can show that $\mathcal{M}^{\text{split}}$ may be further suppressed by powers of $m/E$ or $|\mathbf{k}_\perp|/E$ in some splitting configurations, which we are excluding with our assumptions. If the assumptions hold, and if we momentarily restrict to the $\delta_m \sim \delta_\perp$ regime for simplicity, $\mathcal{M}^{\text{split}} \sim \delta_{m,\perp} E$ and the factorized contributions to the amplitudes in eq.s (5.1) and (5.2) scale like

$$\mathcal{M} \sim \frac{1}{\delta_{m,\perp}} E^{3-L}. \tag{5.4}$$

The complete scattering process has one external leg more than the hard reaction, therefore its amplitude has energy dimension $3 - L$ and would scale naively as $E^{3-L}$. The factorized contribution is enhanced by an IR effect (i.e., the collinear splitting) and is larger by a factor of $1/\delta_{m,\perp}$.

If instead the hard or the splitting amplitudes are power-like suppressed in $m/E$ or $|\mathbf{k}_\perp|/E$, eq.s (5.1) and (5.2) should be interpreted with care. They still correctly estimate, as we will see, the magnitude of the contribution of the resonant diagrams to the complete amplitude, allowing us to conclude that no $1/\delta_{m,\perp}$ IR enhancement is present. On the other hand they cannot be used to compute the complete scattering amplitude since non-resonant contributions can be equally important. In short, the factorization formulas in eq.s (5.1) and (5.2) only capture the $1/\delta_{m,\perp}$ IR-enhanced contribution to the amplitude, when present. Similar considerations obviously apply if the hard or the splitting amplitudes are suppressed not by $m/E$, but by small coupling constants. The factorization formulas would not give a good approximation if the complete reaction can be mediated also by different topologies that do not involve splittings but benefit from much larger couplings.

The rest of this section is organized as follows. In Section 5.1 we show how the factorization of the amplitude as in eq.s (5.1) and (5.2) is straightforwardly proved in the Goldstone Equivalent formalism. Manifest power-counting is a key element of the derivation, but the equivalent propagator introduced in Sections 2.3 and 3.3 also plays a major role. In particular it will allow us to deal with splittings involving off-shell massive vectors. In Section 5.2 we describe the calculation of the splitting amplitudes (listed in Appendix B) and of the splitting probabilities. We also apply our results concretely to the emission of a collinear vector $V = W, Z, \gamma$ from the initial state, proving the validity of the so-called "Effective Vector Approximation" (EVA) [51–54].

## 5.1  Amplitude Factorization

Collinear factorization is a statement about contributions to scattering amplitudes that are enhanced, relative to the naive scaling with energy, in the collinear limit. In a formalism like ours where power-counting is manifest at the level of individual Feynman diagrams it is obvious that such enhancements can only emerge from "resonant" diagrams where the real particles involved in the splitting are connected to the rest of the diagram by a single propagator. Namely we can interpret the pictorial representation in Figure 8 of the virtual particles emission/absorption as a quantitative representation of the dominant Feynman diagrams.[12] This is so because enhancements can only emerge from low-virtuality propagators, while in diagrams that are not of the resonant type all the internal lines have high virtuality. The scaling with energy of the non-resonant diagrams contribution to the process ($AX \rightarrow BY$ or $XC \rightarrow BCY$, for initial or final state splitting) is thus the naive one or less

$$\mathcal{M}_{\text{n.r.}} \lesssim E^{3-L} \,, \tag{5.5}$$

where $L$ is the number of legs of the hard scattering subprocess as previously defined. Eq. (5.5) provides an upper bound on the non-resonant contribution, additional suppressions can emerge from the mass-parity selection rule. We will return to this point below.

In order to prove factorization we can thus focus on the resonant Feynman amplitudes and expand them for $\delta_m = m/E \ll 1$ and $\delta_\perp = |\mathbf{k}_\perp|/E \ll 1$. We will work out the expansion explicitly only in the case of vector resonant propagator, which we denote with the habitual double line as in Figure 8. The discussion is fully analogous, but simpler, if the resonant propagator is a fermion or a Higgs propagator. The additional complication in the case of a vector stems from the fact that the standard propagator is not well-behaved in the limit where its 4-momentum components $k_\mu$ are large compared to $k$. This problem was solved in Sections 2.3 and 3.3 by introducing the equivalent propagator $G^{\text{eq}}$, which is well-behaved, and by showing that it can be used in place of the standard propagator in the resonant lines. We showed that the standard propagator can be replaced by $G^{\text{eq}}$ also in multiple resonant lines that arise in the same process. Hence the discussion that follows straightforwardly generalizes to multiple splittings.

We focus for definiteness on the case of a single vector $W$ of mass $m$ and a single Goldstone scalar $\pi$, which we embed as usual in the $\Phi_M = (W_\mu, \pi)$ multiplet. Strictly speaking this only covers the charged vector sector of the SM, however adapting the derivation to the neutral sector poses no additional challenges. The only difference is that the final result for the amplitude will contain a coherent sum over $Z$, photon and (possibly) Higgs intermediate states, producing interference effects to be duly taken into account in the amplitude squared as we will see in Section 5.2.

In the single-vector case, eq. (3.32) (supplemented by the tree-level expression for the form factors in Section 4.2) allows us to write the resonant contribution to the amplitude as

$$
\begin{aligned}
\mathcal{M}_{\text{res}} \;=\;& \mathcal{A}^{\text{A}}\left\{\Phi^M[k]\right\} G^{\text{eq}}_{MN}[k]\mathcal{A}^{\text{C}}\left\{\Phi^N[-k]\right\}, \\[4pt]
\;=\;& \mathcal{A}^{\text{A}}\left\{\Phi^M[k]\right\}\left[\frac{1}{Q^2}\sum_{h=0,\pm}\mathcal{E}^h_M[k]\overline{\mathcal{E}}^h_N[k] + \frac{1}{k^2}\mathcal{P}_{\text{SM}}\mathcal{P}_{\text{SN}}\right]\mathcal{A}^{\text{C}}\left\{\Phi^N[-k]\right\}. \tag{5.6}
\end{aligned}
$$

The propagator momentum $k^\mu$ is oriented to have positive energy component, hence it can be interpreted as the momentum of the virtual vector. The annihilation and creation amplitudes

---

[12]Notice that the dominance of the resonant topology diagrams does not hold for massive gauge theories in the standard covariant formalism, due to the lack of manifest power-counting, as discussed in detail in Ref. [23]. This is the reason why collinear factorization in the SM has been studied until now [21, 23, 55] only in non-covariant (axial) gauges.

$\mathcal{A}^A\{\Phi^M[k]\}$ and $\mathcal{A}^C\{\Phi^M[-k]\}$ correspond to the portions of the resonant Feynman diagrams where the virtual vector is annihilated and created, respectively, as in Figure 2. For the initial-state splitting, $\mathcal{A}^A$ is the amputated amplitude of the hard process $XC \to Y$ (with $C$ the vector) and $\mathcal{A}^C$ corresponds to the $A \to BC^*$ splitting. The interpretation is reversed for final-state splitting. The polarization vectors, at tree-level, are simply

$$
\begin{aligned}
\mathcal{E}_M^0[k] &= \left(e_\mu^0[k], +i\frac{m}{k}\right), \\
\overline{\mathcal{E}}_M^0[k] &= \left(e_\mu^0[k], -i\frac{m}{k}\right),
\end{aligned}
\tag{5.7}
$$

with $e_\mu^0$ as in eq. (2.35).

In the kinematical configuration we are interested in the virtual vector 3-momentum $\vec{k}$ is large, $|\vec{k}| \sim E$, while its virtuality $Q^2$ is small, either of order $|\mathbf{k}_\perp|^2$ or of order $m^2$ depending on which one of the two is larger (see eq. (5.20)). We can thus approximate $k^\mu$ by an on-shell momentum $k_{\text{on}}^\mu$, with $k_{\text{on}}^2 = m^2$. The precise definition of $k_{\text{on}}^\mu$ is ambiguous within the uncertainties introduced by the factorized approximation. We momentarily take $k_{\text{on}}^\mu$ to have the exact same 3-momentum component as $k^\mu$, and the energy component dictated by the on-shell condition. With this choice

$$
\begin{aligned}
\mathcal{E}_M^\pm[k] &= \mathcal{E}_M^\pm[k_{\text{on}}], \\
\mathcal{E}_M^0[k] &= \begin{pmatrix} k/m\left[1+\mathcal{O}(\delta_{m,\perp}^2)\right]\delta_\mu^\nu & \mathbf{0}_{4\times1} \\ \mathbf{0}_{1\times4} & m/k \end{pmatrix}_M^N \mathcal{E}_N^0[k_{\text{on}}],
\end{aligned}
\tag{5.8}
$$

and similarly for the outgoing polarizations. The resonant amplitude thus becomes

$$
\mathcal{M}_{\text{res}} = \mathcal{M}_{\text{res}}^{(\text{pole})}\left[1+\mathcal{O}(\delta_{m,\perp}^2)\right] + \mathcal{M}_{\text{res}}^{(\text{local})}\left[1+\mathcal{O}(\delta_{m,\perp}^2)\right],
\tag{5.9}
$$

where $\mathcal{M}_{\text{res}}^{(\text{pole})}$ and $\mathcal{M}_{\text{res}}^{(\text{local})}$ are

$$
\mathcal{M}_{\text{res}}^{(\text{pole})} = \mathcal{A}^A\{\Phi^M[k]\}\left[\frac{1}{Q^2}\sum_{h=0,\pm}\mathcal{E}_M^h[k_{\text{on}}]\overline{\mathcal{E}}_N^h[k_{\text{on}}]\right]\mathcal{A}^C\{\Phi^N[-k]\},
\tag{5.10}
$$

$$
\mathcal{M}_{\text{res}}^{(\text{local})} = \mathcal{A}^A\{W^\mu[k]\}\left[\frac{1}{m^2}e_\mu^0[k_{\text{on}}]\overline{e}_\nu^0[k_{\text{on}}]\right]\mathcal{A}^C\{W^\nu[-k]\}.
\tag{5.11}
$$

The "pole" term $\mathcal{M}_{\text{res}}^{(\text{pole})}$ is readily seen to produce the factorized expressions in eq.s (5.1) and (5.2) by taking the on-shell limit $k^\mu \to k_{\text{on}}^\mu$ in the amputated amplitude that corresponds to the hard process. This is $\mathcal{A}^A$ in the case of initial-state and $\mathcal{A}^C$ in the case of final-state splitting

$$
\begin{cases}
i\mathcal{M}^{\text{hard}}(XW_h \to Y) = \mathcal{A}^A\{\Phi^M[k_{\text{on}}]\}\mathcal{E}_M^h[k_{\text{on}}] & \text{for initial–state splitting}, \\
i\mathcal{M}^{\text{hard}}(X \to YW_h) = \overline{\mathcal{E}}_M^h[k_{\text{on}}]\mathcal{A}^C\{\Phi^M[-k_{\text{on}}]\} & \text{for final–state splitting}.
\end{cases}
\tag{5.12}
$$

The amplitude that corresponds to the splitting process is instead

$$
\begin{cases}
i\mathcal{M}^{\text{split}}(A \to BW_h^*) = \overline{\mathcal{E}}_N^h[k_{\text{on}}]\mathcal{A}^C\{\Phi^N[-k]\} & \text{for initial–state splitting}, \\
i\mathcal{M}^{\text{split}}(W_h^* \to BC) = \mathcal{A}^A\{\Phi^N[k]\}\mathcal{E}_N^h[k_{\text{on}}] & \text{for final–state splitting}.
\end{cases}
\tag{5.13}
$$

The corrections introduced by the on-shell approximation $k^\mu \to k_{\text{on}}^\mu$ in the hard amplitude, as well as the ones in eq. (5.9), are quadratic in the expansion parameters $\delta_m$ and $\delta_\perp$.[13]

---

[13]When sending $k^\mu \to k_{\text{on}}^\mu$, the other external momenta of the hard process must also be readjusted to ensure energy conservation. This can be done without introducing linear corrections.

They can be safely ignored since comparable or larger (linear) corrections will emerge from other sources. Also notice that the on-shell hard amplitudes are physical gauge-independent quantities and do not necessarily need to be computed in the Goldstone Equivalent formalism. The splitting amplitudes are also gauge-independent, because of the gauge-independence of the complete scattering amplitude. However they are defined and should be computed in our formalism.

Let us now turn to the estimate of the corrections to the factorized formula. They emerge from the non-resonant diagrams contribution $\mathcal{M}_{\text{n.r.}}$ and from the "local" term $\mathcal{M}_{\text{res}}^{(\text{local})}$ in eq. (5.11). The second one happens to be either of the same order or smaller than the first one. In order to see this, we start by giving a slightly more refined estimate of $\mathcal{M}_{\text{n.r.}}$, by exploiting the mass-parity symmetry in eq. (4.25). The complete scattering process $AX \to BY$ or $X \to BCY$ can be even or odd. In the latter case, the symmetry implies that all diagrams contributing to the process, and in particular the resonant ones, are proportional to at least one power of $m$. The non-resonant amplitude thus scales as

$$\mathcal{M}_{\text{n.r.}}^+ \sim E^{3-L}, \quad \mathcal{M}_{\text{n.r.}}^- \sim mE^{2-L}, \tag{5.14}$$

for even and odd amplitudes, respectively. The mass-parity symmetry also tells us about the two amputated amplitudes that appear in the local term $\mathcal{M}_{\text{res}}^{(\text{local})}$. If the complete process is even, the two amplitudes must have the same parity. When they are both even we obtain $\mathcal{M}_{\text{res}}^{(\text{local})} \sim E^{3-L}$ precisely like the non-resonant amplitude $\mathcal{M}_{\text{n.r.}}^+$, since the two powers of $m$ in the denominator of eq. (5.11) cancel the "$m$" factors in $e_\mu^0[k_{\text{on}}]$ (see eq. (2.35)). On the other hand, if both amplitudes are odd, $\mathcal{M}_{\text{res}}^{(\text{local})}$ is further suppressed compared to $\mathcal{M}_{\text{n.r.}}^+$. In the case the complete scattering amplitude is odd under the mass-parity symmetry, the creation and annihilation amplitudes in eq. (5.11) must instead have opposite parity. One of the two brings one power of $m$, therefore $\mathcal{M}_{\text{res}}^{(\text{local})} \sim m E^{2-L}$, again like the non-resonant amplitude $\mathcal{M}_{\text{n.r.}}^-$. We conclude that $\mathcal{M}_{\text{n.r.}}$ provides a conservative estimate of the corrections to factorization, including the effect of the local term.

The corrections are controlled by two separate expansion parameters, $\delta_m$ and $\delta_\perp$, and hierarchies are possible between them. We discuss the various options in turn.

**I) $\delta_m \sim \delta_\perp$**
As already anticipated in eq. (5.4), the factorized component of the amplitude scales like $E^{3-L}/\delta_{m,\perp}$ in this case. This follows from our assumption that power-like energy suppressions are absent in both the hard amplitude, that scales as in eq. (5.3), and in the splitting one, that is of order $\mathcal{M}^{\text{split}} \sim \delta_{m,\perp} E$. The factorized term is thus larger than $\mathcal{M}_{\text{n.r.}}$ by a factor of $1/\delta_{m,\perp}$ if the amplitude is even, and by a factor $1/\delta_{m,\perp}^2$ if it is odd. The relative correction in eq.s (5.1) and (5.2) are thus of order $\delta = \delta_{m,\perp}$ for an even process, $\delta = \delta_{m,\perp}^2$ for an odd one.

**II) $\delta_m \ll \delta_\perp$**
In order to deal with this case one has to notice that the hard scattering amplitude must be even under mass-parity not to experience an energy suppression. This implies that the splitting amplitude has the same parity as the complete scattering process and consequently it is $\mathcal{M}^{\text{split}} \sim m$ if the process is odd and $\mathcal{M}^{\text{split}} \sim |\mathbf{k}_\perp|$ if it is even. The virtuality $Q^2$ in the denominators of eq.s (5.1) and (5.2) is of order $|\mathbf{k}_\perp|^2$, therefore the factorized amplitude is of order $E^{3-L}/\delta_\perp$ if the amplitude is even and of order $m E^{2-L}/\delta_\perp^2$ if it is odd. The corrections to factorizations therefore are $\delta = \delta_\perp$ or $\delta = \delta_\perp^2$ if the amplitude is even or odd, respectively.

**III) $\delta_m \gg \delta_\perp$**
The virtuality is $Q^2 \sim m^2$, therefore for odd amplitudes the resonant component scales like $E^{3-L}/\delta_m$ and the corrections to factorization are $\delta = \delta_m^2$. The situation is different if the

amplitude is even. The splitting amplitude is of order $|\mathbf{k}_\perp|$ and thus the resonant component scales as $E^{3-L}\delta_\perp/\delta_m^2$. The corrections are $\delta = \delta_m^2/\delta_\perp$.

We thus conclude that for any hierarchy between $\delta_m$ and $\delta_\perp$, the corrections in eq.s (5.1) and (5.2) are always quadratic, namely $\delta = \text{Max}[\delta_m^2, \delta_\perp^2]$, if the amplitude is odd. For even amplitudes the final estimate for $\delta$ is instead less favorable and reads

$$\delta = \text{Max}[\delta_m, \delta_\perp, \delta_m^2/\delta_\perp], \tag{5.15}$$

compatibly with what was found in Ref. [23]. The result implies in particular that even if $\delta_m$ and $\delta_\perp$ are small, factorization does not hold when the hierarchy between them is such that $\delta_m^2/\delta_\perp \gtrsim 1$. This is not surprising. Factorization has to capture IR-enhanced contributions to the amplitude, and we just saw that there is no enhancement in this configuration. Notice that this peculiar violation of factorization has limited practical relevance because the kinematical regime where $|\mathbf{k}_\perp|$ is much smaller than $m$ (such that $\delta_\perp/\delta_m \to 0$) is a small region of the phase space where the amplitude is not enhanced.

In the previous discussion we had in mind splittings with a virtual massive vector boson. However our considerations and results hold for an arbitrary splitting, barring the case in which all the particles involved are much lighter than the Electroweak scale $m$. If for instance they are exactly massless, we have that $Q^2 \sim |\mathbf{k}_\perp|^2$ independently of $m$. The corrections due to the on-shell approximation $k^\mu \to k^\mu_{\text{on}}$, which only depend on $Q^2$, are also independent of $m$ and of order $\delta_\perp^2$. The corrections to factorization are thus $\delta = \delta_\perp$ or $\delta = \delta_\perp^2$ for even and odd amplitudes respectively, regardless of the hierarchy between $\delta_\perp$ and $\delta_m$. Factorization for massless splittings holds also in the $E \lesssim m$ regime that we are excluding from our analysis. The standard treatment of factorization for photons, gluons and light quarks and leptons, in QED and QCD, applies in that case.

## 5.2 Splitting Amplitudes and Splitting Functions

The splitting amplitudes may now be evaluated as a straightforward application of eq. (5.13), plus the obvious generalization for the splitting of a virtual fermion or scalar. However few more manipulations and approximations are needed in order to cast the result in a simple and synthetic format. In particular, since factorization only holds in the collinear limit $\delta_{m,\perp} \ll 1$, we are allowed to expand eq. (5.13) in $\delta_{m,\perp}$ and retain only the leading term. We start by considering the splitting of a particle "$A$" moving along the $z$ axis in the positive direction. The 3-momenta of the particles involved in the splitting are parameterized as

$$\begin{aligned}
\vec{k}_A &= \left(0, 0, |\vec{k}_A|\right), \\
\vec{k}_B &= \left(|\mathbf{k}_\perp|\cos\phi, |\mathbf{k}_\perp|\sin\phi, (1-x)|\vec{k}_A|\right), \\
\vec{k}_C &= \left(-|\mathbf{k}_\perp|\cos\phi, -|\mathbf{k}_\perp|\sin\phi, x|\vec{k}_A|\right),
\end{aligned} \tag{5.16}$$

where $|\vec{k}_A|$ is large, of order $E$, and $|\mathbf{k}_\perp| \ll E$. Since we are interested in collinear splittings, and not in soft ones, the longitudinal momentum fraction $x$ ranges from 0 to 1 and it is far from the extremes.[14] Both for initial-state ($A \to BC^*$) and final-state ($A^* \to BC$) splittings, $\mathcal{M}^{\text{split}}$ in eq. (5.13) is given by the 3-point amputated tree-level amplitude of the fields that interpolate for the $A, B, C$ particles, times the corresponding polarization vectors (or spinor wavefunctions) evaluated with on-shell 4-momenta. Notice that the virtual particle momentum $k^\mu$ is carried on-shell (sending $k^\mu \to k^\mu_{\text{on}}$ in the polarization vector as discussed above eq. (5.9)) by preserving its 3-momentum. Therefore if we adopt the same 3-momenta parametrization

---

[14]Namely, $x$ and $1-x$ should not be much smaller than one. This was implicitly assumed in the estimates of Section 5.1.

in eq. (5.16) for the $A \to BC^*$ and for the $A^* \to BC$ splittings, the on-shell momenta of the three particles is the same both for initial- and for final-state splitting amplitudes. The difference between initial- and final-state only emerges from the amputated amplitude, which is evaluated with on-shell $k_{A,B}^\mu$ and off-shell $k_C^\mu = (k_A - k_B)^\mu$, or with on-shell $k_{B,C}^\mu$ and off-shell $k_A^\mu = (k_B + k_C)^\mu$, respectively. However it is not difficult to prove (or to verify by direct calculation) that the result is the same at the leading order in $\delta_{m,\perp}$ and that differences only appear in the second order of the splitting amplitude expansion, i.e. at $\mathcal{O}(\delta_{m,\perp}^2)$. Order $\delta_{m,\perp}^2$ corrections to the factorization formulas (5.1) and (5.2) are present in any case. Therefore we can employ leading-order splitting amplitudes $\mathcal{M}^{\text{split}}(A \to BC)$, that take the same form for initial-state and for final-state splittings, without degrading the accuracy of the approximation.

The complete list of SM splitting amplitudes is reported in Appendix B. Depending on the amount of helicity violation ($\Delta h = h_B + h_C - h_A$) that occurs in the splitting, they take the form

$$
\mathcal{M}^{\text{split}}(A \to BC) = \begin{cases} \sum_p m_p f_{ABC}^{(p)}(x) & \text{for } \Delta h = 0, \\ e^{\mp i\phi} |\mathbf{k}_\perp| f_{ABC}(x) & \text{for } \Delta h = \pm 1, \\ \lesssim \mathcal{O}(|\mathbf{k}_\perp|\delta_\perp) & \text{for } |\Delta h| \geq 2, \end{cases} \tag{5.17}
$$

where the sum on the first line runs over the particles $p = A, B, C$ involved in the splitting. Splitting amplitudes with $\Delta h = 0$ are independent of $\phi$ and of $\mathbf{k}_\perp$. They are proportional to the masses $m_{A,B,C}$ and they vanish in the massless case. Splittings of this type, dubbed "ultra-collinear" in Ref. [21], are peculiar of the SM. They give rise to interesting phenomena such as the emission of a longitudinal vector boson from a massless fermion. Notice that the $\Delta h = 0$ splittings, since their amplitude is proportional to the masses, are odd under the mass-parity symmetry. For $\Delta h = \pm 1$ we recover instead the structure of the standard massless QED and QCD splittings. The dependence of the splitting amplitudes on $\phi$ and on $\mathbf{k}_\perp$ is dictated by rotational symmetry, as we will briefly review in Appendix B. In particular rotational symmetry implies that $|\Delta h| \geq 2$ amplitudes are proportional to at least two powers of $\mathbf{k}_\perp$, hence they are at most of $\mathcal{O}(|\mathbf{k}_\perp|\delta_\perp)$. They are suppressed with energy and thus can be ignored as we discussed.

We now turn to the generic configuration where the particle "$A$" moves in an arbitrary direction. Denoting as $\Theta \in [0, \pi]$ and $\Phi \in [0, 2\pi)$ the polar and azimuthal angles of $\vec{k}_A$ (as in eq. (A.1)), we can define a standard "Jacob–Wick" rotation

$$
R_{\text{JW}}(\Theta, \Phi) \equiv R_{\text{Eul.}}(\Phi, \Theta, -\Phi) = e^{-i\Phi J_z} e^{-i\Theta J_y} e^{+i\Phi J_z}, \tag{5.18}
$$

where $R_{\text{Eul.}}(\alpha, \beta, \gamma)$ denotes the generic Euler rotation. The inverse of $R_{\text{JW}}$ brings $\vec{k}_A$ along the positive $z$ axis, therefore we can parametrize the momenta as the $R_{\text{JW}}$ rotation acting on eq. (5.16). Namely, we define the variables $\phi$, $|\mathbf{k}_\perp|$, and $x$ that characterize the splitting as

$$
\begin{aligned}
\vec{k}_A &= R_{\text{JW}}(\Theta, \Phi) \cdot \left(0, 0, |\vec{k}_A|\right), \\
\vec{k}_B &= R_{\text{JW}}(\Theta, \Phi) \cdot \left(|\mathbf{k}_\perp| \cos\phi, |\mathbf{k}_\perp| \sin\phi, (1-x)|\vec{k}_A|\right), \\
\vec{k}_C &= R_{\text{JW}}(\Theta, \Phi) \cdot \left(-|\mathbf{k}_\perp| \cos\phi, -|\mathbf{k}_\perp| \sin\phi, x|\vec{k}_A|\right).
\end{aligned} \tag{5.19}
$$

Geometrically, $\phi - \Phi$ is the angle between the oriented plane formed by the $z$ axis and $\vec{k}_A$, and the plane of the splitting oriented from $\vec{k}_A$ to $\vec{k}_B$. Of course $|\mathbf{k}_\perp|$ and $x$ are nothing but the transverse momentum and the longitudinal momentum fraction of $\vec{k}_C$ relative to $\vec{k}_A$, respectively. With these definitions the splitting amplitudes are identical in form to the ones (previously discussed and reported in Appendix B) obtained for the kinematical configuration in eq. (5.16) only up to phase factors. However we will show in the Appendix that these phases do not play any role and can be safely ignored in the discussion that follows.

It is important to remark that unlike the ordinary splitting functions, the explicit form of the splitting amplitudes does depend on the conventions adopted for the polarization vectors and the spinor wave-functions. Once one convention is chosen for the splitting amplitudes, the exact same one must be employed in the evaluation of the hard scattering amplitude in order for eq.s (5.1) and (5.2) to apply. Our conventions follow from the original Jacob–Wick [56] definition of helicity eigenstates and are reported in Appendix A.

**Splitting Functions**

The factorization formulas for the amplitudes in eq.s (5.1) and (5.2) contain all the information about the complete scattering processes $AX \to BY$ or $X \to BCY$ in the collinear limit. By employing the approximate (to $\mathcal{O}(\delta_{\perp,m}^2)$) expressions [15]

$$Q^2 = \begin{cases} (k_A - k_B)^2 - m_C^2 = -\dfrac{1}{1-x}\widetilde{k}_\perp^2 & \text{for initial–state splitting } (A \to BC^*), \\[2mm] (k_B + k_C)^2 - m_A^2 = +\dfrac{1}{x(1-x)}\widetilde{k}_\perp^2 & \text{for final–state splitting } (A^* \to BC), \end{cases} \tag{5.20}$$

for the virtuality $Q^2$, by squaring the amplitude and multiplying it by the appropriate phase-space factors, one easily derives factorized expressions for the fully-differential scattering cross-sections. The factorised amplitude is in general the sum of the contribution of several virtual particles with different helicities. The resulting factorized cross-section thus contains interference terms and must be expressed (see e.g. [57–59]) in the language of density matrices as

$$d\sigma = \text{Tr}[d\rho^{\text{split}} \cdot d\rho^{\text{hard}}], \tag{5.21}$$

where the trace runs over the possible virtual intermediate particles species and helicities. The splitting density matrix $d\rho^{\text{split}}$, differential in the variables $\phi$, $|\mathbf{k}_\perp|$ and $x$ that characterise the splitting is the generalization, to include interference effects, of the ordinary splitting functions. The differential information on the hard process is encapsulated in the hard density matrix $d\rho^{\text{hard}}$. Explicitly, for the initial-state splitting process $AX \to BY$ we find

$$d\rho^{\text{split}}_{C_h C'_{h'}} = \frac{x(1-x)}{16\pi^2} \frac{\mathcal{M}^{\text{split}}(A \to BC_h)[\mathcal{M}^{\text{split}}(A \to BC'_{h'})]^*}{\widetilde{k}_\perp^2(m_A, m_B, m_C)\ \widetilde{k}_\perp^2(m_A, m_B, m_{C'})} dx\, d|\mathbf{k}_\perp|^2 \frac{d\phi}{2\pi},$$

$$d\rho^{\text{hard}}_{C_h C'_{h'}} = \frac{\mathcal{M}^{\text{hard}}(C_h X \to Y)[\mathcal{M}^{\text{hard}}(C'_{h'} X \to Y)]^*}{4E_X E_C |\mathbf{v}_C - \mathbf{v}_X|} d\Phi_Y, \tag{5.22}$$

where $d\Phi_Y$ is the phase-space factor of the hard final state $Y$, including $(2\pi)^4$ times the energy-momentum conservation delta function. For final-state splitting processes $X \to BCY$, instead

$$d\rho^{\text{split}}_{A_h A'_{h'}} = \frac{x(1-x)}{16\pi^2} \frac{\mathcal{M}^{\text{split}}(A_h \to BC)[\mathcal{M}^{\text{split}}(A'_{h'} \to BC)]^*}{\widetilde{k}_\perp^2(m_A, m_B, m_C)\ \widetilde{k}_\perp^2(m_{A'}, m_B, m_C)} dx\, d|\mathbf{k}_\perp|^2 \frac{d\phi}{2\pi},$$

$$d\rho^{\text{hard}}_{A_h A'_{h'}} = \frac{\mathcal{M}^{\text{hard}}(X \to A_h Y)[\mathcal{M}^{\text{hard}}(X \to A'_{h'} Y)]^*}{4E_{X_1} E_{X_2} |\mathbf{v}_{X_1} - \mathbf{v}_{X_2}|} d\Phi_{AY}. \tag{5.23}$$

The derivations that lead to eq.s (5.22) and (5.23) are straightforward and need not be reported here. The only aspect that requires clarification is related with the on-shell momentum $k_{\text{on}}$ for the "C" particle in the initial-state splitting $A \to BC^*$. In Section 5.1 we took the on-shell limit by preserving the virtual particle 3-momentum. Namely the spatial components of

---

[15]We define $\widetilde{k}_\perp^2(m_A, m_B, m_C) = |\mathbf{k}_\perp|^2 - x(1-x)m_A^2 + x m_B^2 + (1-x)m_C^2$.

the momentum $k_{on}$ the hard amplitude is evaluated on (see eq. (5.12)) should match the exact splitting kinematics in eq. (5.19). Since $\vec{k}_C$ depends on $\phi$ and $|\mathbf{k}_\perp|$ in our parametrisation, this introduces an inconvenient dependence of the hard amplitude on the details of the splitting kinematics. However we saw that amplitude factorisation only holds up $\mathcal{O}(\delta_\perp)$ corrections, barring special circumstances where the corrections are smaller. Up to corrections of the same order we can further approximate $k_{C,on}$ by taking the collinear limit [16]

$$k_{C,on}^\mu \;\rightarrow\; k_{C,coll}^\mu = \left\{ \sqrt{x^2|\vec{k}_A|^2 + m_C^2},\, R_{JW}(\Theta,\Phi)\cdot\left(0,0,x|\vec{k}_A|\right) \right\}. \tag{5.24}$$

The hard amplitude evaluated on $k_{C,coll}$, and in turn $d\rho^{split}$ in eq. (5.22), is now independent of $\phi$ and $|\mathbf{k}_\perp|$ and it only depends on the momentum fraction $x$ of the particle that participates to the hard scattering. For final state splittings instead, $d\rho^{split}$ is completely independent of the splitting variables in our parametrization.

Since the hard component of eq. (5.21) is independent of $\phi$ and $|\mathbf{k}_\perp|$, the factorized cross-section inclusive on these variables can be expressed in terms of an integrated splitting density matrix. It is customary to integrate at least over $\phi$ because the azimuthal structure of the radiation is often of limited phenomenological importance and because the density matrix becomes diagonal in the helicity after the $\phi$ integration. This latter property, namely the cancellation of the interference between the contributions of intermediate particles of different helicity upon $\phi$ integration, follows from the dependence of the splitting amplitudes on $\phi$ as in eq. (5.17). In QED (and in QCD, after summing over color), the splitting density matrix collapses to a single number (one for each intermediate particle helicity) after $\phi$ integration because no interference is possible between particles of different species. One can thus abandon the density matrix formalism and state the result in terms of ordinary splitting functions, to be interpreted as splitting probabilities or as parton distribution functions. In the SM instead, for neutral vector bosons splittings, the interference between the $Z$ and the photon persists and the density matrix formalism is needed even after the integral over $\phi$. In some cases, e.g. for the emission of a collinear top-anti-top pair, interference with the Higgs boson exchange should also be taken into account.

A curious fact about $\phi$-integrated collinear splittings is that the corrections to factorization are smaller than for the fully-differential cross-section. In the latter, corrections of $\mathcal{O}(\delta_{\perp,m})$ are generically present. However it can be shown that the linear $\mathcal{O}(\delta_{\perp,m})$ corrections are canceled by the integration, and one is left with $\mathcal{O}(\delta_{\perp,m}^2)$. This fact was pointed out in Ref. [23] in the context of the Effective $W$ Approximation, but the result is of general validity and applies to arbitrary splitting configurations.

**Application: Effective Vector Approximation**

Before concluding, we apply our general results to the proof of the validity of the Effective Vector Approximation (EVA) formula [23,54,55], namely the collinear approximation for the emission of a charged or neutral collinear vector boson from a massless fermion in the initial state. The relevant splitting amplitudes are found in Appendix B and read

$$\mathcal{M}^{split}(f_{L/R} \to f'_{L/R} V_h) = C_V(f_{L/R}) \times \mathcal{S}_{L/R}^{(h)}, \tag{5.25}$$

---

[16]Initial-state splitting often emerge from an incoming particle $A$ moving along the $z$ axis, for which the azimuthal angle $\phi$ is conventionally set to zero. The Jacob–Wick rotation $R_{JW}$ in the equation that follows is then equal to plus or minus the identity when $A$ is parallel or anti-parallel to $z$, respectively.

where $L$ and $R$ denote the fermion helicity and we defined

$$
\mathcal{S}_L^{(h)} = \begin{cases} +\sqrt{2}|\mathbf{k}_\perp|e^{-i\phi}\dfrac{\sqrt{1-x}}{x} & \text{for } h=+, \\[2mm] -\sqrt{2}|\mathbf{k}_\perp|e^{+i\phi}\dfrac{1}{x\sqrt{1-x}} & \text{for } h=-, \\[2mm] -2m_V\dfrac{\sqrt{1-x}}{x} & \text{for } h=0, \end{cases} \qquad \mathcal{S}_R^{(h)} = \begin{cases} +\sqrt{2}|\mathbf{k}_\perp|e^{+i\phi}\dfrac{1}{x\sqrt{1-x}} & \text{for } h=+, \\[2mm] -\sqrt{2}|\mathbf{k}_\perp|e^{-i\phi}\dfrac{\sqrt{1-x}}{x} & \text{for } h=-, \\[2mm] -2m_V\dfrac{\sqrt{1-x}}{x} & \text{for } h=0. \end{cases}
\tag{5.26}
$$

The splitting amplitude is proportional to the vector-fermion gauge coupling $C_V$

$$
C_V(f_L) = \begin{cases} \dfrac{g}{2\sqrt{2}}V_{ff'} & \text{for } V=W^\pm, \\[2mm] q_f e & \text{for } V=\gamma, \\[2mm] \dfrac{g}{c_w}\left(T_f^3 - s_w^2 q_f\right) & \text{for } V=Z, \end{cases} \qquad C_V(f_R) = \begin{cases} 0 & \text{for } V=W^\pm, \\[2mm] q_f e & \text{for } V=\gamma, \\[2mm] -g\dfrac{s_w^2}{c_w}q_f & \text{for } V=Z, \end{cases}
\tag{5.27}
$$

where $q_f$ and $T_f^3$ denote, respectively, the electric charge and the value of the third $\mathrm{SU(2)}_L$ generator for the fermion. The appropriate element of the CKM matrix, in the case of splitting from quarks, is denoted as $V_{ff'}$.

When the splitting is charged, i.e. $f \neq f'$, only the charged $V = W^\pm$ vector boson can mediate the reaction. After integrating over $\phi$ the density matrix thus reduces to the splitting functions

$$
\frac{d\rho_{h=\pm1,0}^{\text{split}}}{dx\,d|\mathbf{k}_\perp|^2} = C_W^2 \left|\mathcal{S}_L^{(h)}\right|^2 \frac{x(1-x)}{16\pi^2 \tilde{k}_\perp^4},
\tag{5.28}
$$

where $\tilde{k}_\perp^2 = \mathbf{k}_\perp^2 + (1-x)m_W^2$. Upon integrating over $|\mathbf{k}_\perp|$, these expressions can be interpreted as the probability to find a $W$ of a given helicity and energy fraction inside the fermion. When the splitting is neutral, i.e. $f = f'$, both $V = Z$ and $V = \gamma$ can be exchanged. We thus obtain a non-diagonal density matrix (here, $\tilde{k}_\perp^2 = \mathbf{k}_\perp^2 + (1-x)m_Z^2$)

$$
\frac{d\rho_{h=\pm1}^{\text{split}}}{dx\,d|\mathbf{k}_\perp|^2} = \begin{pmatrix} C_\gamma^2 & C_\gamma C_Z \dfrac{|\mathbf{k}_\perp|^2}{\tilde{k}_\perp^2} \\[4mm] C_Z C_\gamma \dfrac{|\mathbf{k}_\perp|^2}{\tilde{k}_\perp^2} & C_Z^2 \dfrac{|\mathbf{k}_\perp|^4}{\tilde{k}_\perp^4} \end{pmatrix} \left|\mathcal{S}_{L/R}^{(h)}\right|^2 \frac{x(1-x)}{16\pi^2 |\mathbf{k}_\perp|^4},
\tag{5.29}
$$

when the intermediate vector boson helicity has $h = \pm1$. If the intermediate vector boson is longitudinal, only the $Z$ contributes and we obtain

$$
\frac{d\rho_{h=0}^{\text{split}}}{dx\,d|\mathbf{k}_\perp|^2} = C_Z^2 \left|\mathcal{S}_{L/R}^{(0)}\right|^2 \frac{x(1-x)}{16\pi^2 \tilde{k}_\perp^4}.
\tag{5.30}
$$

## 6 Conclusions and Outlook

In this paper we formalized the notion of "Goldstone Equivalence" and we started exploring its implications in the study of high-energy Electroweak physics. Namely we upgraded the Goldstone Boson Equivalence Theorem to a formalism in which energy and couplings power-counting is manifest at the level of individual Feynman diagrams, and we outlined its possible applications in two distinct directions. The first direction, more pragmatic, is to simplify explicit calculations in the high-energy regime. The second direction, more conceptual, is to

establish general properties of the high-energy cross-sections related with factorization. Let us discuss them in turn.

Manifest power-counting allows to isolate the (manifestly gauge-invariant) combination of Feynman diagrams that is relevant at a given order in the energy and in the couplings expansion. This was illustrated in Section 4.4.1, for tree-level $WW$ scattering, and in Section 4.4.2, where we computed $\mathcal{O}(y_t^2/16\pi^2)$ corrections to the top decay amplitude. Notice that energy and couplings (i.e., in particular, loop) expansions can be carried out independently in our formalism. Indeed in our example we could include the exact tree-level amplitude, to all orders in the $m_W/m_t$ expansion, while only retaining the first order in the one-loop contribution. Another advantage of our formalism is that the relevant diagrams can be computed, at the leading order in the energy expansion, with massless internal line propagators. Higher order terms can be included by treating the mass as a perturbation. Since massless integrals are often easier to compute than massive ones, this could be a crucial advantage for calculations at very high order. One caveat in this program is that the massless limit should be taken with care in diagrams affected by IR divergences (or enhancements, if the divergence is regulated by the finite mass of the vector bosons). One must first isolate and subtract the IR singularities and next take the massless limit. Subtracting IR singularities is a standard problem in QED and QCD calculations, therefore we expect that the issue could be addressed by the powerful techniques developed in those contexts. However the more general structure of the Electroweak vertices compared with the ones of QED and QCD might pose additional challenges.

Manifest energy power-counting is essential to understand the structure of Feynman diagrams in the presence of multiple largely separated energy scales. Our formalism thus finds a natural application to the study of the Electroweak IR problem. We outlined this aspect in Section 5, where we proved collinear factorization at the tree-level order. While the study of tree-level factorization (which includes in particular the Effective Vector Approximation) is of practical interest, it is definitely of limited scope in the context of the general IR problem. However the two key aspects that appear in the derivation are general properties of our formalism that could be useful also in more ambitious problems. The first one is once again that manifest power-counting allows to isolate the relevant diagram topologies. In the collinear limit those are the splitting topologies enhanced by a low-virtuality "nearly-resonant" propagator. The second aspect is that the resonant propagator can be cast in an equivalent form which is well-behaved in the limit of high energy and finite virtuality. One can thus identify the nearly on-shell degrees of freedom and take the on-shell limit smoothly. Manifest power-counting and well-behaved propagators are all-order properties of our formalism, which could be used to extend the study of factorization beyond the tree-level. A reasonable first step in this direction would probably be to include the soft region and the one-loop corrections to derive fixed-order $\alpha \log$ and $\alpha \log^2$ results. It remains to be seen whether and how our formalism can contribute addressing the problem of IR logs resummation.

# Acknowledgements

We thank T. Han, A. Hebbar, K. Mimouni, L. Ricci, R.Rattazzi and R. Torre for useful discussions.

**Funding information** We acknowledge partial support from the Swiss National Science Foundation under contract 200021-178999. The work of G. C. is partially supported by the Swiss National Science Foundation under contract 200020-169696 and through the National Center of Competence in Research SwissMAP. L. V. is supported by the Swiss National Science Foundation under the Sinergia network CRSII2-16081.

# A  Single-Particle States and Wave Functions

We define particles in the helicity basis following Ref. [56]. One-particle states of mass $m$, helicity $h$, and 3-momentum

$$\vec{k} = \left(k_x, k_y, k_y\right) = |\vec{k}|\left(\sin\Theta\cos\Phi, \sin\Theta\sin\Phi, \cos\Theta\right), \tag{A.1}$$

are obtained acting on a reference state $|\vec{k}_{\mathrm{ref}}, h\rangle$ (to be specified below) with the standard Lorentz transformation $\Lambda_{\vec{k}} = R_{\mathrm{JW}}(\Theta, \Phi)\, e^{+i\eta K_z}$

$$|\vec{k}, h\rangle \equiv U(\Lambda_{\vec{k}})|\vec{k}_{\mathrm{ref}}, h\rangle, \tag{A.2}$$

where $R_{\mathrm{JW}}(\Theta, \Phi)$ is defined in eq. (5.18), and $K_z$ is the generator of boosts along the $z$ axis. The reference state $|\vec{k}_{\mathrm{ref}}, h\rangle$ and the value of the rapidity $\eta$ in eq. (A.2) depend on whether the associated particle is massless or massive. If $m = 0$ the reference state has 3-momentum $\vec{k}_{\mathrm{ref}} = +|\vec{k}_{\mathrm{ref}}|\vec{z}$ along the positive $z$ direction and definite helicity $h = J_z$. The standard Lorentz transformation $\Lambda_{\vec{k}}$ must be such that $k^\mu = [\Lambda_{\vec{k}} k_{\mathrm{ref}}]^\mu$, which requires $\ln\eta = |\vec{k}|/|\vec{k}_{\mathrm{ref}}|$. When $m \neq 0$ the reference state $|\vec{k}_{\mathrm{ref}}, h\rangle$ has vanishing 3-momentum, $k^\mu_{\mathrm{ref}} = (m, \vec{0})$, and again $h = J_z$. In the massive case one finds $\tanh\eta = |\vec{k}|/\sqrt{\vec{k}^2 + m^2}$. Eq. (A.2) uniquely defines all states within the domain $\Theta \in [0, \pi)$ and $\Phi \in [0, 2\pi)$. However, particles moving along the negative $z$ axis, i.e. those with $\Theta = \pi$, are only defined up to a phase because their azimuthal angle $\Phi$ (appearing in the standard rotation in eq. (A.2)) is not uniquely determined. We conventionally set $\Phi = 0$ at $\Theta = \pi$, which is equivalent to define the state as

$$
\begin{aligned}
|-|\vec{k}|\vec{z}, h\rangle &\equiv U(R_{\mathrm{JW}}(\pi, 0)\, e^{+i\eta K_z})|\vec{k}_{\mathrm{ref}}, h\rangle \\
&= U(R_{\mathrm{JW}}(\pi, 0))|+|\vec{k}|\vec{z}, h\rangle.
\end{aligned} \tag{A.3}
$$

From the above definitions we can determine the polarization vectors for spin-1 particles and the spinor wave functions completely, up to an irrelevant constant phase. Consider a spin-1 particle of mass $m$, helicity $h$, 3-momentum as in eq. (A.1) and energy $k_0 = \sqrt{\vec{k}^2 + m^2}$. The polarization vectors for $h = +1, -1, 0$ read

$$
\begin{aligned}
\varepsilon^+_\mu[k] &= \frac{1}{\sqrt{2}}\frac{1}{|\vec{k}|(|\vec{k}| + k_z)}
\begin{pmatrix} 0 \\ |\vec{k}|(|\vec{k}| + k_z) - k_x(k_x + ik_y) \\ i|\vec{k}|(|\vec{k}| + k_z) - k_y(k_x + ik_y) \\ -(|\vec{k}| + k_z)(k_x + ik_y) \end{pmatrix}
= -\frac{e^{i\Phi}}{\sqrt{2}}
\begin{pmatrix} 0 \\ -\cos\Theta\cos\Phi + i\sin\Phi \\ -i\cos\Phi - \cos\Theta\sin\Phi \\ \sin\Theta \end{pmatrix}, \\
\varepsilon^-_\mu[k] &= \frac{1}{\sqrt{2}}\frac{1}{|\vec{k}|(|\vec{k}| + k_z)}
\begin{pmatrix} 0 \\ -|\vec{k}|(|\vec{k}| + k_z) + k_x(k_x - ik_y) \\ i|\vec{k}|(|\vec{k}| + k_z) + k_y(k_x - ik_y) \\ (|\vec{k}| + k_z)(k_x - ik_y) \end{pmatrix}
= \frac{e^{-i\Phi}}{\sqrt{2}}
\begin{pmatrix} 0 \\ -\cos\Theta\cos\Phi - i\sin\Phi \\ i\cos\Phi - \cos\Theta\sin\Phi \\ \sin\Theta \end{pmatrix}, \\
\varepsilon^0_\mu[k] &= \frac{k_0}{m}
\begin{pmatrix} \frac{|\vec{k}|}{k_0} \\ -\frac{k_x}{|\vec{k}|} \\ -\frac{k_y}{|\vec{k}|} \\ -\frac{k_z}{|\vec{k}|} \end{pmatrix}
= \frac{k_0}{m}
\begin{pmatrix} \frac{|\vec{k}|}{k_0} \\ -\sin\Theta\cos\Phi \\ -\sin\Theta\sin\Phi \\ -\cos\Theta \end{pmatrix}.
\end{aligned} \tag{A.4}
$$

As dictated by eq. (A.3), the polarization vectors for particles in the backward limit $\vec{k} \to (0, 0, -|\vec{k}|)$ are defined taking $k_x \to 0^+$ with $k_y/k_x \to 0$, and of course $k_z \to -|\vec{k}|$. With this prescription the expressions of the polarization vectors $\varepsilon^h_\mu[k]$ are regular and single-valued. While the polarization vectors are defined for physical particles with real momentum, the

above definitions of $\varepsilon_\mu^h[k]$ can be extended to complex $k$ momentum by analytic continuation (taking $|\vec{k}| = \sqrt{\vec{k}^2}$).

It is useful to introduce the "conjugate" polarizations $\overline{\varepsilon}_\mu^h[k]$, which appear in the matrix elements with final-state external vectors as well as the completeness relation (3.29). For arbitrary (complex) momenta they are defined as

$$\overline{\varepsilon}_\mu^h[k] \equiv (-1)^h \varepsilon_\mu^{-h}[k]. \tag{A.5}$$

Note that for real momenta $\overline{\varepsilon}$ is the complex conjugate of $\varepsilon$.

Dirac spinors for particles or anti-particles of helicity $h = \pm 1/2$ are given by

$$u_h[k] = \begin{pmatrix} \omega_{-h}[k]\,\chi_h(\vec{k}) \\ \omega_h[k]\,\chi_h(\vec{k}) \end{pmatrix}, \qquad v_h[k] = \begin{pmatrix} 2h\omega_h[k]\,\chi_{-h}(\vec{k}) \\ -2h\omega_{-h}[k]\,\chi_{-h}(\vec{k}) \end{pmatrix}, \tag{A.6}$$

where $\omega_h[k] = \sqrt{k_0 + 2h|\vec{k}|}$ and

$$\chi_{1/2}(\vec{k}) = \frac{1}{\left[2|\vec{k}|\left(|\vec{k}| + k_z\right)\right]^{1/2}} \begin{pmatrix} |\vec{k}| + k_z \\ k_x + ik_y \end{pmatrix} = \begin{pmatrix} \cos\Theta/2 \\ e^{i\Phi}\sin\Theta/2 \end{pmatrix}, \tag{A.7}$$

$$\chi_{-1/2}(\vec{k}) = \frac{1}{\left[2|\vec{k}|\left(|\vec{k}| + k_z\right)\right]^{1/2}} \begin{pmatrix} -k_x + ik_y \\ |\vec{k}| + k_z \end{pmatrix} = \begin{pmatrix} -e^{-i\Phi}\sin\Theta/2 \\ \cos\Theta/2 \end{pmatrix}. \tag{A.8}$$

Similarly to the polarizations for vector particles, the wavefunction for an $h = \pm 1/2$ state with $|\vec{k}| + k_z \to 0^+$ is unambiguously obtained taking the limit $k_x \to 0^+$ with $k_y/k_x \to 0$.

The "conjugate" spinors are defined as

$$\begin{aligned} \overline{u}_h[k] &= v_h^t[k](i\gamma^0\gamma^2) \\ \overline{v}_h[k] &= u_h^t[k](i\gamma^0\gamma^2), \end{aligned} \tag{A.9}$$

where the $\gamma^\mu$ matrices are understood to be in the Weyl representation. The completeness are the standard ones, namely $\sum_h u_h[k]\overline{u}_h[k] = \slashed{k} + \sqrt{k^2}$ and $\sum_h v_h[k]\overline{v}_h[k] = \slashed{k} - \sqrt{k^2}$, for arbitrary complex $k^\mu$. Notice that for real momentum the "conjugate" spinors in eq. (A.9) reduce to the standard $\overline{u}_h = u_h^\dagger\gamma^0$ and $\overline{v}_h = v_h^\dagger\gamma^0$.

# B  Splitting in the Standard Model

In this appendix we derive explicit expressions for the splitting amplitudes defined in eq. (5.13). These depend on the definition of single particle states, which themselves determine the form of the polarization vectors and spinor wave functions as in Appendix A. We also discuss a few key properties of the splitting amplitudes and derive the identities (B.1) and (B.2) that may be used by the reader to calculate the splitting amplitudes we do not report explicitly.

### Properties of the Splitting Amplitudes

We begin discussing the collinear splitting amplitudes $\mathcal{M}^{\text{split}}(A \to BC)$ defined in eq. (5.13) for the standard 3-kinematics specified in eq. (5.16), where $A$ moves exactly in the positive direction of the $z$ axis. Despite not being S-matrix elements, $\mathcal{M}^{\text{split}}(A \to BC)$ transform under all symmetries as physical amplitudes. Their structure is in fact determined by dimensional analysis and angular momentum considerations. From Lorentz invariance follows that these quantities must be proportional to the soft scales $|\mathbf{k}_\perp|, m_{A,B,C}$, up to negligible corrections $\mathcal{O}(\delta_{m,\perp}^2)$.

Conservation of the angular momentum further fixes the dependence on the azimuthal angle $\phi$ introduced in eq. (5.16). Indeed, in the helicity basis defined in eq. (A.2) one finds that under rotations $R_z(\phi') = \exp[-i\phi' J_z]$ around the $z$ axis the 1-particle states transform with a simple phase factor

$$U(R_z(\phi'))|\vec{k}, h\rangle = e^{-i\phi' h}|R_z(\phi')\vec{k}, h\rangle\,,$$

for arbitrary momentum $\vec{k}$. Invariance under rotations then implies

$$\mathcal{M}^{\text{split}}(R_z(\phi')\mathbf{k}_\perp) = e^{-i\phi'\Delta h}\mathcal{M}^{\text{split}}(\mathbf{k}_\perp)\,,$$

with $\Delta h = h_B + h_C - h_A$ the total change in helicity. The latter condition is solved considering the projections of $\mathbf{k}_\perp$ onto the eigenvectors of $J_z$, namely $\mathbf{k}_\perp^1 \pm i\mathbf{k}_\perp^2 = e^{\pm i\phi}|\mathbf{k}_\perp|$. Under $R_z(\phi')$, $e^{\pm i\phi}|\mathbf{k}_\perp| \to e^{\pm i(\phi+\phi')}|\mathbf{k}_\perp|$, and since $\mathcal{M}^{\text{split}}$ is analytic in $\mathbf{k}_\perp$ we conclude that it must have the structure

$$\mathcal{M}^{\text{split}}(A \to BC) \propto |\mathbf{k}_\perp|^{|\Delta h|}e^{-i\phi\Delta h}\,,$$

as shown in eq. (5.17). For $\Delta h = 0$ there is no dependence on $|\mathbf{k}_\perp|$ and the amplitude is proportional to the masses by dimensional analysis.

The $\mathcal{M}^{\text{split}}$'s satisfy two additional useful relations. First, from eq. (5.16) it follows that

$$\mathcal{M}^{\text{split}}(A_{h_A} \to C_{h_C} B_{h_B}) = \mathcal{M}^{\text{split}}(A_{h_A} \to B_{h_B} C_{h_C})\big|_{\phi \to \phi+\pi,\ x \to 1-x,\ m_B \to m_C,\ m_C \to m_B}\,. \qquad (B.1)$$

Moreover, the accidental CP invariance of the tree-level amplitudes introduces another important constraint. Actually, rather than using parity (P) itself, defined as the inversion of the 3 spacial coordinates, it is more convenient to consider the reflection with respect to the $xz$ plane, i.e. the inversion of the $y$ coordinate only. This operation corresponds to the combined action of P and a $\pi$-rotation along the $y$-axis, i.e. $P_y = R_{\text{JW}}(\pi, 0) \cdot P$. The $CP_y$ operator acts on a state $|\vec{k}, h, A\rangle$ describing a particle $A$ of momentum $\vec{k}$ and helicity $h$ as $CP_y|\vec{k}, h, A\rangle = (-1)^{j-h}\bar{\eta}|\mathcal{P}_y\vec{k}, -h, \overline{A}\rangle$ whereas as $CP_y|\vec{k}, h, \overline{A}\rangle = (-1)^{j-h}\bar{\eta}^*|\mathcal{P}_y\vec{k}, -h, A\rangle$ on anti-particles, where $\bar{\eta}$ are phases (or unitary matrices when different particle species can mix) appearing in the field transformations. [17] In the SM the Higgs and the fermionic phases may be absorbed in the Yukawa couplings using chiral rotations and hyper-charge transformations so that we can set $\bar{\eta}_\phi = \bar{\eta}_\psi = 1$. Also, for the vectors $\bar{\eta}_V = 1$. Since $\mathcal{P}_y\vec{k} = |\vec{k}|(\sin\Theta\cos\Phi, -\sin\Theta\sin\Phi, \cos\Theta)$ is equivalent to an inversion of the azimuthal angle, we obtain

$$\mathcal{M}^{\text{split}}(\overline{A}_{-h_A} \to \overline{B}_{-h_B}\overline{C}_{-h_C}) = \prod_{k=A,B,C}(-1)^{j_k-h_k}\mathcal{M}^{\text{split}}(A_{h_A} \to B_{h_B}C_{h_C})\big|_{\phi \to -\phi}\,. \qquad (B.2)$$

**Splitting in an arbitrary direction**

In this subsection we demonstrate that the very same $\mathcal{M}^{\text{split}}(A \to BC)$ obtained with the standard 3-momentum given in eq. (5.16) can be employed for the calculation of the factorized amplitudes in eq.s (5.1) and (5.2)) even if $A$ moves along an arbitrary direction. We parametrize the general splitting kinematic configuration by rotating the standard $\vec{k}_{A,B,C}^{\text{std}}$ in eq. (5.16) with a common matrix

$$\vec{k}_{A,B,C} = R_{\text{JW}}(\Theta_A, \Phi_A)\vec{k}_{A,B,C}^{\text{std}}\,, \qquad (B.3)$$

---

[17]The scalar $\phi(x)$, Dirac fermion $\psi(x)$, and vector $V_\mu(x) = V_\mu^a(x)T^a$ transform respectively as $(CP_y)\phi(x)(CP_y)^\dagger = \bar{\eta}_\phi^*\phi^\dagger(x')$, $(CP_y)\psi(x)(CP_y)^\dagger = \bar{\eta}_\psi^*\gamma^5\psi^\dagger(x')$, $(CP_y)V_\mu(x)(CP_y)^\dagger = -(\mathcal{P}_y)_\mu^\nu V_\nu^*(x')$, with $x' = \mathcal{P}_y x$ and $\mathcal{P}_y = \text{diag}(1, 1, -1, 1)$.

as in eq. (5.19). The action of a Jacob-Wick rotation on one-particle states reads

$$
\begin{aligned}
U(R_{\rm JW}(\Theta_A,\Phi_A))|\vec{k}_A^{\rm std},h_A\rangle &= |\vec{k}_A,h_A\rangle, \\
U(R_{\rm JW}(\Theta_A,\Phi_A))|\vec{k}_B^{\rm std},h_B\rangle &= e^{i\Psi_B}|\vec{k}_B,h_B\rangle, \\
U(R_{\rm JW}(\Theta_A,\Phi_A))|\vec{k}_C^{\rm std},h_C\rangle &= e^{i\Psi_C}|\vec{k}_C,h_C\rangle.
\end{aligned}
\tag{B.4}
$$

There is no phase associated to $A$ because, according to the definition in eq. (A.2), the $R_{\rm JW}(\Theta_A,\Phi_A)$ rotation acting on a state $A$ moving in the positive $z$ direction precisely generates a state with rotated 3-momentum. The phases show up in the splitting amplitudes, which are related to those evaluated with the standard momenta (5.16) by

$$
\mathcal{M}^{\rm split}(A_{\vec{k}_A} \to B_{\vec{k}_B} C_{\vec{k}_C}) = e^{i(\Psi_B+\Psi_C)}\mathcal{M}^{\rm split}(A_{\vec{k}_A^{\rm std}} \to B_{\vec{k}_B^{\rm std}} C_{\vec{k}_C^{\rm std}}).
\tag{B.5}
$$

The phases in eq. (B.5) that are associated to exactly on-shell particles are obviously unphysical because they disappear from the squared amplitude. Only the phase of the virtual state can potentially be relevant in the calculation of the factorized amplitude. In the case of splitting in the final state $A^* \to BC$ the only relevant phase would thus be the one associated to $A$, but this vanishes by construction. As a result, in the analysis of an arbitrary (single and multiple) final state splitting we can safely use in eq. (5.2) the splitting functions $\mathcal{M}^{\rm split}(A \to BC)$ calculated with the standard 3-kinematics specified in eq. (5.16).

Consider next an initial-state splitting $A \to BC^*$. Here $B$ is on-shell, and so $e^{i\Psi_B}$ is again unphysical, but $e^{i\Psi_C}$ can play a role. The case of a single splitting in the initial state, when $A$ moves exactly along the positive $z$ axis, corresponds to the reference kinematics. However if multiple splittings occur, some of the initial state particles are slightly tilted from the $z$ axis. These are thus associated to initial state splittings in which the corresponding $A$ state is rotated by a small $\Theta_A$, far from $\Theta_A = \pi$. An explicit computation shows that $\Psi_C = \mathcal{O}(\delta_\perp)$ in such a situation. This guarantees that, when considering multiple splittings from an initial state moving along the positive $z$ axis, the phase in eq. (B.5) at most affects the subleading term in eq. (5.1).

Because of the ambiguity in the definition of backward-moving particles (see the discussion around eq. (A.3)), the situation is a bit more involved when the initial state splitting takes place from an original particle $A$ moving opposite to the $z$ axis. Here we consider a $\vec{k}_A$ that is nearly but not exactly parallel to the negative $z$ axis (i.e., $\pi - \Theta_A \lesssim \mathcal{O}(\delta_\perp)$). However the discussion also covers the case $\Theta_A = \pi$ (for which $\Phi_A = 0$ by convention). The phase $e^{i\Psi_C}$ in eq. (B.5) becomes of order in unity in this case, and superficially might invalidate our claim. Fortunately, though, the $\mathcal{O}(1)$ contribution to $\Psi_C$ gets compensated by an analogous and opposite phase showing up in the hard process when taking the collinear limit, leaving in the end a negligible correction of $\mathcal{O}(\delta_\perp)$ to the factorized amplitude, similarly to the previous case. To see this recall that in eq. (5.1) the splitting amplitude is multiplied by the hard matrix element $\mathcal{M}^{\rm hard}(CX \to Y)$ calculated for an on-shell $C$ moving along $\vec{k}_C = R_{\rm JW}(\Theta_A,\Phi_A)\vec{k}_C^{\rm std}$, which is not exactly parallel to $\vec{k}_A$. Even if $\vec{k}_C$ gets parallel to $\vec{k}_A$ in the collinear limit (i.e., $\vec{k}_C \to \vec{k}_{C,{\rm coll}} = \vec{k}_A |\vec{k}_C|/|\vec{k}_A|$ as in eq. (5.24)), the state that describes the $C$ particle approaches the backward-moving state defined in eq. (A.3) only up to a phase. Correspondingly the $C$-particle wave function and in turn the hard matrix element approaches the one computed for exactly collinear $C$ only up to a phase.

In order to show that the latter phase cancels the $e^{i\Psi_C}$ factor in the splitting amplitude, let us reabsorb the $e^{i\Psi_C}$ phase into the definition of a non-standard state

$$
|\vec{k}_C,h_C\rangle^\Psi \equiv e^{i\Psi_C}|\vec{k}_C,h_C\rangle = U(R_{\rm JW}(\Theta_A,\Phi_A))|\vec{k}_C^{\rm std},h_C\rangle,
\tag{B.6}
$$

using eq. (B.4). For this state, obviously, the large $\Psi_C$ phase appears in the hard matrix element

$$
\mathcal{M}^{\rm hard}(C_{\vec{k}_C}^\Psi X \to Y) = e^{i\Psi_C}\mathcal{M}^{\rm hard}(C_{\vec{k}_C} X \to Y).
\tag{B.7}
$$

It turns out the non-standard state $|\vec{k}_C, h_C\rangle^\Psi$ smoothly approaches the conventional Jacob–Wick state for collinear momentum $\vec{k}_{C,\text{coll}}$ in the limit $\delta_\perp \to 0$. Indeed, the standard state $|\vec{k}_C^{\text{std}}, h_C\rangle$ approaches $||\vec{k}_C|\hat{z}, h_C\rangle$ without phases and thus

$$\lim_{\delta_\perp \to 0} |\vec{k}_C, h_C\rangle^\Psi = U(R_{\text{JW}}(\Theta_A, \Phi_A))||\vec{k}_C|\hat{z}, h_C\rangle = |\vec{k}_{C,\text{coll}}, h_C\rangle . \tag{B.8}$$

Correspondingly, the wave function and in turn the hard amplitude (B.7) for the non-standard state smoothly approaches the one evaluated with collinear $C$ without extra phases.

## Splitting Amplitudes for a General Gauge Theory

We now present the Feynman rules relevant for the evaluation of the splitting functions in the tree approximation, namely those associated to 3-particle vertices. We parametrize the couplings in terms of generic functions $C_{ABC}$ so that our results can be straightforwardly applied to general renormalizable gauge theories. An explicit expression for $C_{ABC}$ is presented in the case of the SM. We subsequently collect all the independent splitting amplitudes arising from the Feynman rules. The unlisted amplitudes can be obtained from (B.1) and (B.2).

### Relevant Feynman Rules

- Fermionic Vertices

The interaction vertices between two fermions $f^a, f^b$ of masses $m_{a,b}$ and a vector $V$ or a scalar $h$ are defined below in terms of general couplings $C_{L,R}$, $y_f$, and $P_L = (1 - \gamma_5)/2$, $P_R = (1 + \gamma_5)/2$ denoting the chirality projectors.

In the SM the Yukawa coupling $y_f$ is diagonal in fermion flavor and related to the fermion mass $m_f = m_a = m_b$ through the Higgs VEV as $y_f = \sqrt{2}m_f/v$, whereas the parameters $C_{L,R}$ are collected in this table

| $V$ | $W^-$ ($f^a = d, f^b = u$) | $Z$ ($f^a = f^b$) | $\gamma$ ($f^a = f^b$) | $G$ ($f^a = f^b$) |
|---|---|---|---|---|
| $C_L$ | $\frac{g V_{ud}}{\sqrt{2}}$ | $\frac{g}{c_w}(T^3 - s_w^2 q_f)$ | $e q_f$ | $g_s t^\alpha_{i_b i_a}$ |
| $C_R$ | $0$ | $\frac{g}{c_w}(-s_w^2 q_f)$ | $e q_f$ | $g_s t^\alpha_{i_b i_a}$ |

Here $V_{ud}$ is an element of the CKM matrix, $T^A$ are the weak SU(2) and $t^\alpha$ the color SU(3) generators in the fundamental representation, $e = g s_w$ ($g_s$) is the QED (QCD) coupling and finally $q_f$ is the electric charge.

The Feynman rule for the coupling between two fermions and a Goldstone boson $\pi$, when present, may be derived from the $f^a f^b V$ vertex using the tree-level version of the generalized Ward identity in eq. (3.23), $ik^\mu \mathcal{A}\{V_\mu[k]\} = -m_V \mathcal{A}\{\pi[k]\}$, and the Dirac equation. Note that $m_V$ is the mass of the vector associated to the Goldstone boson.

$$V_\mu \;=\; i\gamma_\mu(C_L P_L + C_R P_R)$$

$$\pi \;=\; \frac{m_b}{m_V}(C_L P_L + C_R P_R) - \frac{m_a}{m_V}(C_L P_R + C_R P_L)$$

$$h \;=\; -i\frac{y_f}{\sqrt{2}}$$

- Bosonic Scalar Vertices

We now move to purely bosonic vertices involving at least one scalar particle. The Feynman rule for the cubic scalar coupling is defined as $-i6\lambda v$. In the SM $\lambda$ is the quartic Higgs coupling and $v$ its VEV. In the case of general scalar theories one just replaces $6\lambda v$ with the appropriate trilinear. Vertices with two scalars and one Goldstone are forbidden. On the other hand, vertices of the type $h\pi^a\pi^b$ are related to the $hV^aV^b$ vertex via (3.23).

Renormalizable $hV^aV^b$ vertices, with a scalar boson and two massive vectors, are parametrized in general in terms of a dimensionful coupling $C\sqrt{m_{V_a}m_{V_b}}$. In the Standard model this is given by:

| $V^a V^b$ | $W^- W^+$ | $ZZ$ | $\gamma\gamma$ | $GG$ |
|---|---|---|---|---|
| $C\sqrt{m_a m_b}$ | $gm_W$ | $\frac{g}{c_{\mathrm{w}}}m_Z$ | $0$ | $0$ |

Similarly to the fermionic couplings to Goldstone bosons and $h\pi^a\pi^b$, the vertex $h\pi^a V^b$ written below can be shown to be related to $hV^aV^b$ via the generalized Ward identity.

A vertex with two scalars and a transverse vector should also be included in general, though it is absent in the SM. Without loss of generality we assume the very same Feynman rule as $h\pi^a V^b$, and the associated splitting amplitudes are shown at the very end of our list with $\pi \to h'$.

$$h, h \to h \quad = -i\frac{3}{2}g\frac{m_h^2}{m_W}$$

$$V_\mu^a, V_\nu^b \to h \quad = iC\sqrt{m_a m_b}\,\eta_{\mu\nu}$$

$$\pi_a (k_2), V_\nu^b (k_1) \to h \quad = \frac{C}{2}\sqrt{\frac{m_b}{m_a}}(k_2 - k_1)_\nu$$

$$\pi_a, \pi_b \to h \quad = -iC\frac{m_h^2}{2\sqrt{m_a m_b}}$$

- Vector and Goldstone Vertices

Finally, we present the Feynman rules for three vector and Goldstone bosons. The most general (renormalizable) vertex involving three vectors depends on a coupling $C_{abc}$ fully antisymmetric in the three indices $a, b, c$. In the SM

| $V^a V^b V^c$ | $W^- W^+ \gamma$ | $W^- W^+ Z$ | $G^\alpha G^\beta G^\gamma$ |
|---|---|---|---|
| $C_{abc}$ | $e$ | $g c_{\rm w}$ | $-ig_s f_{\alpha\beta\gamma}$ |

with $f_{\alpha\beta\gamma}$ denoting the SU(3) structure constants. The corresponding vertices with one or two Goldstone bosons, when present, are related to the three-vector vertex via the generalized Ward identity in eq. (3.23). No three-Goldstone vertex can arise from spontaneous breaking. This can also be explicitly confirmed using eq. (3.23).

$$V_\rho^c = iC_{abc}\left[\eta_{\mu\nu}(k_a - k_b)_\rho + \eta_{\nu\rho}(k_b - k_c)_\mu + \eta_{\rho\mu}(k_c - k_a)_\nu\right]$$

$$\pi_c = C_{abc}\eta_{\mu\nu}\frac{m_b^2 - m_a^2}{m_c}$$

$$V_\mu^c = -\frac{i}{2}C_{abc}\frac{m_a^2 + m_b^2 - m_c^2}{m_a m_b}(k_a - k_b)_\mu$$

**Splitting Amplitudes**

- Fermions and Vectors

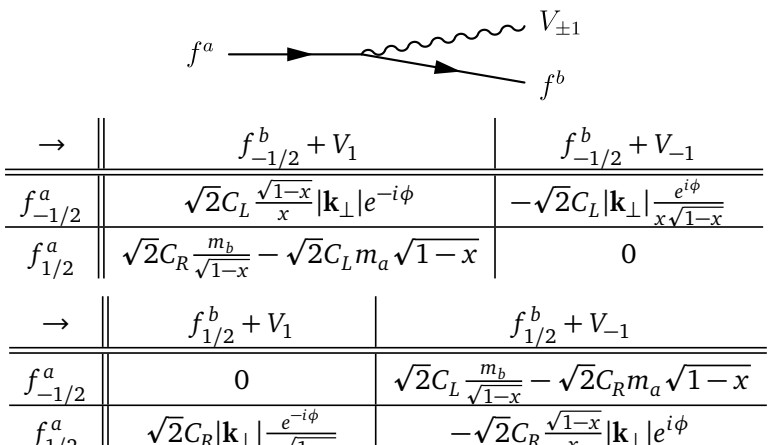

| $\rightarrow$ | $f_{-1/2}^b + V_1$ | $f_{-1/2}^b + V_{-1}$ |
|---|---|---|
| $f_{-1/2}^a$ | $\sqrt{2}C_L\frac{\sqrt{1-x}}{x}\|\mathbf{k}_\perp\|e^{-i\phi}$ | $-\sqrt{2}C_L\|\mathbf{k}_\perp\|\frac{e^{i\phi}}{x\sqrt{1-x}}$ |
| $f_{1/2}^a$ | $\sqrt{2}C_R\frac{m_b}{\sqrt{1-x}} - \sqrt{2}C_L m_a\sqrt{1-x}$ | $0$ |

| $\rightarrow$ | $f_{1/2}^b + V_1$ | $f_{1/2}^b + V_{-1}$ |
|---|---|---|
| $f_{-1/2}^a$ | $0$ | $\sqrt{2}C_L\frac{m_b}{\sqrt{1-x}} - \sqrt{2}C_R m_a\sqrt{1-x}$ |
| $f_{1/2}^a$ | $\sqrt{2}C_R\|\mathbf{k}_\perp\|\frac{e^{-i\phi}}{x\sqrt{1-x}}$ | $-\sqrt{2}C_R\frac{\sqrt{1-x}}{x}\|\mathbf{k}_\perp\|e^{i\phi}$ |

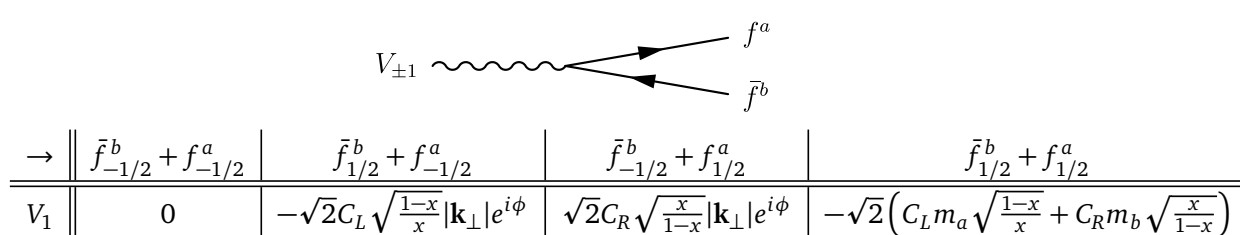

| $\rightarrow$ | $\bar{f}_{-1/2}^b + f_{-1/2}^a$ | $\bar{f}_{1/2}^b + f_{-1/2}^a$ | $\bar{f}_{-1/2}^b + f_{1/2}^a$ | $\bar{f}_{1/2}^b + f_{1/2}^a$ |
|---|---|---|---|---|
| $V_1$ | $0$ | $-\sqrt{2}C_L\sqrt{\frac{1-x}{x}}\|\mathbf{k}_\perp\|e^{i\phi}$ | $\sqrt{2}C_R\sqrt{\frac{x}{1-x}}\|\mathbf{k}_\perp\|e^{i\phi}$ | $-\sqrt{2}\left(C_L m_a\sqrt{\frac{1-x}{x}} + C_R m_b\sqrt{\frac{x}{1-x}}\right)$ |

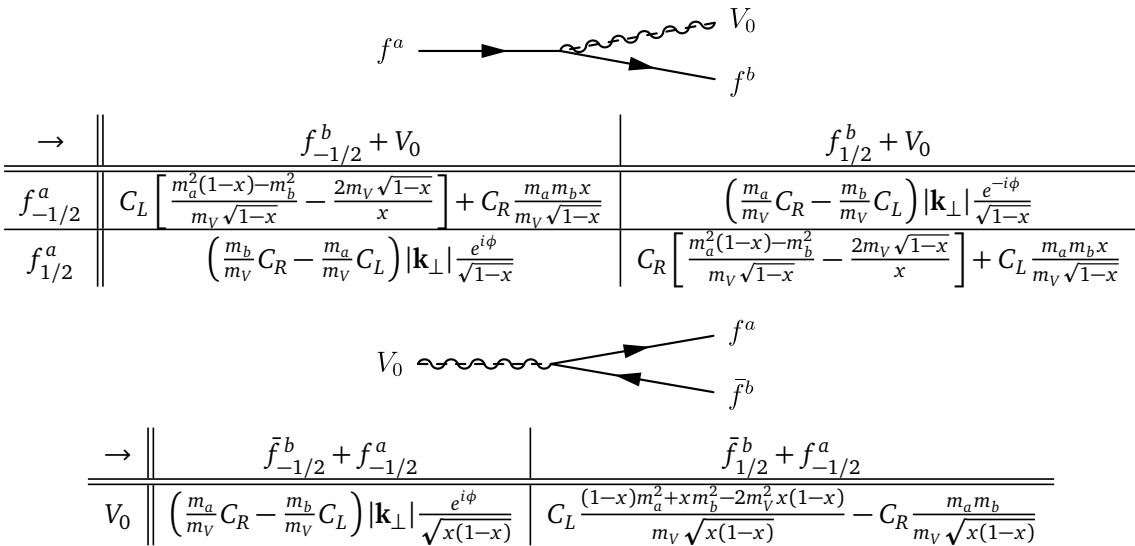

| → | $f^b_{-1/2} + V_0$ | $f^b_{1/2} + V_0$ |
|---|---|---|
| $f^a_{-1/2}$ | $C_L\left[\frac{m_a^2(1-x)-m_b^2}{m_V\sqrt{1-x}} - \frac{2m_V\sqrt{1-x}}{x}\right] + C_R\frac{m_a m_b x}{m_V\sqrt{1-x}}$ | $\left(\frac{m_a}{m_V}C_R - \frac{m_b}{m_V}C_L\right)|\mathbf{k}_\perp|\frac{e^{-i\phi}}{\sqrt{1-x}}$ |
| $f^a_{1/2}$ | $\left(\frac{m_b}{m_V}C_R - \frac{m_a}{m_V}C_L\right)|\mathbf{k}_\perp|\frac{e^{i\phi}}{\sqrt{1-x}}$ | $C_R\left[\frac{m_a^2(1-x)-m_b^2}{m_V\sqrt{1-x}} - \frac{2m_V\sqrt{1-x}}{x}\right] + C_L\frac{m_a m_b x}{m_V\sqrt{1-x}}$ |

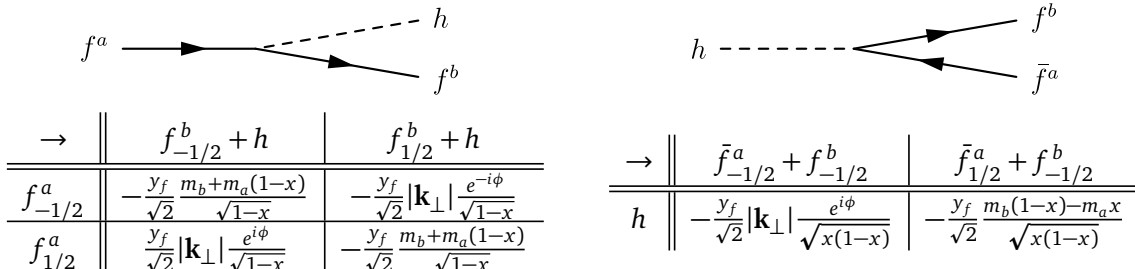

| → | $\bar{f}^b_{-1/2} + f^a_{-1/2}$ | $\bar{f}^b_{1/2} + f^a_{-1/2}$ |
|---|---|---|
| $V_0$ | $\left(\frac{m_a}{m_V}C_R - \frac{m_b}{m_V}C_L\right)|\mathbf{k}_\perp|\frac{e^{i\phi}}{\sqrt{x(1-x)}}$ | $C_L\frac{(1-x)m_a^2+xm_b^2-2m_V^2 x(1-x)}{m_V\sqrt{x(1-x)}} - C_R\frac{m_a m_b}{m_V\sqrt{x(1-x)}}$ |

- Fermions and Scalars

| → | $f^b_{-1/2} + h$ | $f^b_{1/2} + h$ |
|---|---|---|
| $f^a_{-1/2}$ | $-\frac{y_f}{\sqrt{2}}\frac{m_b+m_a(1-x)}{\sqrt{1-x}}$ | $-\frac{y_f}{\sqrt{2}}|\mathbf{k}_\perp|\frac{e^{-i\phi}}{\sqrt{1-x}}$ |
| $f^a_{1/2}$ | $\frac{y_f}{\sqrt{2}}|\mathbf{k}_\perp|\frac{e^{i\phi}}{\sqrt{1-x}}$ | $-\frac{y_f}{\sqrt{2}}\frac{m_b+m_a(1-x)}{\sqrt{1-x}}$ |

| → | $\bar{f}^a_{-1/2} + f^b_{-1/2}$ | $\bar{f}^a_{1/2} + f^b_{-1/2}$ |
|---|---|---|
| $h$ | $-\frac{y_f}{\sqrt{2}}|\mathbf{k}_\perp|\frac{e^{i\phi}}{\sqrt{x(1-x)}}$ | $-\frac{y_f}{\sqrt{2}}\frac{m_b(1-x)-m_a x}{\sqrt{x(1-x)}}$ |

- Triple Scalar and Scalar-Vector Splittings

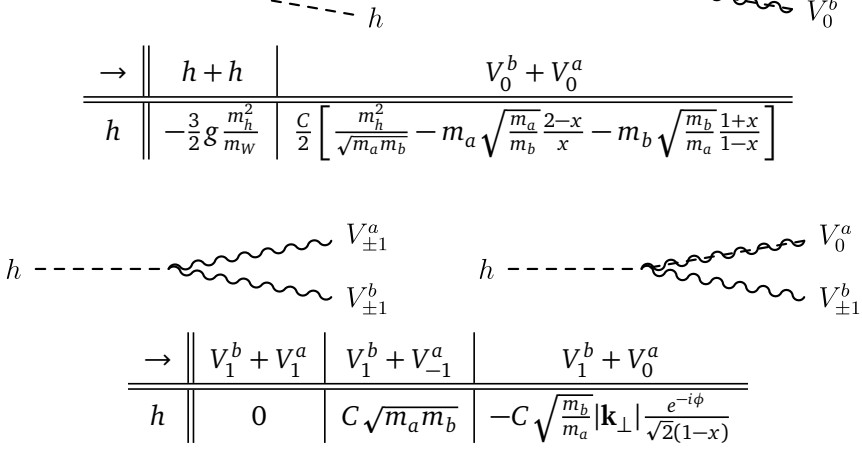

| → | $h+h$ | $V_0^b + V_0^a$ |
|---|---|---|
| $h$ | $-\frac{3}{2}g\frac{m_h^2}{m_W}$ | $\frac{C}{2}\left[\frac{m_h^2}{\sqrt{m_a m_b}} - m_a\sqrt{\frac{m_a}{m_b}}\frac{2-x}{x} - m_b\sqrt{\frac{m_b}{m_a}}\frac{1+x}{1-x}\right]$ |

| → | $V_1^b + V_1^a$ | $V_1^b + V_{-1}^a$ | $V_1^b + V_0^a$ |
|---|---|---|---|
| $h$ | $0$ | $C\sqrt{m_a m_b}$ | $-C\sqrt{\frac{m_b}{m_a}}|\mathbf{k}_\perp|\frac{e^{-i\phi}}{\sqrt{2}(1-x)}$ |

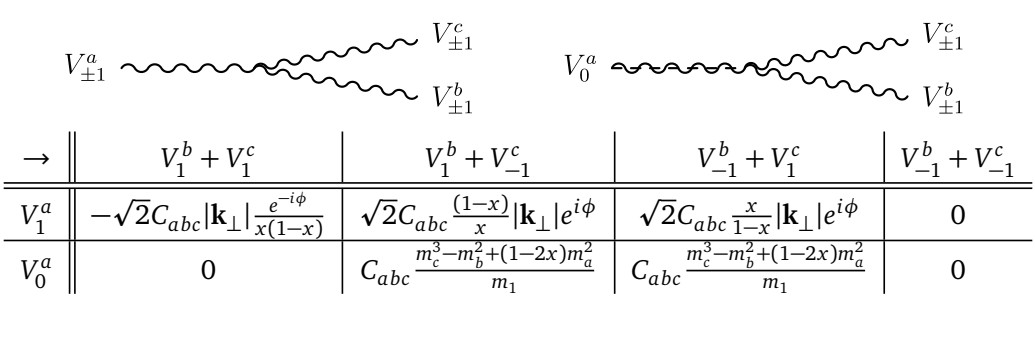

| $\rightarrow$ | $h + V_1^b$ | $h + V_{-1}^b$ | $h + V_0^b$ |
|---|---|---|---|
| $V_1^a$ | $-C\sqrt{m_a m_b}$ | $0$ | $-C\sqrt{\frac{m_a}{m_b}}|\mathbf{k}_\perp|\frac{e^{i\phi}}{\sqrt{2}}$ |

| $\rightarrow$ | $h + V_1^b$ | $h + V_{-1}^b$ | $h + V_0^b$ |
|---|---|---|---|
| $V_0^a$ | $C\sqrt{\frac{m_b}{m_a}}|\mathbf{k}_\perp|\frac{e^{-i\phi}}{\sqrt{2x}}$ | $-C\sqrt{\frac{m_b}{m_a}}|\mathbf{k}_\perp|\frac{e^{i\phi}}{\sqrt{2x}}$ | $\frac{C}{2}\left[-\frac{m_h^2}{\sqrt{m_a m_b}}+m_a\sqrt{\frac{m_a}{m_b}}(1-2x)-m_b\sqrt{\frac{m_b}{m_a}}\frac{2-x}{x}\right]$ |

- Triple Vector Splittings

| $\rightarrow$ | $V_1^b + V_1^c$ | $V_1^b + V_{-1}^c$ | $V_{-1}^b + V_1^c$ | $V_{-1}^b + V_{-1}^c$ |
|---|---|---|---|---|
| $V_1^a$ | $-\sqrt{2}C_{abc}|\mathbf{k}_\perp|\frac{e^{-i\phi}}{x(1-x)}$ | $\sqrt{2}C_{abc}\frac{(1-x)}{x}|\mathbf{k}_\perp|e^{i\phi}$ | $\sqrt{2}C_{abc}\frac{x}{1-x}|\mathbf{k}_\perp|e^{i\phi}$ | $0$ |
| $V_0^a$ | $0$ | $C_{abc}\frac{m_c^3-m_b^2+(1-2x)m_a^2}{m_1}$ | $C_{abc}\frac{m_c^3-m_b^2+(1-2x)m_a^2}{m_1}$ | $0$ |

| $\rightarrow$ | $V_1^b + V_0^c$ | $V_{-1}^b + V_0^c$ | $V_0^b + V_0^c$ |
|---|---|---|---|
| $V_1^a$ | $C_{abc}\left[\frac{m_b^2-m_a^2}{m_c}+m_c\frac{(2-x)}{x}\right]$ | $0$ | $-\frac{C_{abc}}{\sqrt{2}}\frac{m_b^2+m_c^2-m_a^2}{m_b m_c}|\mathbf{k}_\perp|e^{i\phi}$ |

| $\rightarrow$ | $V_1^b + V_0^c$ | $V_{-1}^b + V_0^c$ |
|---|---|---|
| $V_0^a$ | $-\frac{C_{abc}}{\sqrt{2}}\frac{m_a^1+m_c^2-m_b^2}{m_a m_c(1-x)}|\mathbf{k}_\perp|e^{-i\phi}$ | $\frac{C_{abc}}{\sqrt{2}}\frac{m_a^1+m_c^2-m_b^2}{m_a m_c(1-x)}|\mathbf{k}_\perp|e^{i\phi}$ |

| $\rightarrow$ | $V_0^b + V_0^c$ |
|---|---|
| $V_0^a$ | $\frac{C_{abc}}{2m_a m_b m_c}\left[m_a^2(m_b^2+m_c^2-m_a^2)(1-2x)-m_b^2(m_a^2+m_c^2-m_b^2)\frac{1+x}{1-x}+m_c^2(m_a^2+m_b^2-m_c^2)\frac{2-x}{x}\right]$ |

- Scalars and Transverse Vector (absent in the SM)

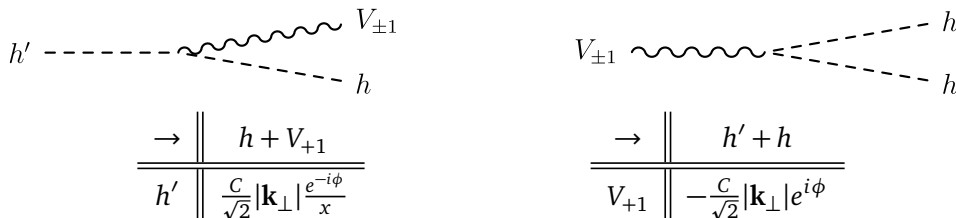

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
