# Peer review of "Goldstone Equivalence and High Energy Electroweak Physics"

_SciPost Physics, doi:SciPost Phys. 8, 078 (2020)_

## Round 1 · Referee Report · Anonymous · 2020-2-14

Strengths

1- The manuscript is well-written and the presentation is organized in a pedagogic way (from toy model to general gauge theory)
2- The authors provide all necessary ingredients to directly apply their formalism in computations of SM processes

Report

In the present work the authors develop a Lorentz covariant formalism in which longitudinal polarization vectors do not grow with energy, such that energy power-counting is manifest at the level of individual Feynman diagrams. This is achieved by formalizing and extending the Goldstone Boson Equivalence theorem with the help of generalized Slavnov-Taylor identities. The formalism is first introduced in the Higgs-Kibble model as a pedagogic example and then extended to general spontaneously broken gauge theories.

The second part of the manuscript is devoted to applications of this “Goldstone Equivalent” formalism, which rely on its manifest energy power-counting at the diagram level. The authors apply their formalism to identify and compute the leading diagrams in the high energy limit of tree-level longitudinal vector boson scattering and radiative corrections to the top quark decay. As a second applications the factorization at tree level of collinear emissions into universal splitting functions and a hard subprocess is shown.

The manuscript is well-structured and clearly written. The "Goldstone Equivalent" formalism is interesting and useful, both for the simplification of explicit computations in the high energy limit and for more theoretical considerations, such as proving factorization without having to rely on axial, i.e. non-covariant, gauges. The applications in the second part of the manuscript also provide the necessary ingredients to directly apply the formalism to different computations in the SM.

For these reasons I recommend the manuscript for publication in its present form. Below I list a couple of optional changes that in my opinion would improve the quality of the manuscript even further.

Requested changes

1- On page 33 the authors describe the power counting rules for their Goldstone-Equivalent formalism in the text. However, as the manifest power counting at the Feynman diagram level is one of the main results, it would be worth to emphasize this result by collecting and presenting the power counting rules for the vertices in a graphical / diagrammatic form (similarly to Figure 5).

2- While reading the manuscript I found the following typos:
a) Eq. (2.17): the “general Slavnov-Taylor identity” allows an additional gauge invariant operator O as in Eq. (2.6)
b) Page 10: “...subtracting the scalar polarization vector $\mathcal{K}_M$ TO…” -> from
c) Page 16: “Giving for granted…” ,”Given this for granted…” -> “Taking for granted…”, “Taken for granted…"
d) Eq. (3.9): M,N should be contracted
e) Eq. (3.21): $f_{a \dot{I}}$ -> $f_{b \dot{I}}$
f) Page 38: “… is much smaller THAT…” -> than
g) Page 42: “...compatibly with what found in Ref…” -> ...compatible with what was found in Ref...
h) Page 54: “bosonic vertices involving at leas A scalar particle.” -> one

  • validity: high
  • significance: top
  • originality: high
  • clarity: high
  • formatting: excellent
  • grammar: good

Author:  Luca Vecchi  on 2020-03-21  [id 774]

(in reply to Report 1 on 2020-02-14)

We thank the referee for carefully reading the manuscript and for his/her comments. Below is a summary of the changes made to the paper (see version2). We believe that thanks to these corrections our paper has gained in clarity.

1) We agree with the referee that a table with the energy power-counting associated to each vertex can improve the presentation. This is now done in Figure 5. We stress however that the Feynman rules (and the associated energy scalings) are exactly the same as the standard ones, since our formalism does not affect the Lagrangian itself. What the Goldstone Equivalent formalism does is: (1) it instructs us to follow the prescription summarized in Fig.1 and replace the polarization vector for external vector lines with ${\rm e}^\mu_0$, (2) it makes energy power-counting manifest, thus allowing us to identify unambiguously which diagrams contribute at each order in $m/E$.

2) The typos b-h) have been corrected. We appreciate the effort the referee has put in catching them. Regarding a), before (2.17) we clarified that we refer to eq. (2.6) with $O=1$. In addition, we noticed that our convention for the connected amplitudes implicitly assumed that the Dirac delta function for momentum conservation was removed. We added a brief comment about this below (2.17).

---

## Round 1 · Referee Report · Anonymous · 2020-2-28

Strengths

1. A new formalism to power-count the high-energy behavior of gauge bosons amplitudes is presented.
2. Applications to Standard Model calculations and equivalence with standard techniques are discussed.
3. The paper is generally well-written.

Weaknesses

1. I find that sometimes the discussion is quite pedantic. A few passages could be simplified in favor of a clearer message. I discuss more in the Requested changes form.

Report

By exploiting the consequences of the Slavnov-Taylor identities for gauge-fixing function operators, the authors introduce a new formalism where longitudinal polarization of gauge bosons are well-behaved at high-energy.
This provides a useful tool to characterize the leading contribution of single Feynman diagrams, both for on-shell and off-shell vectors. This new formalism is extensively explained in Section 2, in the context of the Higgs-Kibble model and then it is extended for general gauge theories in Section 3. The faith of unstable vectors (particularly relevant for Standard Model processes) is also discussed very clearly.

The authors apply this formalism to the Standard Model and show its equivalence with the standard techniques. In particular, they discuss applications to $WW$ scattering and radiative corrections to top decay. As another application, in the last section the authors provide proof of collinear factorization at tree-level and the splitting functions in the Standard Model are computed.

The paper is interesting, with a potential impact on the scientific community. I recommend it for publication although I suggest making some changes.

Requested changes

1. I would like the authors to improve the clarity of the main text in page 33 regarding the $WW$ scattering. As it is written now, the discussion about the identification of the diagrams contributing to $\mathcal{M}_0$ and $\mathcal{M}_1$ is confusing and hard to follow. It can help to add the explicit diagrams by using the corresponding Feynman rules of the Goldstone-Equivalent formalism (e.g. those presented in Figure 1).

2. Although according to the authors the application of this formalism is straightforward, in Section 4.4.1 it is worth showing explicitly the computation of the leading terms of the $WW$ amplitude (especially $\mathcal{M}_1$), reproducing the Standard Model result with the dependence on the weak angle. This would easily convince the reader about the usefulness of the Goldstone-Equivalent formalism.

3. I would like the authors to stress the advantages of this formalism with respect to computing the decoupling limit of the theory (e.g. using the Stuckelberg trick). For example, it is not clear if the computation of $\mathcal{M}_0$ and $\mathcal{M}_1$ is simpler with this new formalism.

4. The authors should consider refining the discussion on page 17 about the equivalence of using Eq.2.47 for generic scattering amplitudes. Although the diagrammatic relation (Fig.3) is widely explained, it may help to add a few relevant equations to clarify the discussion.

  • validity: high
  • significance: high
  • originality: high
  • clarity: good
  • formatting: good
  • grammar: good

Author:  Luca Vecchi  on 2020-03-21  [id 775]

(in reply to Report 2 on 2020-02-28)

Below is a summary of the changes made to the paper (see version2) and our replies to the referee’s comments/suggestions.

1) The diagrams contributing to ${\cal M}_{0,1}$ are explicitly shown in Fig.6 (up to obvious crossings). The Feynman rules for the vertices (and the associated energy scalings) are exactly the same as the standard ones, since our formalism does not affect the Lagrangian itself. What the Goldstone Equivalent formalism does is: (1) it instructs us to follow the prescription summarized in Fig1 and replace the polarization vector for external vector lines with ${\rm e}^\mu_0$, (2) it makes energy power-counting manifest, thus allowing us to identify unambiguously which diagrams contribute at each order in $m/E$.

The power-counting argument by which the relevant diagrams are identified is now illustrated more explicitly by referring to the new figure 5. This information plus Fig.1 readily shows that the diagrams relevant to $W_0W_0$ scattering are those in Fig.6.

2) Adding further details on the evaluation of the diagrams (already reported in Fig.6) seems to us not appropriate. Computing them is a standard exercise, and we believe it would not add much to the paper. Notice that obtaining the dependence on the weak angle does not pose issues of any sort. The sentence "Ignoring for simplicity the dependence on the weak angle, ..", which might have confused the referee, simply means that we present for brevity the power-counting treating the cosine and sine of the Weinberg angle as order unit factors (see now caption of the new figure 5). One could also easily count the powers of cosines and sines, or, which is the same, the powers of g and g' coupling separately. The above confusing sentence has been modified to be more clear.

3) The advantages of the Goldstone Equivalent formalism in computing ${\cal M}_{0,1}$ has been explained in detail. Very briefly put, their computation is not just “simpler”: those amplitudes become identifiable and directly accessible (associated with Feynman diagrams) only within our formalism. More generally, the advantages of our formalism (massless integrals in loops, the possibility of selecting the relevant diagrams based on energy scaling and the hierarchy of couplings, the unique possibility of proving factorisation in a covariant formalism, ...) are widely explained in the paper and supported by several concrete examples and results.

We do not understand why the referee is mentioning the Stueckelberg trick: the Goldstones are already in the Lagrangian and need not to be re-introduced. Since he/she also mentioned the decoupling limit, he/she has probably in mind theories (unlike the SM) where the Higgs is heavy or absent. It is evident from the general treatment in section 3 that we could easily deal with such theories. In particular one can employ the Stueckelberg non-linear representation for the Goldstone fields and all derivations of section 3 (which does not rely on a specific Lagrangian) would go through. We see no reason to discuss theories with heavy Higgs explicitly, since the SM Higgs is light. If instead he/she refers to the "gauge-less limit", this is readily obtained in our formalism (since coupling power-counting is manifest) by setting the gauge coupling to zero. This would of course not be legitimate in the calculation of WW scattering since the gauge contribution is sizeable for light Higgs. In summary, our formalism goes well beyond (and trivially incorporates) the approaches mentioned by the referee.

4) We debated at length on how to present the proof that all propagators in resonant diagrams can be replaced by $G^{\rm eq}$ given in Eq. (2.46) in a fully rigorous and still accessible way, and concluded that the best way was to use Fig. 3 recursively, as discussed in the paper. The figure is fully equivalent to an equation, however turning it into an equation would require introducing a heavy notation to denote the various objects, which on the other hand are much more intuitively represented as diagrams. In view of this, we do not think that adding equations would help. One might in principle add another figure showing the first iteration of the recursion, however this would be a huge figure that we judge unnecessary.

---

## Round 3 · Referee Report · Anonymous (Referee 1) · 2020-4-21

Report

I recommend that the manuscript is published in its present form.

---

## Round 3 · Referee Report · Anonymous (Referee 2) · 2020-4-22

Report

The authors have answered my questions and clarified my doubts. They made a few requested changes to address the issues pointed out in my previous report. I find the changes appropriate.

However, they believe the clarity of the main text cannot be improved more, especially the discussion on page 17. This fact does not have any impact on the scientific content of the paper, which I consider high-level.

Therefore, I recommend the manuscript for publication in its present form.

---

## Editorial Decision

published